# A CRISPR-Cas9 screen reveals genetic determinants of the cellular response to decitabine

Pinqi Zhang [ID][1,2,5], Zhuqiang Zhang [ID][1,5], Yiyi Wang[1,2], Wenlong Du[1], Xingrui Song[3], Weiyi Lai[3], Hailin Wang [ID][3], Bing Zhu [ID][1,2,4✉] & Jun Xiong [ID][1✉]

## Abstract

**Decitabine (DAC), a well-recognized DNA hypomethylating agent, has been applied to treat acute myeloid leukemia. However, clinic investigations revealed that DNA methylation reduction does not correlate with a clinical response, and relapse is prevalent. To gain a better understanding of its anti-tumor mechanism, we perform a temporally resolved CRISPR-Cas9 screen to identify factors governing the DAC response. We show that DNA damage generated by DNMT-DNA adducts and 5-aza-dUTP misincorporation through the dCMP deaminase DCTD act as drivers of DAC-induced acute cytotoxicity. The DNA damage that arises during the next S phase is dependent on DNA replication, unveiling a trans-cell cycle effect of DAC on genome stability. By exploring candidates for synthetic lethality, we unexpectedly uncover that KDM1A promotes survival after DAC treatment through interactions with ZMYM3 and CoR-EST, independent of its demethylase activity or regulation of viral mimicry. These findings emphasize the importance of DNA repair pathways in DAC response and provide potential biomarkers.**

**Keywords** Decitabine; DNA Damage; DCTD; DNA Methylation; KDM1A
**Subject Categories** Cancer; DNA Replication, Recombination & Repair

## Introduction

Decitabine (DAC, 5-aza-dC) has emerged as a cornerstone of single-agent therapy for acute myeloid leukemia (AML) and myelodysplastic syndrome (MDS) (Jones et al, 2019). Although initial high-dose regimens were proved to be ineffective due to the high toxicity (Taby and Issa, 2010), its subsequent identification as DNA methylation inhibitor paved the way for its clinical success as a hypomethylating agent (HMA) (Jones and Taylor, 1980; Sato et al, 2017). Recent studies revealed a viral mimicry response triggered by low-dose of DAC treatment (Chiappinelli et al, 2015; Mehdipour et al, 2020; Roulois et al, 2015). In addition, DAC at

clinically relevant concentration could induce abnormal mitosis, leading to apoptosis of human AML cells (Yabushita et al, 2023). These newly discovered mechanisms suggest that DAC may exert its anti-tumor effects through a broader range of pathways than previously understood. Moreover, DAC increases the expression of neoantigens and activates cancer-testis antigens that are silenced by DNA methylation (Brocks et al, 2017; Li et al, 2014a). Such effect has been exploited to augment cancer immunotherapy in both preclinical and clinical settings (Jones et al, 2019).

However, its anti-tumor activity cannot be solely attributed to DNA demethylation. Clinical investigations in AML patients with relapse showed limited association between the extent of DAC-induced DNA demethylation and clinical response (Welch et al, 2016). Investigations into the impact of mutations in DNA methylation-related genes like *DNMT3A*, *TET2*, and *IDH1/2* on treatment response yielded inconsistent results (Braun et al, 2011; DiNardo et al, 2014; Emadi et al, 2015; Itzykson et al, 2011; Traina et al, 2014). Most recent research showed the newly developed reversible DNA methyltransferase 1 (DNMT1) inhibitor GSK-3484862 is ineffective to elicit a robust inhibition on cell proliferation (Azevedo Portilho et al, 2021; Chen et al, 2023). This result further decouples DNA demethylation ability from the cellular toxicity of DNMT inhibitors. Therefore, a more comprehensive elucidation on the diverse mechanisms of DAC's anti-tumor activity is urgent for its better application in clinic and to provide reliable biomarkers.

DAC covalently traps DNMTs upon attempted methylation reaction after incorporating into genomic DNA (Juttermann et al, 1994; Santi et al, 1984), leading to the formation of DNMT-DNA adduct and subsequent DNA damage. The formation of and how to repair DAC-induced DNA lesions have been attracting more and more attentions (Borgermann et al, 2019; Carnie et al, 2024b; Liu et al, 2024; Liu et al, 2021; Weickert et al, 2023). However, conflicting results exist regarding the spectrum of mutations observed in cell models treated with DAC (Jackson-Grusby et al, 1997; Maslov et al, 2012; Oz et al, 2014), puzzling the precise role of DAC-induced DNA damage in its therapeutic efficacy. In addition, it remains largely unknown that how the DNA demethylation activity of DAC interacts with its ability to induce DNA damage, and collectively how they influence the cellular response to DAC.

[1]State Key Laboratory of Epigenetic Regulation and Intervention, Institute of Biophysics, Chinese Academy of Sciences, Beijing 100101, China. [2]College of Life Sciences, University of Chinese Academy of Sciences, Beijing 100049, China. [3]The State Key Laboratory of Environmental Chemistry and Ecotoxicology, Research Center for Eco-Environmental Sciences, Chinese Academy of Sciences, Beijing 100085, China. [4]New Cornerstone Science Laboratory, Institute of Biophysics, Chinese Academy of Sciences, Beijing 100101, China. [5]These authors contributed equally: Pinqi Zhang, Zhuqiang Zhang. ✉E-mail: zhubing@ibp.ac.cn; xiongjun@ibp.ac.cn

Despite its success in AML and MDS, less than half of patients respond to DAC due to intrinsic and acquired resistance (Sato et al, 2017). Identifying responders remains a challenge. Currently, the clinical impact of DAC is primarily confined to hematological malignancies (Jones et al, 2019; Sato et al, 2017). Expanding its therapeutic applicability to other cancer types would significantly enhance its overall impact. Combination strategies with other epigenetic drugs, particularly histone deacetylase (HDAC) inhibitors, hold promise in this regard. However, so far, clinical trials have not shown definitive improvements yet (Prebet et al, 2014). Rationally designed combination regimens, informed by a deeper understanding of underlying mechanisms, are needed to unlock the full potential of DNMT inhibitors.

In this work, we carry out a genome-wide CRISPR-Cas9 knockout (KO) screen with temporal resolution to comprehensively elucidate mechanisms underlying the action of DAC. Our findings present a paradigm shift, indicating that DNA damage generated by DNMT-DNA adducts and 5-aza-dUTP misincorporation, rather than DNA demethylation, serves as the primary drivers of DAC-induced cytotoxicity. The extent of DNA damage is regulated by both global DNA methylation and DNMT protein abundance, as reducing either factor confers resistance to DAC. The kinetics of γH2A.X after a DAC pulse reveals that the massive DNA damages induced by DAC requires DNA replication in the next S phase. Furthermore, our study uncovers a pro-survival role for KDM1A in response to DAC, independent of its enzymatic activity and regulation of the viral mimicry state. Combined targeting of KDM1A for degradation and DAC treatment exhibited effective synthetic lethality in cell models.

# Results

## A genome-wide CRISPR KO screen identifies genes that modulate DAC response

To comprehensively identify genes that regulate cellular sensitivity and resistance to DAC, we performed a genome-wide CRISPR-Cas9 knockout screen with temporal resolution in HAP1 cells. We generated a monoclonal HAP1 cell line (HAP1-Cas9-F5; Fig. EV1A), which constitutively expresses FLAG-tagged Cas9 protein, by integrating the Cas9 expression cassette into the AAVS1 locus. HAP1-Cas9-F5 cells showed a dose-dependent response to DAC, which is readily dampened by depleting DCK (Fig. EV1B), the deoxycytidine kinase responsible for DAC's activation. HAP1-Cas9-F5 cells transduced with CRISPR knockout library were treated for 10 days with DMSO or DAC (Fig. 1A). To obtain a temporal kinetics of the effect on cellular response to DAC, we collected samples every other day and processed for next-generation sequencing (Fig. 1A). The MAGeCK maximum-likelihood estimation (MLE) algorithm (Wang et al, 2019) was used to determine the relative abundance of each single guide RNA (sgRNA) and the differential β-scores were calculated by comparing DAC to DMSO treatment at each time point (Fig. EV1C and Dataset EV1). Positively or negatively selected genes (see Methods) shared by at least two time points among Day 6, 8, and 10 were considered as high-fidelity hits during the time period of screening. With this criterion, we identified 228 and 238 positively and negatively selected high-fidelity hits, respectively (Fig. 1B,C and

Dataset EV1). Hierarchical clustering of those high-fidelity hits further revealed two interesting clusters: the loss of genes in Cluster 1 led to a rapid and lasting inhibition on proliferation in response to DAC treatment; conversely, cells with impaired function of genes in Cluster 6 gradually dominated the population, suggesting a pro-DAC effect of those positively selected genes (Fig. 1D). DAC needs to be activated through the nucleotide metabolism pathway before it can be incorporated into the genome. Indeed, gene ontology (GO) and KEGG pathway analysis revealed that Cluster 6 was enriched for genes related to pyrimidine metabolic process (Figs. 1E and EV1D). On the other hand, genes related to DNA repair were highly enriched in Cluster 1 (Figs. 1E and EV1D), about half of which could be assigned to well-defined functions in DNA damage repair, including DNA recombination, Fanconi anemia pathway, homology-directed repair (HDR), non-homologous end joining (NHEJ), and cohesin loading (Fig. EV1E). It is noteworthy that enrichment of functional terms was not unidirectional. For example, genes associated with DNA double-strand break (DSB) repair via homologous recombination (HR) function exhibited enrichment in both positive and negative selections (Fig. 1E).

Thus, our genome-wide KO screen revealed three critical networks dictating how cells respond to DAC: (1) nucleotide metabolism pathways which regulate the influx of DAC into the DNA synthesis pool, thus governing its therapeutic potential; (2) global DNA methylation and DNMT protein levels that determine the extent of DNA lesions arising from DNMT-DNA adducts (discussed in detail below); (3) DNA repair network which confers cell the ability to repair DAC-induced DNA damage, influencing cellular survival and treatment efficacy.

## Depletion of DCTD leads to cellular resistance to DAC

SLC29A1 encodes a transmembrane nucleoside transporter that mediates the cellular uptake of DAC (Agrawal et al, 2018). In our screen, SLC29A1, together with DCK, emerged as the leading hits in the positive selection (Figs. 1F,G and EV1C). SAMHD1 is a deoxynucleoside triphosphate (dNTP) triphosphohydrolase and can mitigate the anti-tumor effect of DAC by converting 5-aza-dCTP to its inactive form 5-aza-dC (Oellerich et al, 2019). As expected, we found that SAMHD1 was one of the top-ranking hits in the negative selection (Figs. 1F,G and EV1C). DCTPP1 has been recognized to deactivate DAC-triphosphate (DAC-TP, 5-aza-dCTP) and prevent DAC-induced global DNA demethylation through hydrolyzing DAC-TP to its monophosphorylated form (DAC-monophosphate, DAC-MP, 5-aza-dCMP) (Requena et al, 2016). DCTD, a dCMP deaminase on the other hand, can convert DAC-MP to 5-aza-dUMP and reduce the genomic incorporation of 5-aza-dCTP (Zhao et al, 2023). Unexpectedly, depletion of either DCTPP1 or DCTD resulted in a significant increase in cellular resistance to DAC treatment (Figs. 1H and EV2A). Uracil and its derivatives have been documented as cellular intrinsic sources of DNA damage (Fugger et al, 2021; Saxena et al, 2024; Weiner et al, 1995). Our results here indicate that 5-aza-dUTP derived from DAC-MP by DCTD would be mis-incorporated and recognized as DNA lesions (Fig. 1G). In accordance, loss of DCTD reduced the level of DAC-induced γH2A.X, a marker of DNA damage (Fig. EV2B). For validation, we generated isogenic DCTD KO clones and found that all KO clones exhibited enhanced survival under DAC treatment (Fig. EV2C,D). Such pro-DAC activity of

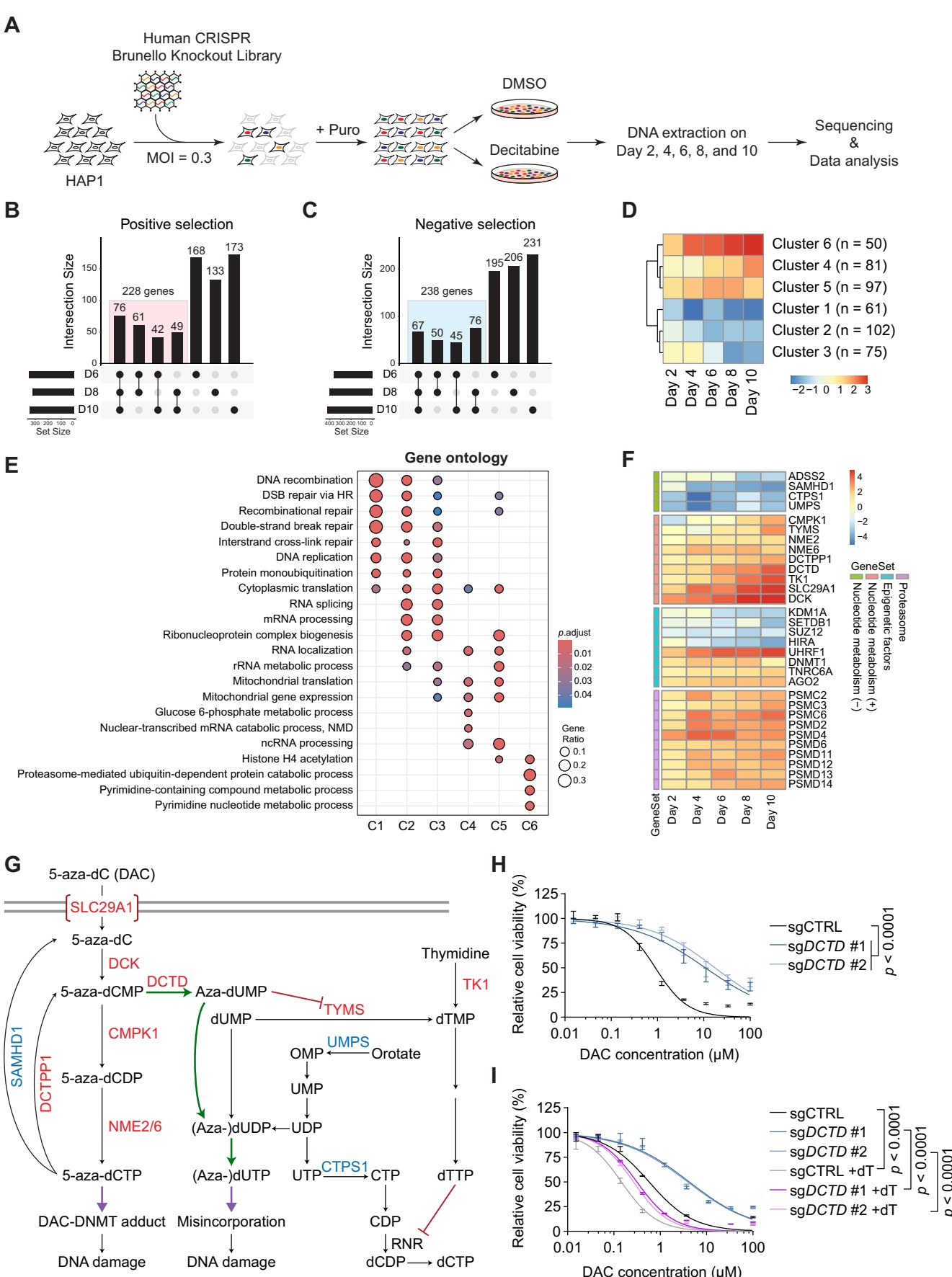

◀ **Figure 1. A CRISPR-Cas9 screen identifies genes that modulate DAC response.**

(A) Schematic of the decitabine CRISPR-Cas9 screen in HAP1 cells. (B) Upset plot of positively selected genes in the screen. Genes that were consistently identified in any two of Day 6, Day 8, or Day 10 upon DAC treatments were defined as high-fidelity positive hits. (C) Upset plot of negatively selected genes in the screen. Genes that were consistently identified in any two of Day 6, Day 8, or Day 10 upon DAC treatment were defined as high-fidelity negative hits. (D) Heatmap for averaged differential β scores of high-fidelity hits with clustering. (E) Gene ontology terms enriched in six clusters of high-fidelity hits. The $p$ values were determined using the hypergeometric distribution and adjusted by the Benjamini-Hochberg procedure. (F) Heatmap for differential β scores of manually selected representative genes. (G) Diagram of nucleotide metabolism network related to DAC. Genes labeled with red and blue were enriched in high-fidelity positive and negative selections, respectively. (H) Dose–response curves of HAP1 cells transduced with control sgRNA (sgCTRL) or two different sgRNAs targeting *DCTD* (sg*DCTD* #1 and #2) upon DAC treatment at indicated concentrations. Cell viability was measured by CellTiter-Glo after 3 days of DAC treatment. Data are presented with ± SEM ($n = 4$ replicates). The $p$ values were determined using nonlinear regression followed by the extra sum-of-squares F test. (I) Dose–response curves of control and *DCTD*-depleted cells in (H) upon DAC treatment at indicated concentrations with or without (w/o) thymidine. Cell viability was measured by CellTiter-Glo after 3 days of DAC treatment. Data are presented with ± SEM. Experiments performed in duplicates. The $p$ values were determined using nonlinear regression followed by the extra sum-of-squares F test. Source data are available online for this figure.

DCTD was also observed in Jurkat, a T lymphocyte cell line (Fig. EV2E), suggesting a general function of DCTD in regulating DAC's cytotoxicity.

Homo-deletion is the most common mutation of *DCTD* observed in tumor cell lines (Fig. EV2F, data from http://www.cbioportal.org/). Given the importance of dTTP/dCTP ratio in DAC incorporation (Gu et al, 2021), we questioned whether the DAC sensitivity could be restored in DCTD-depleted cells by manipulating the cellular dNTP pool. To test this, we supplemented cultures with non-lethal doses of thymidine (50 μM for HAP1 and 10 μM for Jurkat cells; Fig. EV2G), based on reported synergistic effect between thymidine and DAC cytotoxicity (Momparler et al, 1979; Zaharko and Covey, 1984). Notably, these thymidine additions restored DAC sensitivity in DCTD-depleted HAP1 and Jurkat cells (Figs. 1I and EV2H,I). These cells exhibited a particularly pronounced proliferation defect upon combined DAC and thymidine compared to DAC treatment alone. In corroboration with our hypothesis, depletion of thymidylate synthase (TYMS), the sole enzyme responsible for de novo dTMP production, or thymidine kinase 1 (TK1) also conferred resistance to DAC (Figs. 1F and EV2J). Together, these results reveal an underappreciated pathway involving the activity of DCTD and probable 5-aza-dUTP misincorporation that contributes significantly to the cytotoxicity of DAC. In summary, our screen results provide a holistic view of the complex cellular response to this therapeutic drug.

## Defects in DNA repair pathways sensitize cells to DAC

The formation of DNMT-DNA adducts is analogous to the PARP (poly (ADP-ribose) polymerase) trapping by PARP inhibitors (Murai et al, 2012), where trapping efficiency correlates with cytotoxicity (Murai et al, 2012; Murai et al, 2014). We compared drug sensitivities of all the cancer cell lines to DAC and three PARP inhibitors with varying PARP trapping abilities in the Cancer Dependency Map (DepMap) database (Corsello et al, 2020). Consistent with our hypothesis, we observed positive correlations between DAC sensitivity and talazoparib or olaparib (Pearson's $r = 0.478$ and 0.330, respectively), but not veliparib (Pearson's $r = 0.075$; Fig. 2A). This finding aligns with the known PARP trapping profiles of these inhibitors, where veliparib exhibits limited trapping capacity and fails to induce significant synthetic lethality in preclinical models compared to other potent trappers (Murai et al, 2012; Murai et al, 2014). Given the established efficacy of PARP inhibitors in cancers with BRCA1/2 mutation-related homologous recombination deficiencies (HRD) (Bryant et al, 2005;

Farmer et al, 2005), we explored the therapeutic potential of DAC against cancer cell lines derived from solid tumors harboring HRD mutations. Utilizing a panel of breast tumor cell lines, we observed a remarkable difference in DAC sensitivity, with *BRCA1* mutated HRD cell lines (MDA-MB-436 and HCC1937) demonstrating significantly enhanced susceptibility even at a very low drug concentration (15 nM), compared to homologous recombination proficient (HRP) cell lines (Fig. 2B).

*BRCA1* is a Fanconi anemia (FA)-associated gene. KEGG pathway analysis of our screening results revealed a significant enrichment of the FA pathway among negatively selected genes (Fig. EV1D). We individually deleted several key FA genes (*FANCA*, *FANCD2*, *FANCL*, and *UBE2T/FANCT*) that were significantly selected in our screen and found that these genetic alterations resulted in a pronounced sensitization to DAC treatment (Fig. 2C). Consistently, cells lacking *FANCG* are sensitive to DAC as well (Orta et al, 2013). Thus, compromised DNA repair ability, particularly through FA gene mutations, enhances cellular sensitivity to DAC treatment, potentially via synthetic lethality mechanisms.

The robust connection between DNA repair capacity and DAC sensitivity prompted us to investigate uncharacterized hits potentially involved in DNA damage response pathways. Depleting CTDNEP1, a serine/threonine protein phosphatase with diverse functions in membrane biogenesis, nuclear pore insertion, and genome integrity (Merta et al, 2021), conferred sensitivity to DAC treatment (Fig. EV1C). We confirmed this finding through individual sgRNA-mediated depletion of *CTDNEP1* (Fig. 2D) or its regulatory partner *CNEP1R1* in HAP1 cells (Fig. 2E). Interestingly, CTDNEP1 mainly localizes at nuclear envelop and cytosolic lipid droplets in HAP1 cells (Fig. EV3A). To verify its potential role in DNA damage repair, we employed direct-repeat green fluorescent protein (DR-GFP) (Pierce et al, 1999) and EJ7-GFP (Bhargava et al, 2018) reporter systems in HAP1 cells to monitor the efficiency of DSB repair through HR and canonical non-homologous end joining (NHEJ) pathways, respectively. Depleting CTDNEP1 significantly impaired both HR and NHEJ-mediated DSB repair (Fig. 2F,G), suggesting a previously unrecognized and critical role for this protein in maintaining genome integrity. This may contribute to the observed sensitizing effect of CTDNEP1 depletion on DAC treatment. Collectively, our results highlight the paramount contribution of DNA damage, partly arising from a common mechanism shared by other DNA-protein adducts and also the misincorporation of 5-aza-dUTP, to the acute cytotoxic effect of DAC treatment.

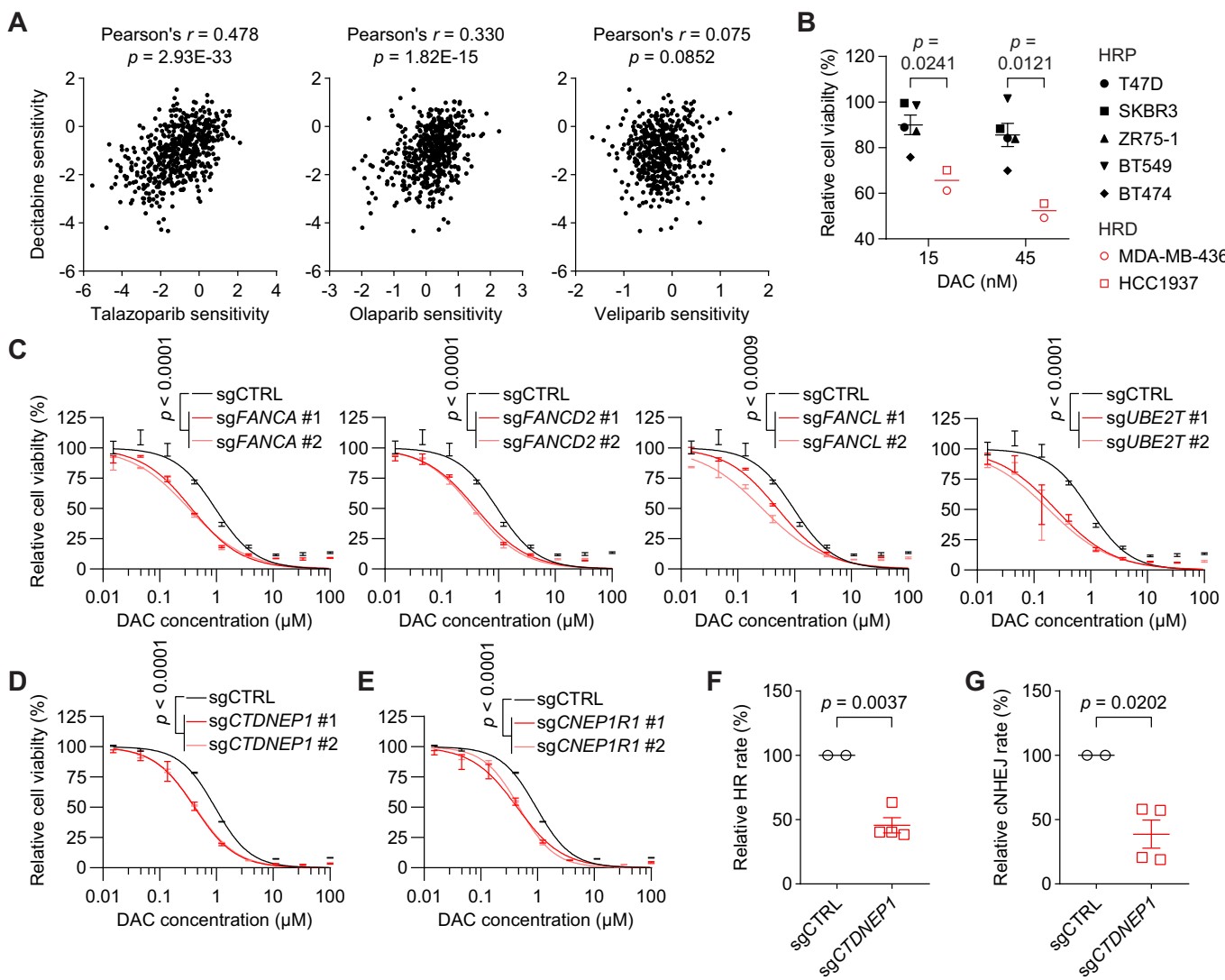

**Figure 2. DNA repair deficiency confers cellular sensitivity to DAC.**

(A) Correlation analysis between DAC sensitivity and talazoparib ($n = 558$), olaparib ($n = 552$), and veliparib ($n = 534$) sensitivity (PRISM Repurposing Primary Screen). Data were obtained from the DepMap database. Each point represents a cancer cell line. The $p$ values were determined using linear regression. (B) Survival analysis of indicated breast cancer cell lines after treatment with indicated doses of DAC for 8 days. Cell viability was measured by CellTiter-Glo. HRP, homologous recombination proficient; HRD, homologous recombination deficient. Data represent the means ± SEM (HRP cell lines, $n = 5$; HRD cell lines, $n = 2$). The $p$ values were determined using two-tailed unpaired $t$-test. (C) Dose–response curves of HAP1 cells transduced with sgCTRL or sgRNAs targeting *FANCA*, *FANCD2*, *FANCL*, or *UBE2T* (each by two different sgRNAs, #1 and #2) upon DAC treatment at indicated concentrations. Cell viability was measured by CellTiter-Glo after 3 days of DAC treatment. Data are presented with ± SEM. Experiments performed in duplicates. The $p$ values were determined using nonlinear regression followed by the extra sum-of-squares F test. (D) Dose–response curves of HAP1 cells transduced with sgCTRL or two different sgRNAs targeting *CTDNEP1* (sg*CTDNEP1* #1 and #2) upon DAC treatment at indicated concentrations. Cell viability was measured by CellTiter-Glo after 3 days of DAC treatment. Data are presented with ± SEM. Experiments performed in duplicates. The $p$ values were determined using nonlinear regression followed by the extra sum-of-squares F test. (E) Dose–response curves of HAP1 cells transduced with sgCTRL or two different sgRNAs targeting *CNEP1R1* (sg*CNEP1R1* #1 and #2) upon DAC treatment at indicated concentrations. Cell viability was measured by CellTiter-Glo after 3 days of DAC treatment. Data are presented with ± SEM. Experiments performed in duplicates. The $p$ values were determined using nonlinear regression followed by the extra sum-of-squares F test. (F) Homologous recombination (HR) efficiency in HAP1 DR-GFP reporter cells transduced with sgCTRL or sgRNAs targeting *CTDNEP1* (two different sgRNAs, #1 and #2; each measured twice, $n = 2$). Data represent the means ± SEM. The $p$ values were determined using two-tailed unpaired $t$-test. (G) Canonical non-homologous end joining (cNHEJ) efficiency in HAP1 EJ7-GFP reporter cells transduced with sgCTRL or sgRNAs targeting *CTDNEP1* (two different sgRNAs, #1 and #2; each measured twice, $n = 2$). Data represent the means ± SEM. The $p$ values were determined using two-tailed unpaired $t$-test. Source data are available online for this figure.

## Prior DNA demethylation and reduction in DNMT1 protein abundance confer cellular resistance to DAC

DNA methylation maintenance during S phase depends on DNMT1 and its critical co-factor UHRF1 (Bostick et al, 2007; Li et al, 1992;

Sharif et al, 2007). In our screen, deletion of either *DNMT1* or *UHRF1* rendered significant cellular resistance to DAC (Figs. 1F and EV1C). This result deviates from the established paradigm of their demethylation synergy with DAC (Giovinazzo et al, 2019; Vispe et al, 2015). Individual sgRNA-mediated *DNMT1* or *UHRF1* depletion

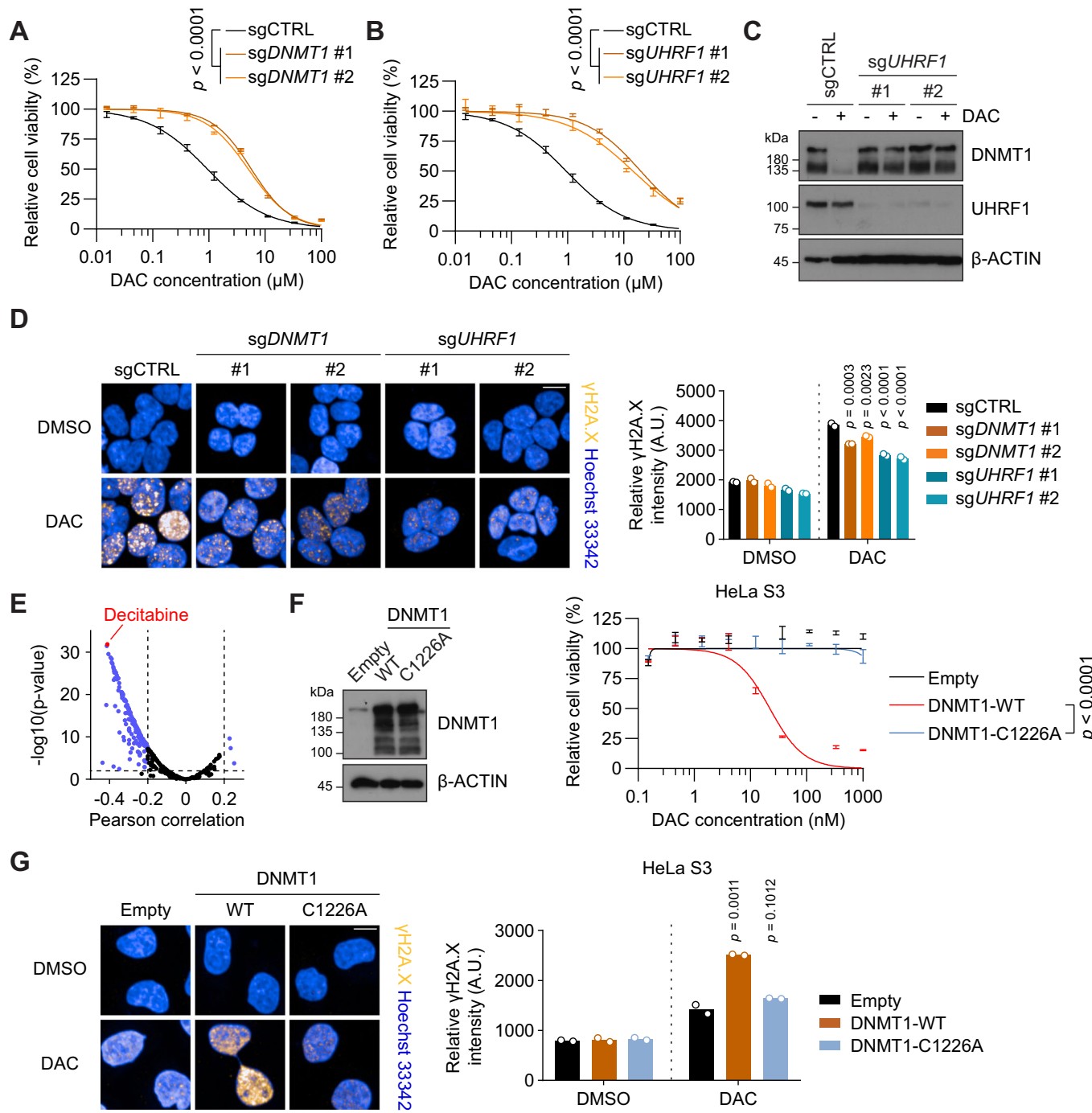

in HAP1 cells mirrored the screening results (Figs. 3A,B and EV3B,C). We reasoned that DNMT1 protein abundance and global DNA methylation levels might directly affect the extent of DNA damage elicited by DAC. UHRF1 plays a vital role in facilitating DNMT1 during both DNA replication-coupled and -uncoupled DNA methylation maintenance (Ming et al, 2020). Indeed, *UHRF1* deletion, which led to a global loss of DNA methylation (Fig. EV3D), blunted DAC-induced DNMT1 degradation (Fig. 3C). The γH2A.X signal was diminished as well in both DNMT1- and UHRF1-depleted cells upon

DAC treatment (Fig. 3D). Treating HAP1 cells with GSK-3484862, a selective DNMT1 reversible inhibitor targeting the active-site loop for substrate DNA binding (Azevedo Portilho et al, 2021; Chen et al, 2023), resulted in a significant reduction in global DNA methylation level (Fig. EV3E) and a mild decrease in DNMT1 protein abundance (Fig. EV3F). Pretreatment of HAP1 cells with GSK-3483862 enhanced cell viability (Fig. EV3G) and diminished γH2A.X signal (Fig. EV3H) in subsequent DAC treatment. Thus, global DNA demethylation from prior experience (UHRF1 depletion or DNMT1 inhibition) renders

**Figure 3.  Global DNA methylation level and DNMT1 protein abundance impact the extent of DAC-induced DNA damage.**

(A) Dose–response curves of HAP1 cells transduced with sgCTRL or two different sgRNAs targeting *DNMT1* (sg*DNMT1* #1 and #2) upon DAC treatment at indicated concentrations. Cell viability was measured by CellTiter-Glo after 3 days of DAC treatment. Data are presented with ± SEM. Experiments performed in duplicates. The *p* values were determined using nonlinear regression followed by the extra sum-of-squares F test. (B) Dose–response curves of HAP1 cells transduced with sgCTRL or two different sgRNAs targeting *UHRF1* (sg*UHRF1* #1 and #2) upon DAC treatment at indicated concentrations. Cell viability was measured by CellTiter-Glo after 3 days of DAC treatment. Data are presented with ± SEM. Experiments performed in duplicates. The *p* values were determined using nonlinear regression followed by the extra sum-of-squares F test. (C) Immunoblot analysis showing the DNMT1 protein level in control and *UHRF1*-depleted HAP1 cells in (B) w/o DAC treatment (500 nM) for 3 days with β-ACTIN as a loading control. (D) Representative images of γH2A.X immunostaining in control and *DNMT1*- or *UHRF1*-depleted HAP1 cells treated with DMSO or DAC (500 nM) for 24 h (left). Quantification of γH2A.X immunostaining signal intensity in control and *DNMT1*- or *UHRF1*-depleted HAP1 cells after DAC treatment (right). Each point represents the mean of one experiment (*N* = 2 experiments; *n* > 2000 cells for each genotype and treatment). The *p* values were determined using one-way ANOVA followed by Dunnett's multiple comparisons test. The scale bar is 10 μm. A.U., arbitrary unit. (E) Correlation analysis between DNMT1 expression level and drug sensitivity (AUC values in the CTD^2 database). Data were obtained from the DepMap database and DAC (decitabine) is highlight in red. The *p* values were determined using linear regression, *n* = 545. (F) Immunoblot analysis (left) showing the DNMT1 protein level in HeLa S3 cells transduced with empty vector (Empty) and lentiviral vectors expressing wild-type DNMT1 (WT) or catalytically inactive DNMT1-C1226A mutant with β-ACTIN as a loading control. Dose–response curves of these cells upon DAC treatment at indicated concentrations (right). Cell viability was measured by CellTiter-Glo after 3 days of DAC treatment. Data are presented with ± SEM. Experiments performed in duplicates. The *p* value was determined using nonlinear regression followed by the extra sum-of-squares F test. (G) Representative images of γH2A.X immunostaining in HeLa S3 cells overexpressing WT or C1226A mutated DNMT1 treated with DMSO or DAC (150 nM) for 24 h (left). Quantification of γH2A.X immunostaining signal intensity in these cells after DAC treatment (right). Each point represents the mean of one experiment (*N* = 2 experiments; *n* > 1800 cells for each genotype and treatment). The *p* values were determined using one-way ANOVA followed by Dunnett's multiple comparisons test. The scale bar is 10 μm. Source data are available online for this figure.

reduced sensitivity to DAC. This observation suggests that acquired DAC resistance observed in clinical settings might be attributable to decreased DNA methylation and DNMT1 protein levels by prior therapeutic regimens using this drug.

We analyzed the correlation between DNMT1 expression level and cellular sensitivity to all drugs and drug combinations (*n* = 545) in the Cancer Target Discovery and Development (CTD^2) database across a large panel of human cancer cell lines (798 cell lines in total). A striking negative correlation, which was the statistically most significant one among all drugs in the database, was revealed between AUC (Area Under the Curve) value of DAC and DNMT1 expression level (Pearson's *r* = −0.412, *p* = 1.64E−32, *n* = 761; Fig. 3E), confirming that DNMT1 protein as the primary target of DAC. To test the hypothesis that elevated DNMT1 level would augment DAC-induced DNA damage (Juttermann et al, 1994), we introduced exogenous wild-type (WT) DNMT1 or its catalytic-dead mutant (DNMT1-C1226A) into HeLa cells, which exhibited innate resistance to DAC (Fig. EV3I). Overexpression of wild-type DNMT1 resulted in a substantial cell death upon DAC treatment, even at extremely low drug concentrations (Fig. 3F), essentially converting a DAC-refractory cell line into a hypersensitive one. This effect was absent in cells expressing the catalytically inactive DNMT1 mutant, highlighting the requirement of DNMT1 catalytic activity in mediating DAC-induced DNA damage (Fig. 3F). Consistent with this mechanism, we observed a significant increase in γH2A.X signal upon DAC treatment in wild-type DNMT1, but not the catalytic-dead counterpart, overexpressing HeLa cells (Fig. 3G). Together, these findings demonstrate that the functional status of DNMT1 and global DNA methylation level jointly exert a direct and significant influence on the level of DAC-induced DNA damage and associated cytotoxicity. These results also suggest the potential of using DNMT1 expression as a biomarker for guiding DAC treatment in clinical applications.

## DAC induces an extensive DNA damage response in the second S phase

To further investigate how DAC induces DNA damage, we monitored DNMT1 protein levels and γH2A.X signals in synchronized cells following a DAC pulse. U2OS cells were synchronized in S phase by a single round of thymidine treatment and then released with or without a 30-min DAC pulse, followed by analysis at indicated time points. In line with recent studies (Carnie et al, 2024b; Liu et al, 2024; Liu et al, 2021), DNMT1 was initially trapped on chromatin and degraded quickly within a short period (Fig. 4A). However, no significant increase in γH2A.X signals was observed in DAC-pulsed cells compared to controls within the time frame for DNMT-DNA adducts removal (Fig. 4B). Instead, both DAC-treated and untreated cells exhibited a gradual reduction in γH2A.X signals, likely reflecting the clearance of DNA damages resulting from replication stress induced by the thymidine block (Fig. 4B). Notably, a pronounced increase in γH2A.X signal was detected only in DAC-treated cells 24 h after thymidine release and DAC pulse (Fig. 4B), when most cells had entered the second S phase (see below). These findings suggest a trans-cell cycle effect of DAC, promoting DNA damage and replication stress in subsequent S phase.

To test whether DNA replication in the next S phase is necessary for DAC-induced increase in γH2A.X signal, we specifically labeled S-phase cells with an EdU pulse before sample collection at different time points (Fig. 4C). This experiment showed that only cells that entered the second S phase and underwent active DNA replication (EdU positive, 18 and 20 h after release from thymidine block) displayed elevated γH2A.X levels (Fig. 4D). Furthermore, blocking cells in the subsequent G1 phase with CDK4/6 inhibitor palbociclib, which prevents entry into the next S phase, abolished the DAC-induced upregulation of γH2A.X signals (Fig. 4E). Similar results were obtained using a second round of thymidine to block cells at the G1/S boundary. Two rounds of thymidine treatment led to the accumulation of γH2A.X signals at the end point of the cell cycle synchronization, which were rapidly reduced in control cells after release from the second thymidine block for 1 h (Fig. 4F). Notably, only DAC-pulsed cells released into the second S phase exhibited a significantly higher γH2A.X signal compared to untreated controls, while cells still arrested at the G1/S boundary showed comparable γH2A.X signal levels (Fig. 4F). Together, these data suggest that DAC does not directly induce extensive DNA damage marked by γH2A.X, but requires successive DNA replication in subsequent S phases to exert its cytotoxicity.

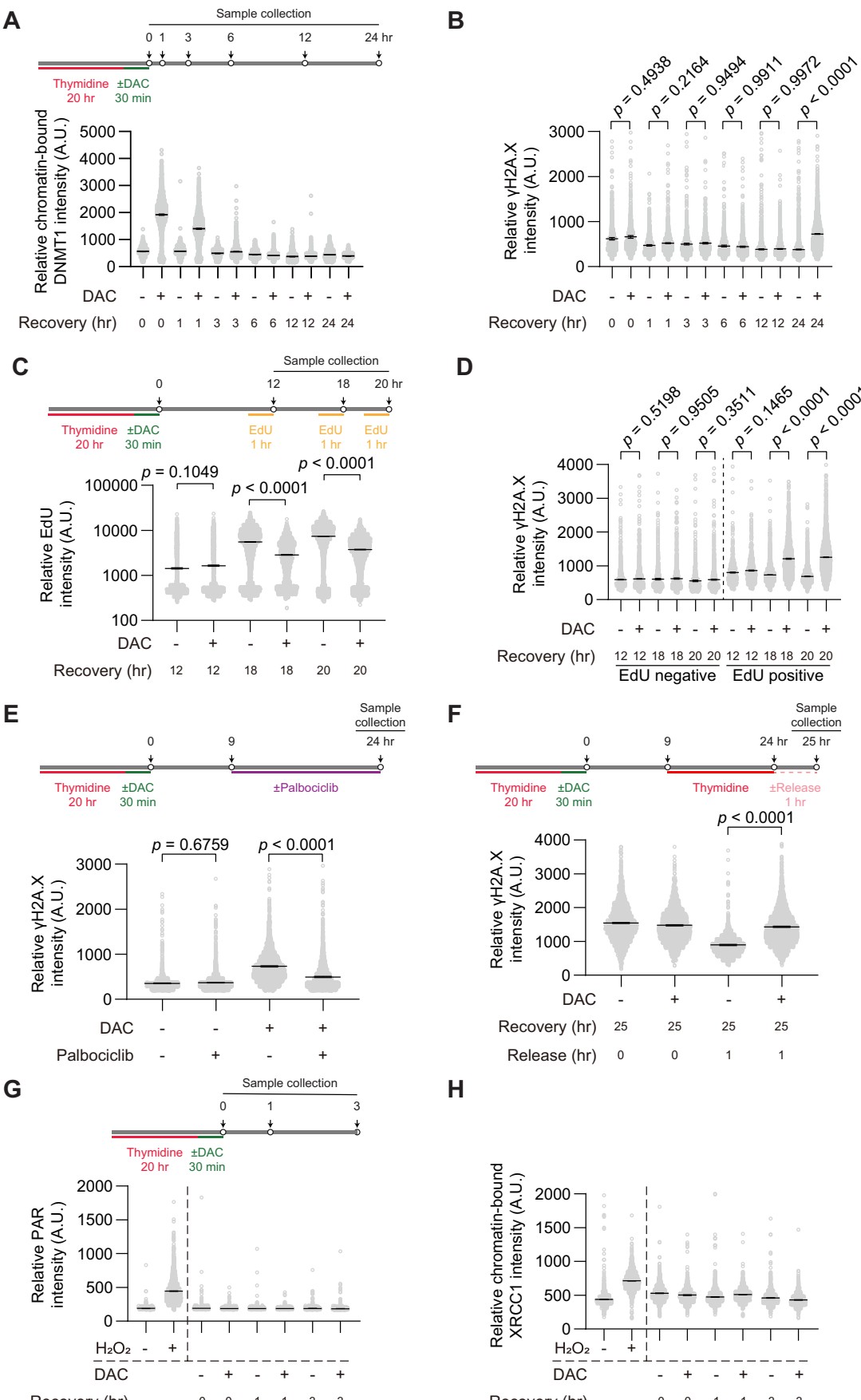

**Figure 4. Extensive DNA damage induced by DAC requires DNA replication in the next S phase.**

(A) DNMT1 protein levels in U2OS cells at the indicated time points after release from single-round thymidine synchronization, with or without a 30-min DAC (10 μM) pulse. Data represent the means ± SEM ($n > 2000$ cells for each condition). (B) γH2A.X levels in U2OS cells at the indicated time points after release from single-round thymidine synchronization, with or without a 30-min DAC (10 μM) pulse. Data represent the means ± SEM ($n > 2000$ cells for each condition). The p values were determined using one-way ANOVA followed by Sidak's multiple comparisons test. (C) EdU levels in U2OS cells at the indicated time points after release from single-round thymidine synchronization, with or without a 30-min DAC (10 μM) pulse. Prior to sample collection, cells were pulsed with 10 μM EdU for 1 h to label cells undergoing DNA replication. Data represent the means ± SEM ($n > 2500$ cells for each condition). The p values were determined using one-way ANOVA followed by Sidak's multiple comparisons test. (D) γH2A.X in U2OS cells at the indicated time points after release from single-round thymidine synchronization, with or without a 30-min DAC (10 μM) pulse. Prior to sample collection, cells were pulsed with 10 μM EdU for 1 h to label cells undergoing DNA replication. Data represent the means ± SEM ($n > 500$ cells for each condition). The p values were determined using one-way ANOVA followed by Sidak's multiple comparisons test. (E) γH2A.X in U2OS cells 24 h after release from single-round thymidine synchronization, with or without a 30-min DAC (10 μM) pulse. Palbociclib (1 μM) was added at 9 h after thymidine release to prevent progression into the next S phase. Data represent the means ± SEM ($n > 4000$ cells for each condition). The p values were determined using one-way ANOVA followed by Sidak's multiple comparisons test. (F) γH2A.X in U2OS cells 25 h after release from the first round of thymidine synchronization, with or without a 30-min DAC (10 μM) pulse. A second round of thymidine block was applied 9 h after the first release to arrest cells at the G1/S boundary. Prior to sample collection, cells were released into the subsequent S phase for 1 h (25 h after the initial thymidine release). Data represent the means ± SEM ($n > 1500$ cells for each condition). The p values were determined using one-way ANOVA followed by Sidak's multiple comparisons test. (G) PARylation levels in U2OS cells at the indicated time points after release from single-round thymidine synchronization, with or without a 30-min DAC (10 μM) pulse. A 30-min $H_2O_2$ (100 μM) treatment was used as a positive control. Data represent the means ± SEM ($n > 1900$ cells for each condition). (H) Chromatin-bound XRCC1 levels in U2OS cells at the indicated time points after release from single-round thymidine synchronization, with or without a 30-min DAC (10 μM) pulse. A 30-min $H_2O_2$ (100 μM) treatment was used as a positive control. Data represent the means ± SEM ($n > 1900$ cells for each condition). Source data are available online for this figure.

Since both DNMT-DNA adducts and 5-aza-dU contribute to DAC-induced DNA damage, it is possible that the DNA-damaging intermediates derived from DAC could be processed by alternative DNA repair pathways, such as the single-strand break (SSB) repair pathway. This is particularly relevant because processing proximal SSBs can lead to DSBs, and unrepaired SSBs can result in replication stress. To investigate whether the SSB repair pathway is involved in the early processing of DAC-induced DNA damage, we monitored the levels of PARylation catalyzed by PARP1 and chromatin-bound XRCC1 as markers of SSB repair activity following a DAC pulse in synchronized cells. While $H_2O_2$ treatment, used as a positive control, efficiently elevated the levels of both PARylation and chromatin-bound XRCC1, the DAC pulse had no significant effect on either of these markers (Fig. 4G,H). These results indicate that DAC may not immediately trigger a strong SSB response after incorporation into the genome. Consistently, our screening, along with a recent work, identified two upstream components of TC-NER, CSA and CSB, but not other downstream endonucleases that influence cellular response to DAC (Dataset EV1) (Carnie et al, 2024a). Nevertheless, these results do not rule out the possibility that SSB repair could contribute to mitigating DAC-induced DNA damage at a lower, more subtle level.

## Loss of KDM1A but not its catalytical activity increases sensitivity to DAC

The activation of viral mimicry by DAC-induced DNA demethylation has been exploited in combined cancer therapy with immune checkpoint drugs to boost the anti-cancer immunity (Chiappinelli et al, 2015; Mehdipour et al, 2020; Roulois et al, 2015). Retroelements, like ERVs, LINEs, and SINEs, are typically silenced by epigenetic mechanisms and marked with repressive modifications, including DNA methylation and H3K9me3 (Greenberg and Bourc'his, 2019). Our screen identified key epigenetic factors like HIRA, KDM1A, SETDB1, and SUZ12 that regulate the cellular response to DAC (Fig. 1F). Notably, recent studies have linked the inactivation of Polycomb Repressive Complex 2 (PRC2), which

contains SUZ12 as a core subunit, to enhanced anti-tumor cytotoxicity of DAC through retroelement reactivation and the associated inflammatory response (Chomiak et al, 2024; Kim et al, 2023; Patel et al, 2022). KDM1A is the first identified histone demethylase specific for mono- and di-methylated lysine 4 of histone H3 (Shi et al, 2004), ablation of which has been shown to activate retroelement expression and induce a viral mimicry state (Sheng et al, 2018). Interestingly, it has also been shown to promote efficient DNA damage repair (Mosammaparast et al, 2013). To explore potential synthetic lethality of DAC with epigenetic factors, we focused on KDM1A. Depleting KDM1A with individual sgRNAs in HAP1 and JeKo-1, a mantle cell lymphoma, cells led to increased cell deaths upon DAC treatment compared to control cells (Figs. 5A and EV4A,B). Building upon this, we generated two monoclonal KDM1A knockout HAP1 cell lines using the CRISPR-Cas9 strategy (Fig. EV4C). These KO clones also showed enhanced DAC sensitivity (Fig. 5B), suggesting the notion of KDM1A acting as a pro-survival factor in the context of DAC treatment.

To determine whether KDM1A's pro-survival role during DAC treatment stemmed from its demethylase activity, first, we tested the combined effect of KDM1A inhibitors with DAC on cell viability. ORY-1001 and GSK2879552 are KDM1A catalytic inhibitors, however, we did not observe any evident synergistic effect between either inhibitor and DAC in the term of cell death (Fig. EV4D,E). Then we complemented KDM1A KO clones with wild-type KDM1A or two catalytically inactive mutants: A539E/K661A (AE/KA) and D553A/D555A/D556A (3DA) (Fig. EV4C). Restoring wild-type KDM1A reversed the enhanced DAC sensitivity in KDM1A-depleted clones (Fig. 5C). Surprisingly, complementation with either catalytic-dead mutant elicited an effect similar to wild-type KDM1A (Fig. 5C). These results indicate that KDM1A's pro-survival role in this context is independent of its demethylase activity. Thus, targeting KDM1A for degradation emerges as a necessary and promising therapeutic strategy for potential combination therapy. As a proof-of-concept, we generated two knock-in clones expressing KDM1A fused to the FKBP12(F36V) degron tag (FKBP-KDM1A) for inducible degradation (Nabet et al, 2018). Acute and potent KDM1A depletion was

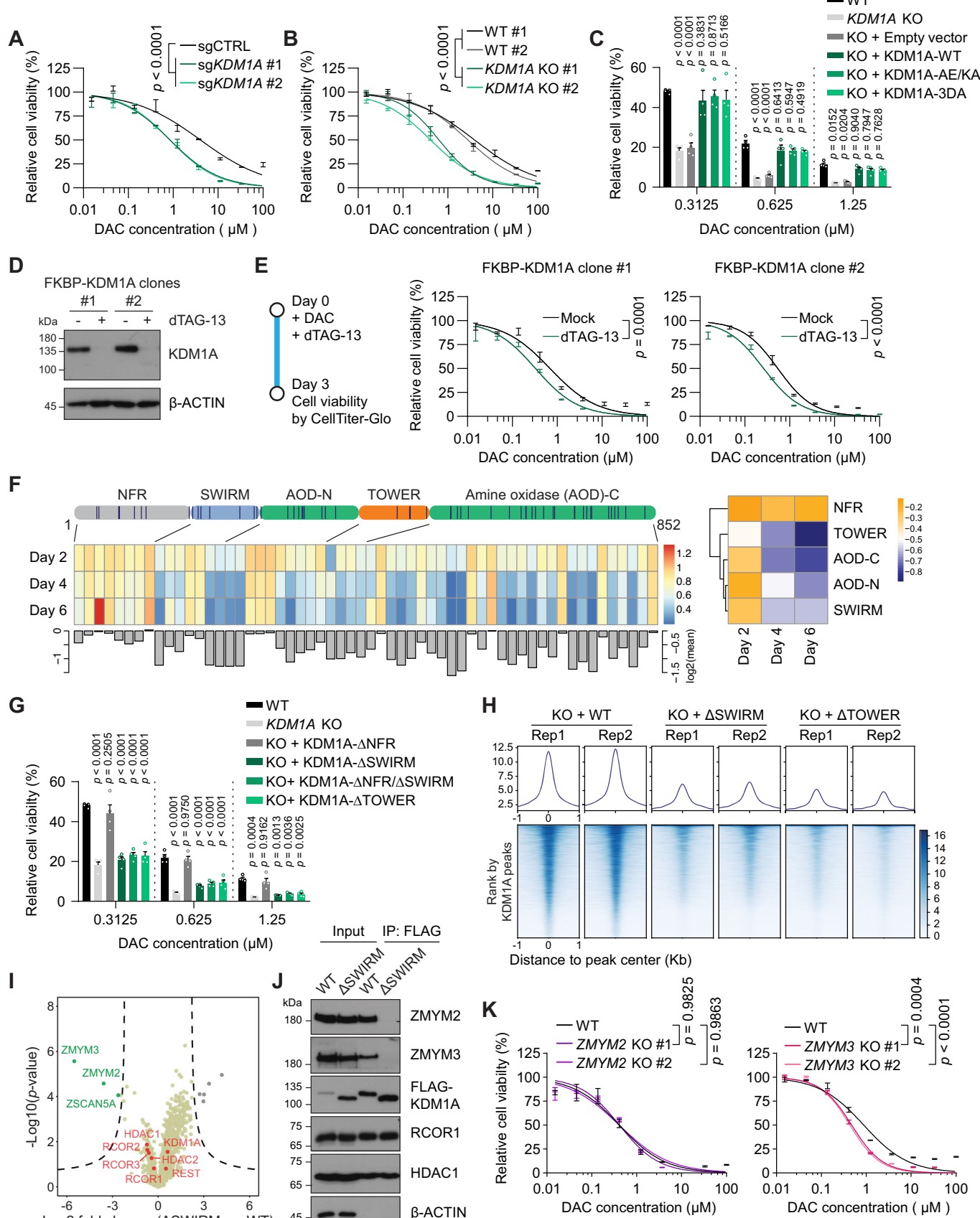

**Figure 5. Loss of KDM1A protein, but not its demethylase activity, sensitizes cells to DAC.**

(A) Dose–response curves of HAP1 cells transduced with sgCTRL or two different sgRNAs targeting *KDM1A* (sg*KDM1A* #1 and #2) upon DAC treatment at indicated concentrations. Cell viability was measured by CellTiter-Glo after 3 days of DAC treatment. Data are presented with ± SEM. Experiments performed in duplicates. The *p* values were determined using nonlinear regression followed by the extra sum-of-squares F test. (B) Dose–response curves of wild-type HAP1 cell and isogenic *KDM1A* KO clones upon DAC treatment at indicated concentrations. Cell viability was measured by CellTiter-Glo after 3 days of DAC treatment. Data are presented with ± SEM. Experiments performed in duplicates. The *p* values were determined using nonlinear regression followed by the extra sum-of-squares F test. (C) Survival analysis of WT, *KDM1A* KO clones, and *KDM1A* KO clones complemented with either empty vector, wild-type KDM1A (KDM1A-WT), or catalytically inactive KDM1A-AE/KA and KDM1A-3DA mutants after treatment with DAC at indicated concentrations for 3 days. Cell viability was measured by CellTiter-Glo. Data represent the means ± SEM of 2 biological independent clones (2 technique replicates each, *n* = 4). The *p* values were determined using two-way ANOVA followed by Dunnett's multiple comparisons test. (D) Immunoblot analysis showing the KDM1A protein level in two HAP1 clones expressing FKBP-KDM1A fusion protein treated with or without dTAG-13 for 2 days with β-ACTIN as a loading control. (E) Dose–response curves of two HAP1 clones expressing FKBP-KDM1A fusion protein upon DAC treatment at indicated concentration. Cells were treated with or without dTAG-13 at the same time with DAC. Data are presented with ± SEM. Experiments performed in duplicates. The *p* values were determined using nonlinear regression followed by the extra sum-of-squares F test. (F) Heatmap and bar plot (left) depicting sgRNA abundance changes of KDM1A domain scanning at indicated time points versus Day 0, normalized against non-targeting control sgRNAs. The positions of domain scanning sgRNAs in the coding region of *KDM1A* are indicated as vertical lines in the above schematic diagram. Heatmap (right) for averaged differential β scores of sgRNAs in KDM1A domains with clustering. (G) Survival analysis of WT, and *KDM1A* KO clones, and *KDM1A* KO clones complemented with KDM1A-ΔNFR, KDM1A-ΔSWIRM, KDM1A-ΔNFR/ΔSWIRM, and KDM1A-ΔTOWER after treatment with DAC at indicated concentrations for 3 days. Cell viability was measured by CellTiter-Glo. Data represent the means ± SEM of 2 biological independent clones (2 technical replicates each, *n* = 4). The *p* values were determined using two-way ANOVA followed by Dunnett's multiple comparisons test. (H) KDM1A ChIP-seq metagene profiles of KDM1A binding peaks in *KDM1A* KO clone complemented with KDM1A-WT, KDM1A-ΔSWIRM, and KDM1A-ΔTOWER (upper). Heatmap representation of KDM1A occupancies at KDM1A binding peaks in these cells ranked by the intensity of KDM1A ChIP-seq signal (lower). (I) Volcano plot showing differential interacted proteins between KDM1A-WT and KDM1A-ΔSWIRM obtained from label-free quantification MS with triplicates. The *p* values were determined by two-sample *t*-test using a permutation-based FDR approach. (J) Immunoblot analysis for immunoprecipitations of 3×FLAG-tagged KDM1A-WT and KDM1A-ΔSWIRM. (K) Dose–response curves of wild-type HAP1 and isogenic *ZMYM2* (left) or *ZMYM3* (right) KO clones upon DAC treatment at indicated concentrations. Cell viability was measured by CellTiter-Glo after 3 days of DAC treatment. Data are presented with ± SEM. Experiments performed in duplicates. The *p* values were determined using nonlinear regression followed by the extra sum-of-squares F test. Source data are available online for this figure.

achieved upon administration of the PROTAC dTAG-13 (Fig. 5D). Remarkably, combining dTAG-13 with DAC resulted in enhanced cell death in these cells (Fig. 5E), validating our hypothesis.

## The pro-survival role of KDM1A upon DAC treatment depends on both SWIRM and TOWER domains

As a multi-domain protein, KDM1A partners with other proteins to form a complex exhibiting both histone deacetylation, through HDAC1/2 interaction, and histone demethylation activities (Song et al, 2020). To pinpoint the domain(s) crucial for KDM1A's pro-survival role in DAC treatment, we conducted a CRISPR-Cas9 domain scanning in HAP1-Cas9-F5 cells. Cas9-induced indels have a 1/3 probability of being in-frame, potentially producing proteins with local deletions. Cells with functional in-frame mutations are likely to survive under negative selection, while those with deletions in critical regions will be sensitized. Consequently, sgRNAs targeting essential domains will be depleted during the screening (Shi et al, 2015). Our domain scanning library consisted of 58 sgRNAs spanning the *KDM1A* coding region (Fig. 5F), alongside 10 controls and 2 *DCK*-targeting sgRNAs. Notably, except for the N-terminal flexible region (NFR), targeting any other domains, including the catalytic domain, led to reduced cell viability, albeit to varying degrees (Fig. 5F). Complementation with KDM1A lacking the NFR (ΔNFR) in *KDM1A* KO cells reverted the heightened sensitivity to DAC observed, similar to that with wild-type KDM1A (Figs. 5G and EV4F). However, complementation with KDM1A lacking either SWIRM (ΔSWIRM) or TOWER (ΔTOWER) domain has no such effect (Figs. 5G and EV4F). These results indicate that the integrity of KDM1A protein, including the correct architecture of the catalytic domain, rather than its demethylase activity, is crucial for its pro-survival function in response to DAC.

The unique TOWER domain protrudes from the catalytic domain and mediates the interaction with the Linker-SANT2

segment of the CoREST protein (encoded by *RCOR1/2/3*) (Fig. EV4G) (Kim et al, 2020b; Shi et al, 2005). The interaction with CoREST is critical for the formation of KDM1A/CoREST/HDAC complex (Song et al, 2020). We derived several monoclonal *RCOR1* KO cell lines (Fig. EV4H), which exhibited slightly increased cell death upon DAC treatment (Fig. EV4I). Based on one of the *RCOR1* KO clones (*RCOR1* KO #2), we generated *RCOR1/2* double knockout cell lines (Fig. EV4J). Deleting *RCOR2* further enhanced DAC sensitivity (Fig. EV4K). Utilizing chromatin immunoprecipitation followed by sequencing (ChIP-seq) experiments, we analyzed the chromatin-binding ability of KDM1A truncations in the genome. Complete loss of KDM1A signal in KO cells confirmed the specificity of our assay (Fig. EV4L), while wild-type KDM1A complementation in KO cells restored binding to original sites (Fig. 5H). KDM1A lacking the TOWER domain exhibited significantly impaired chromatin binding (Fig. 5H), in line with its function in mediating nucleosome recognition in vitro (Kim et al, 2020b). Interestingly, the ΔSWIRM mutant displayed similar defects in chromatin binding compared to the ΔTOWER mutant (Fig. 5H), suggesting an unanticipated role of SWIRM domain in KDM1A's chromatin association.

The SWIRM domain folds back against the catalytic domain and contributes to the formation of a groove that is involved in substrate binding (Chen et al, 2006). To further unravel the function of the SWIRM domain, we compared the interactomes of WT and SWIRM-deleted KDM1A by immunoprecipitation-coupled label-free quantitative mass spectrometry. While core complex with CoREST and HDAC1/2 remained intact, ZMYM3 and ZMYM2 exhibited the most pronounced reductions in association with KDM1A upon SWIRM deletion (Fig. 5I and Dataset EV2). Western blotting of immunoprecipitated samples corroborated this finding (Fig. 5J). ZMYM2 could recruit KDM1A to its genomic targeting sites in mouse ESCs (Yang et al, 2020). Our results suggest that KDM1A interacts with ZMYM2/3 through its

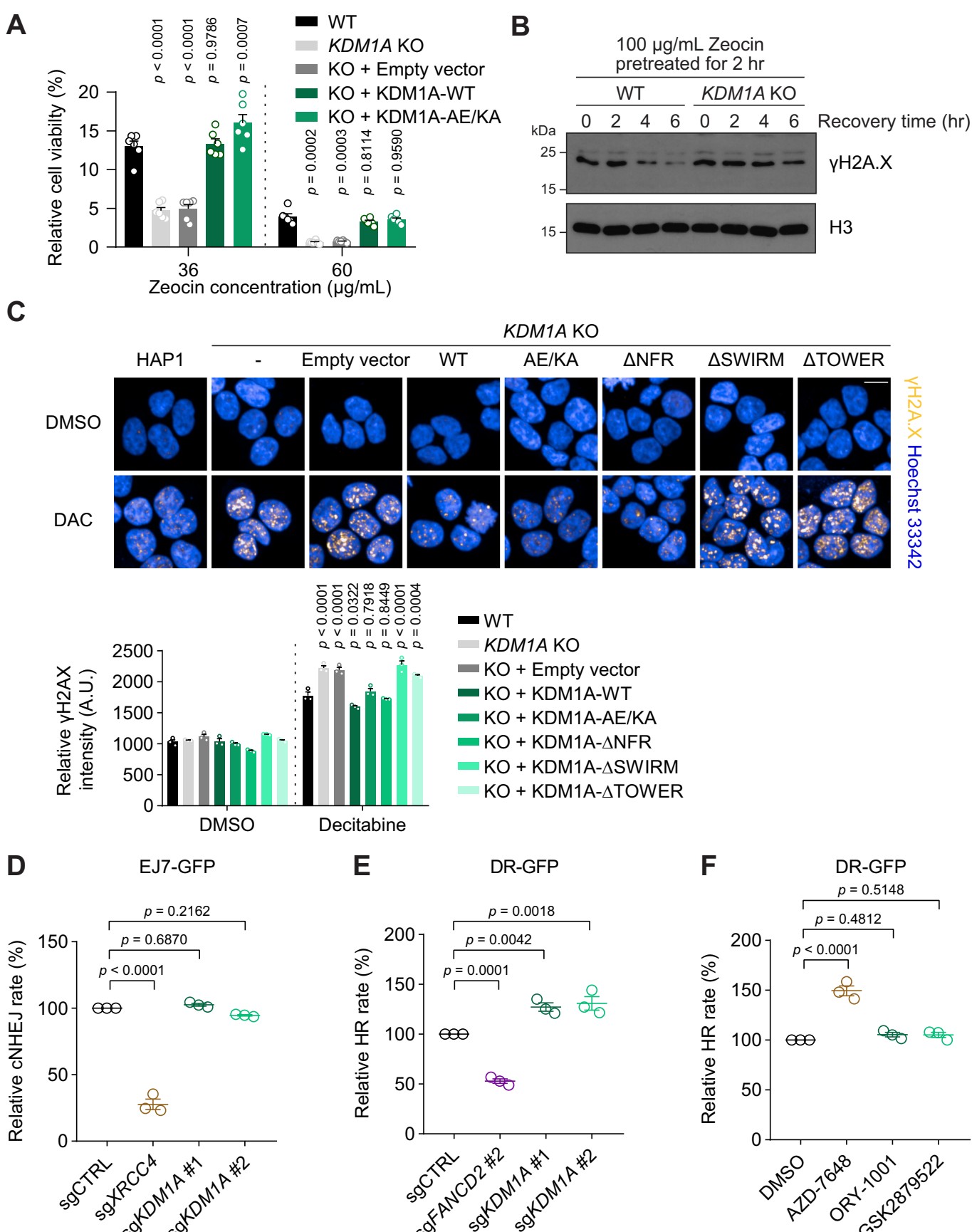

◀ **Figure 6. KDM1A confers cellular resistance to decitabine through promoting DNA repair.**

(A) Survival analysis of WT, *KDM1A* KO clones, and *KDM1A* KO clone complemented with either empty vector, KDM1A-WT, or KDM1A-AE/KA after treatment with Zeocin at indicated concentrations for 3 days. Cell viability was measured by CellTiter-Glo. Each point represents one experiment. Data represent the means ± SEM ($n = 6$ replicates). The *p* values were determined using two-way ANOVA followed by Dunnett's multiple comparisons test. (B) Immunoblot analysis showing the γH2A.X protein level in WT and *KDM1A* KO HAP1 clones harvested at indicated time points after released from prior Zeocin treatment with histone H3 as a loading control. (C) Representative images of γH2A.X immunostaining in WT and *KDM1A* KO clones and *KDM1A* KO clone complemented with empty vector, KDM1A-WT, KDM1A-AE/KA, KDM1A-ΔNFR, KDM1A-ΔSWIRM, and KDM1A-ΔTOWER treated with DMSO or DAC (150 nM) for 24 h (upper). Quantification of γH2A.X immunostaining signal intensity in these cell lines after DAC treatment (lower). Data represent the means ± SEM ($N = 3$ experiments; $n > 2500$ cells for each condition). The *p* values were determined using one-way ANOVA followed by Dunnett's multiple comparisons test. The scale bar is 10 μm. (D) Relative NHEJ efficiency in HAP1 EJ7-GFP reporter cells transduced with either sgCTRL, *XRCC4*-targeting sgRNA (sgXRCC4), or two different sgRNAs targeting *KDM1A* (sgKDM1A #1 and #2). Each data point represents an independent experiment. Data represent the means ± SEM ($n = 3$ replicates). The *p* values were determined using one-way ANOVA followed by Dunnett's multiple comparisons test. (E) Relative HR efficiency in HAP1 DR-GFP reporter cells transduced with either sgCTRL, *FANCD2*-targeting sgRNA (sgFANCD2 #2), or two different sgRNAs targeting *KDM1A* (sgKDM1A #1 and #2). Each data point represents an independent experiment. Data represent the means ± SEM ($n = 3$ replicates). The *p* values were determined using one-way ANOVA followed by Dunnett's multiple comparisons test. (F) Relative HR efficiency in HAP1 DR-GFP reporter cells treated with DNA-PK inhibitor (AZD-7648) or two KDM1A inhibitors, ORY-1001 and GSK2879552. Each data points represent an independent experiment. Data represent the means ± SEM ($n = 3$ replicates). The *p* values were determined using one-way ANOVA followed by Dunnett's multiple comparisons test. Source data are available online for this figure.

SWIRM domain to stably engage with chromatin. Interestingly, while *ZMYM2* KO showed no impact on DAC's cytotoxicity in HAP1 cells (Figs. 5K and EV4M), *ZMYM3* deficiency enhanced cell death upon DAC treatment (Figs. 5K and EV4N), hinting at distinct roles for these ZMYM proteins in regulating the cellular response to DAC.

## KDM1A deletion sensitizes cells to DAC due to DNA repair deficiency

To dissect the pathway through which KDM1A modulates the cellular response to DAC, we performed RNA-seq to analyze KDM1A's impact on gene expression and viral mimicry induced by global DNA demethylation. 238 and 101 genes are upregulated and downregulated, respectively, in *KDM1A* KO cells (Fig. EV5A). Consistent with a recent study (Zeng et al, 2023), complementation with WT or catalytically inactive (AE/KA) KDM1A both largely restored the normal expression patterns (Fig. EV5A). However, we did not find a synergistic activation of retroelements between *KDM1A* deletion and DAC treatment (Fig. EV5B,C). In addition, genes related to interferon pathways minimally altered, with 5 upregulated and 4 downregulated out of 194 genes upon *KDM1A* deletion (Fig. EV5D). Thus, the enhanced DAC cytotoxicity in *KDM1A*-depleted cells cannot be attributed to increased viral mimicry. Moreover, our analysis revealed no significant alterations in genes related to pyrimidine metabolism or DNA repair (Fig. EV5E,F). The possibility that KDM1A may regulate the extent of DAC-induced DNA damage was further ruled out by three lines of evidence: First, loss of *KDM1A*, as well as *ZMYM2*, or *ZMYM3*, did not change the proportion of cells in S phase as examined by EdU-positive cells after a short EdU pulse (Fig. EV6A,B), indicating the incorporation of DAC is unaffected by *KDM1A* KO. Second, both DNMT1 protein level and global DNA methylation were not changed by KDM1A depletion or catalytic inactivation (Fig. EV6C,D). Third, DAC-induced genome-wide demethylation remained unaffected by these same manipulations (Fig. EV6D). Furthermore, the formation and the degradation kinetics of DNMT-DNA adducts were comparable between WT and *KDM1A* KO cells (Fig. EV6E).

Both ZMYM2 and ZMYM3 are known to promote HR via 53BP1 antagonizing and BRCA1 accumulation (Lee et al, 2022; Leung et al, 2017). We hypothesized that the pro-survival role of

KDM1A during DAC treatment hinges on its direct involvement in DNA repair. To test this hypothesis, we first examined the impact of KDM1A on the kinetics of repairing DNA damages caused by other sources. We treated HAP1 cells with Zeocin, a DSB-inducing radiomimetic drug. *KDM1A* KO cells displayed heightened sensitivity to Zeocin-induced DNA damage in a three-day treatment, which would be restored by complementation with either WT or catalytic-dead KDM1A (Fig. 6A), phenocopying observations with DAC treatment. We monitored γH2A.X signal by western blot following a short-term Zeocin treatment and found a delayed clearance of DNA damage in *KDM1A*-deficient cells (Fig. 6B), indicating impaired DSB repair kinetics in the absence of KDM1A. Similarly, under DAC treatment for 24 h (to allow DAC incorporation at S phase), *KDM1A* KO cells accumulated a significantly higher level of γH2A.X (Fig. 6C). This elevated γH2A.X signal could be rescued by complementation with wild-type KDM1A, the catalytic-dead mutant, or the NRF-deleted mutant (Fig. 6C), but not the SWIRM- or TOWER-deleted KDM1A variants (Fig. 6C). Thus, both SWIRM and TOWER domains are crucial for KDM1A's role in DNA damage repair, aligning with their contributions to cellular vulnerability upon DAC treatment. To rule out the possibility that KDM1A affects early processes, such as SSB repair, following the degradation of DNMT-DNA adducts by DAC treatment, we measured PARylation and XRCC1 levels in synchronized WT and *KDM1A* KO cells after a DAC pulse. Comparable levels of PARylation and XRCC1 were observed (Fig. EV6F,G), suggesting that *KDM1A* loss does not impact potential intermediates during early repair. Finally, we examined the role of KDM1A in two major repair pathways of DSB —HR and NHEJ—using established reporter systems (Fig. 2F,G). *KDM1A* deletion had no significant impact on canonical NHEJ-mediated DSB repair as measured by the EJ7-GFP reporter assay, while depletion of XRCC4, a well-known core factor of NHEJ and DNA ligase 4 binding partner (Grawunder et al, 1998; Li et al, 1995), dramatically impaired NHEJ activity (Fig. 6D). Conversely, in the DR-GFP assay, *KDM1A* deletion led to an increased proportion of DSB repair products by HR (Fig. 6E), in accordance with previous data (Mosammaparast et al, 2013). However, while DNA-PK inhibitor AZD-7648 promoted HR efficiency as expected, treating with KDM1A enzymatic inhibitors (ORY-1001 and GSK2879552) did not affect this pathway (Fig. 6F). Collectively, these results demonstrate that KDM1A, independent of its

demethylase activity, promotes cellular survival in response to DAC through its involvement in DNA repair processes.

## Both SWIRM and TOWER domains are required for KDM1A recruitment to DNA damage sites

Considering the critical roles of SWIRM and TOWER domains in KDM1A's chromatin association, we questioned if they played a part in recruiting KDM1A to DNA damage sites. We expressed KDM1A proteins fused with enhanced GFP (EGFP) in U2OS cells (Fig. 7A). Micro-irradiation-induced KDM1A recruitment was evident at regions marked by RPA32 (Fig. 7B). While NFR deletion had no noticeable effect, ablation of either SWIRM or TOWER domain abolished KDM1A recruitment to DNA damage sites (Fig. 7B). Notably, despite comparable expression levels, the nuclear signals for ΔSWIRM and ΔTOWER mutants were eliminated due to the pre-extraction protocol for immunostaining (Fig. 7B), supporting their role in stabilizing chromatin engagement of KDM1A. To delineate the kinetics of KDM1A recruitment to DNA damage sites, we monitored the real-time mobilization of KDM1A to laser-induced lesions and found that both ΔSWIRM and ΔTOWER exhibited significantly reduced enrichment at laser-irradiated sites compared to WT KDM1A (Fig. 7C). Consistently, the colocalization of KDM1A with γH2A.X could be detected in S-phase cells using proximity ligation assay (PLA); however, the loss of either SWIRM or TOWER domain abolished this colocalization (DMSO group, Fig. 7D). Prior DAC treatment during the previous S phase increased the colocalization foci of wild-type KDM1A with γH2A.X, suggesting that DAC-induced DNA damage can recruit KDM1A to damage sites. This recruitment was absent in ΔSWIRM and ΔTOWER mutants (Fig. 7D). Hence, both SWIRM and TOWER domains are essential for KDM1A's recruitment and binding to DNA damage sites.

RCOR1 mutant (ΔLinker/ΔSANT2) lacking the KDM1A binding capacity failed to accumulate at micro-irradiation-induced DNA damage sites in U2OS cells (Fig. EV7A). The nuclear signal of this mutant was also eliminated due to the pre-extraction procedure (Fig. EV7A). Furthermore, the enrichment of WT RCOR1 was diminished in KDM1A-depleted cells (Fig. EV7B), suggesting an inter-dependent localization at DNA damage sites and chromatin binding between KDM1A and RCOR1.

On the other hand, ZMYM2 and ZMYM3 are recruited to DNA damage sites through their SUMO (small ubiquitin-related modifier) or histone binding ability, respectively (Lee et al, 2022; Leung et al, 2017). We asked whether KDM1A is recruited to DNA damage sites through its interaction with ZMYM2 and ZMYM3. To test this idea, we deleted ZMYM2 and ZMYM3 through individual sgRNA in EGFP-KDM1A expressing U2OS cells. While ZMYM2 deletion had minimal impact (Fig. 7E,F), disruption of ZMYM3 significantly impaired KDM1A recruitment to micro-irradiation-induced DNA lesions (Fig. 7G,H). The impaired recruitment of KDM1A to DNA damage sites caused by ZMYM3 depletion was also evident in DAC-pulsed cells, as observed via PLA (Fig. 7I). These findings aligned with their distinct effects on DAC-induced cell death (Fig. 5K), suggesting that ZMYM3, not ZMYM2, facilitates KDM1A recruitment to DNA damage sites, thereby promoting DNA repair and cell survival in response to DAC treatment.

However, loss of ZMYM3 only partially reduced KDM1A localization at DNA damage sites (Fig. 7G–I), consistent with the milder phenotype observed in ZMYM3 KO cells compared to KDM1A KO cells (Fig. 5A,B,K). This suggests that redundant mechanisms exist to recruit KDM1A to DNA damage sites. Supporting this notion, deleting KDM1A in ZMYM3 KO cells further sensitized cells to DAC (Fig. EV7C), whereas ablation of ZMYM3 in KDM1A KO showed no synergistic effect (Fig. EV7D). To explore whether the interaction with KDM1A facilitates ZMYM3 recruitment to DNA damage sites, as seen with RCOR1 (Fig. EV7A), we expressed EGFP-tagged wild-type ZMYM3 or a mutant lacking zinc fingers 8 and 9 (ZMYM3 ΔZNF8&9), which mediate its interaction with KDM1A (Shapson-Coe et al, 2019). Both WT and ΔZNF8&9 mutants were efficiently recruited to DNA damage sites induced by micro-irradiation (Fig. EV7E,F) or DAC (Fig. EV7G). These findings indicate that ZMYM3 operates upstream of KDM1A at DNA damage sites to promote its recruitment.

## Discussion

Using a genome-wide CRISPR-Cas9 knockout screen with temporal resolution, we identified genetic determinants of DAC sensitivity and resistance. Our screen uncovered three key networks intricately linked in governing the cellular response to DAC treatment: nucleotide metabolism pathways, alterations in which influence the availability of nucleotides; DNA methylation maintenance machineries, dysregulation of which may impact the formation and persistence of DNMT-DNA adducts; and DNA repair pathways that render cells resilient to the DNA damage inflicted by DAC.

Both DNMT1 and UHRF1 deletion rendered cells resistance to DAC treatment (Fig. 3A,B), but the mechanisms differed. We established DNMT1 as the pivotal target of DAC, causing DNA damage and ultimately cell death. The striking positive correlation between DNMT1 expression and DAC sensitivity across diverse cancer cell lines suggest that the pro-DAC role of DNMT1 is not cell-type specific and underscore the potential of DNMT1 expression as a valuable biomarker to predict patient response to DAC therapy (Fig. 3E). On the other hand, UHRF1 was initially presumed to synergize with DAC through DNA demethylation (Giovinazzo et al, 2019). Our results revealed that lower methylation, as seen by either UHRF1 deletion or prior DNMT1 inhibitor (GSK-3483862) treatment, blunted DAC-induced DNA damage and subsequent cell death (Figs. 3B–D and EV3E–H). These results suggest that DNA methylation levels may determine the accessibility of incorporated DAC to DNMT1 protein and point out a potential explanation for the inevitable relapse observed in DAC-treated patients (Sato et al, 2017). In this scenario, tumor cells with lowered global DNA methylation level during the long-term administration become less susceptible to this drug, thus, unintentionally diminishing DAC efficacy. Understanding this interplay can guide the development of combination therapies to enhance DAC effectiveness and potentially overcome acquired resistance in clinical applications. It is worth noting that while retroelements were significantly activated upon DAC treatment (Fig. EV5B), which is consistent with the effect of DNA demethylation induced by DAC (Chiappinelli et al, 2015; Goyal et al, 2023; Mehdipour et al, 2020; Roulois et al, 2015), we did not observe a robust activation of type I interferon response in DAC-treated HAP1 cells from our RNA-seq results. This could be due to cell type specific properties or different DAC treatment regimens,

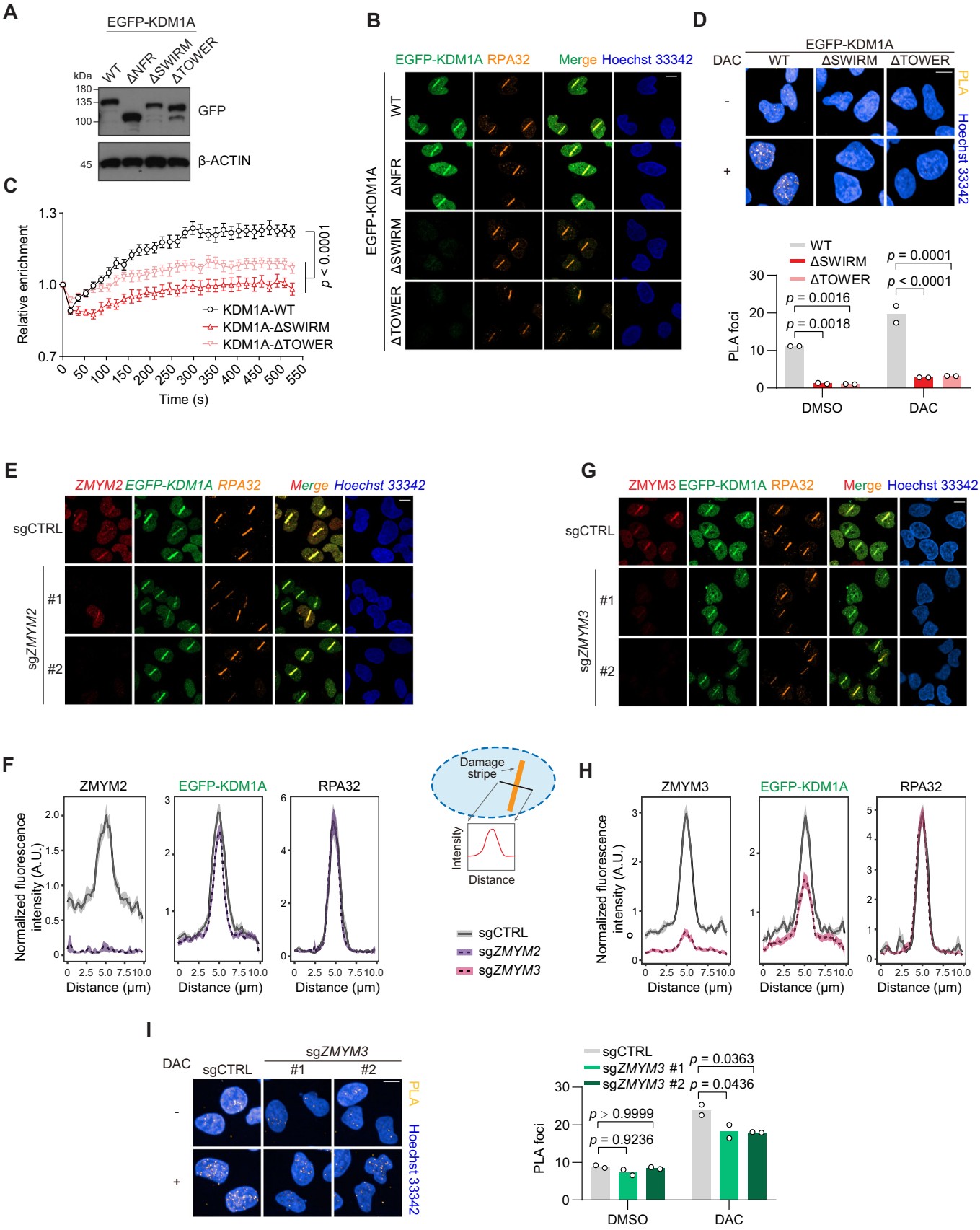

**Figure 7. Efficient recruitment of KDM1A to DNA damage sites relies on both SWIRM and TOWER domains and ZMYM3.**

(A) Immunoblot analysis showing U2OS cells stably expressing EGFP-tagged KDM1A-WT, KDM1A-ΔNFR, KDM1A-ΔSWIRM, or KDM1A-ΔTOWER, with β-ACTIN as a loading control. (B) Representative images of immunostaining for EGFP-KDM1A and the DSB marker RPA32 in U2OS cells expressing either EGFP-tagged KDM1A-WT, KDM1A-ΔNFR, KDM1A-ΔSWIRM, or KDM1A-ΔTOWER after laser-induced DNA damage. The scale bar is 10 μm. (C) Quantification of recruitment kinetics to laser-induced damage sites for EGFP-tagged KDM1A-WT, KDM1A-ΔSWIRM, and KDM1A-ΔTOWER in U2OS cells. Data represent the means ± SEM ($n = 15$ cells per cell type). The $p$ values were determined using one-way ANOVA followed by Dunnett's multiple comparisons test. (D) Representative images (upper) of PLA results between EGFP and γH2A.X in U2OS cells expressing either EGFP-tagged KDM1A-WT, KDM1A-ΔSWIRM, or KDM1A-ΔTOWER 24 h after release from single-round thymidine synchronization, with or without a 30-min DAC (10 μM) pulse. Quantification of PLA foci number (lower) in these cells. Each point represents the mean of one experiment ($N = 2$ experiments; $n > 6000$ cells for each condition). The $p$ values were determined using one-way ANOVA followed by Sidak's multiple comparisons test. The scale bar is 10 μm. (E) Representative images of immunostaining for ZMYM2, EGFP-KDM1A, and RPA32 in EGFP-KDM1A-expressing U2OS cells transduced with either sgCTRL or two different sgRNAs targeting *ZMYM2* (sg*ZMYM2* #1 and #2) after laser-induced DNA damage. The scale bar is 10 μm. (F) Quantification for profiles of ZMYM2, EGFP-tagged KDM1A and RPA32 immunostaining signals along the lines across damage stripes in (E). The relative signal intensities were normalized to the mean of the control U2OS cells. Data represent mean ± SEM ($n = 20$ cells for sgCTRL; $n = 30$ cells for sg*ZMYM2*, two sgRNAs combined). (G) Representative images of immunostaining for ZMYM3, EGFP-KDM1A, and RPA32 in EGFP-KDM1A-expressing U2OS cells transduced with either sgCTRL or two different sgRNAs targeting *ZMYM3* (sg*ZMYM3* #1 and #2) after laser-induced DNA damage. The scale bar is 10 μm. (H) Quantification for profiles of ZMYM3, EGFP-tagged KDM1A and RPA32 immunostaining signals along the lines across damage stripes in (G). The relative signal intensities were normalized to the mean of the control U2OS cells. Data represent mean ± SEM ($n = 20$ cells for sgCTRL; $n = 30$ cells for sg*ZMYM3*, two sgRNAs combined). (I) Representative images (left) of PLA results between EGFP and γH2A.X in EGFP-KDM1A-expressing U2OS cells transduced with either sgCTRL or two different sgRNAs targeting *ZMYM3* (sg*ZMYM3* #1 and #2) 24 h after release from single-round thymidine synchronization, with or without a 30-min DAC (10 μM) pulse. Quantification of PLA foci number (right) in these cells. Each point represents the mean of one experiment ($N = 2$ experiments; $n > 5000$ cells for each condition). The $p$ values were determined using one-way ANOVA followed by Sidak's multiple comparisons test. The scale bar is 10 μm. Source data are available online for this figure.

as has been also suggested by others (Goyal et al, 2023). Moreover, transcriptome analysis on the clinical samples collected from MDS and chronic myelomonocytic leukemia (CMML) patients uncovered that the reactivation of retroelements is unable to discriminate responders and non-responders (Kazachenka et al, 2019). Together, our results indicate a more common role of DNA damage that is induced by DAC acting as a key driver of DAC cytotoxicity in various tumors.

Intriguingly, our screen results showed that depletion of proteasome factors was associated with increased viability upon DAC treatment (Figs. 1E,F and EV1D,F), which are involved in DNA-protein crosslink repair (Stingele et al, 2017). Proteasome-mediated degradation of trapped DNA topoisomerase II (TOP2) cleavage complexes (TOP2ccs) could promote error-prone DNA repair processes, resulting in the accumulation of deleterious mutations (Sciascia et al, 2020). In a similar manner, DNA lesions will be exposed by degrading DNMT-DNA adducts. Therefore, inhibiting proteasome activity would suppress error-prone DNA repair of DNA adducts and preserve genome integrity (Sciascia et al, 2020). However, it is unknown whether DNMT could dissociate from DAC without leaving any collateral damages on DNA strand like previously observed for TOP2 (Sciascia et al, 2020).

Besides the DNMT-DNA adducts and subsequent damages, DCTD-mediated misincorporation of 5-aza-dUTP into DNA emerged as another key source of DNA damages induced by DAC (Figs. 1H and EV2A–E). This hypothesis is in corroboration with a most recent study published by Steven Jackson's group (Carnie et al, 2024b). Analogue DNA damage has also been documented with dUTP and its derivatives like hmdUTP (Fugger et al, 2021; Saxena et al, 2024; Weiner et al, 1995). The most frequent mutation of *DCTD* in cancer cell lines is homo-deletion (Fig. EV2F). To reverse this unfavorable situation, we successfully increased DAC-mediated cell death in *DCTD* KO tumor cell lines by providing excessive thymidine in the medium (Figs. 1I and EV2H,I). Success in this case convinced us that it is feasible to enhance DAC efficacy in patients harboring *DCTD* homo-deletion through manipulating nucleotide abundance.

The striking similarity between the action of PARP inhibitors and DAC is intriguing: both induce extensive DNA damage that is dependent on successive rounds of DNA replication and becomes amplified in HR-deficient cells (Simoneau et al, 2021). While their effects are similar, the underlying mechanisms are distinct. PARP inhibitors induce persistent single-strand DNA gaps, which hinder the DNA replication process (Simoneau et al, 2021). As for DAC, we hypothesize that cells cannot effectively resolve remnants of DNMT-DNA adducts before entering the next S phase. In this scenario, replisomes would be obstructed by these remnants during DNA replication. This hypothesis is supported by the observed reduction in EdU incorporation in DAC-pulsed cells during the second round of DNA replication (Fig. 4C). In addition, a previous study also found that DAC treatment induces phosphorylation of CHK1, a marker of replication stress (Palii et al, 2008). In line with our findings, recent work on uracil cytotoxicity uncovered that the base alteration itself, rather than intermediates from base excision repair or SSB repair, directly induces DNA replication stress in the subsequent S phase (Saxena et al, 2024). This suggests that 5-aza-dU, a derivative of DAC, is a potential source of DNA replication stress as well. In summary, both DNMT-DNA adducts and 5-aza-dU generated from DAC treatment induce DNA damage through interfering with the second round of DNA replication.

Besides of well-known genes in DNA repair pathways, our study also identified novel players previously unknown. One of such is CTDNEP1, which forms a protein complex with the regulatory subunit CNEP1R1. While this complex is crucial for CTDNEP1 function, the exact mechanisms remain a mystery (Han et al, 2012; Kim et al, 2007). Mutations in CTDNEP1 are significantly enriched in MYC-driven medulloblastomas with poor prognosis (Luo et al, 2023), suggesting its potential role in cancer development. Loss of CTDNEP1 resulted in increased genome instability and activation of DNA damage response (Luo et al, 2023). Whether its phosphatase activity directly participate in DNA repair remains an open question, which demands further exploration. By uncovering novel players like CTDNEP1, our study sheds light on the multifaceted nature of DAC's action and its potential for treating cancers with diverse DNA repair deficiencies.

Unexpectedly, our study uncovered a demethylase activity-independent role of KDM1A in protecting cells from DAC-induced DNA damage (Fig. 5). We found that KDM1A deletion does not influence the cell cycle progression and DNMT-DNA adducts formation or degradation. However, in support of a role for KDM1A in mitigating DAC-induced DNA damage, we observed a delayed clearance of γH2A.X signals in KDM1A KO cells compared to WT controls when using Zeocin as an alternative DNA damaging agent. Based on these cumulative findings, we propose that KDM1A primarily participates in DNA repair processes during subsequent S phases to ameliorate the effects of DAC-induced DNA damage.

Both SWIRM and TOWER domains are indispensable for KDM1A's pro-survival function upon DAC treatment and its localization at DNA damage sites (Figs. 5–7). The requirement for TOWER domain indicates the importance of the KDM1A/CoREST/HDAC complex. This is further supported by the evidence that CoREST deletion sensitizes cells to DAC (Fig. EV4H–K). Given that zinc fingers of ZMYM3 mediating its localization at DNA damage sites is also required for interaction with KDM1A (Leung et al, 2017), we speculate that KDM1A, CoREST, and ZMYM3 are recruited as an integral protein complex. However, their impacts on repair pathways go opposites: ZMYM3 promotes HR repair (Leung et al, 2017) while KDM1A suppresses it (Mosammaparast et al, 2013) (Fig. 6E), suggesting an intricate interplay within this complex to maintain a balance between the choices of different repair pathways. While the exact molecular mechanisms remain under investigation, this study proposes several possibilities. KDM1A may regulate HR through its interaction with MRE11 (Porro et al, 2014), a nuclease responsible for DNA end resection. It was reported that MRE11 could directly bind to acetylated histone (Kim et al, 2020a), which could be targeted by HDACs that are co-recruited with KDM1A/CoREST/ZMYM3 complex. In addition, HDACs recruited to DNA damage sites might suppress local transcription, which has been reported to facilitate HR by forming special structures called DR-loops, encompassing both DNA-DNA and DNA-RNA hybrids (Ouyang et al, 2021). The exact pathway by which the altered choice of DNA repair pathways in KDM1A-deficient cells contributes to cellular vulnerability following extensive DNA damage requires further investigation. Nevertheless, our data suggest a scaffold function of KDM1A at DNA damage sites and developing PROTAC targeting this function could offer a novel therapeutic approach.

# Methods

### Reagents and tools table

| Reagent/Resource | Reference or Source | Identifier or Catalog Number |
| --- | --- | --- |
| **Experimental models** | | |
| HAP1 | Horizon | C631 |
| HEK293FT | Procell | CL-0313 |
| JeKo-1 | Procell | CL-0128 |
| Jurkat E6 | Procell | CL-0129 |

| Reagent/Resource | Reference or Source | Identifier or Catalog Number |
| --- | --- | --- |
| U2OS | Cell Resource Center, Peking Union Medical College | 1101HUM-PUMC000028 |
| T47D | Procell | CL-0228 |
| BT549 | Procell | CL-0041 |
| HCC1937 | Procell | CL-0093 |
| SKBR3 | Procell | CL-0211 |
| ZR75-1 | Procell | CL-0247 |
| BT474 | Procell | CL-0040 |
| MDA-MB-436 | Procell | CL-0383 |
| **Recombinant DNA** | | |
| lentiCRISPRv2 | Addgene | #52961 |
| LentiGuide | Addgene | #52963 |
| pLenti-EF1A-PGK-puro | homemade | NA |
| AAVS1-Neo-CAG-Flpe-ERT2 | Addgene | #68460 |
| **Antibodies** | | |
| DCTD | Santa Cruz | sc-376659 |
| DNMT1 | Santa Cruz | sc-20701 |
| DNMT1 | ABclonal | A16729 |
| UHRF1 | Santa Cruz | sc-373750 |
| KDM1A | ABcam | ab17721 |
| γH2A.X | ABcam | ab22551 |
| γH2A.X | ABcam | ab2893 |
| RPA32 | Santa Cruz | sc-56770 |
| ZMYM2 | Thermo Scientific | PA5-28265 |
| ZMYM3 | Thermo Scientific | PA5-51862 |
| RCOR1 | Sigma | 07-455 |
| RCOR2 | ABcam | ab37113 |
| HDAC1 | Proteintech | 6085-1-Ig |
| ACTB | ABclonal | AC026 |
| GFP | ABclonal | AE011 |
| GFP | ABcam | ab13970 |
| GFP | MBL | 598 |
| H3 | ABclonal | A2348 |
| FLAG | Sigma | B3111 |
| PARylation | Santa Cruz | sc-56198 |
| XRCC1 | ABclonal | A0442 |
| Alexa Fluor Plus 555-conjugated goat anti-mouse IgG (H + L) | Thermo Scientific | A32727 |
| Alexa Fluor 647-conjugated donkey anti-rabbit IgG (H + L) | Thermo Scientific | A31573 |
| Alexa Fluor 488-conjugated goat anti-mouse IgG (H + L) | Thermo Scientific | A32723 |

| Reagent/Resource | Reference or Source | Identifier or Catalog Number |
|---|---|---|
| Alexa Fluor 488-conjugated goat anti-chicken IgY (H + L) | ABcam | ab150169 |
| HRP-conjugated Donkey Anti-Mouse IgG(H + L) | Proteintech | SA00001-8 |
| HRP-conjugated Donkey Anti-Rabbit IgG(H + L) | Proteintech | SA00001-9 |
| **Oligonucleotides and other sequence-based reagents** | | |
| sgRNA sequences | This study | Dataset EV3 |
| **Chemicals, Enzymes and other reagents** | | |
| Decitabine | Selleck | S1200 |
| ORY-1001 | Selleck | S7795 |
| GSK2879552 | Selleck | S7796 |
| AZD-7648 | Selleck | S8843 |
| GSK3484862 | TargetMol | T11469 |
| Thymidine | Sigma | T1895 |
| FKBP12 PROTAC dTAG-13 | MCE | HY-114421 |
| Zeocin | Thermo Scientific | R25001 |
| Puromycin | InvivoGen | ant-pr |
| Blasticidin | InvivoGen | ant-bl |
| Hygromycin | VWR | 97064-454 |
| G418 | Inalco Pharmaceuticals | 1758-1811 |
| Hoechst 33342 | Lablead | B2261 |
| BsmBI | NEB | R0580 |
| TRIzol | Thermo Scientific | 15596026 |
| **Software** | | |
| Harmony 4.9 | PerkinElmer | NA |
| Fiji (v.2.15.0) | https://imagej.net/software/fiji/ | NA |
| MAGeCK | Li et al, 2014b | NA |
| MAGeCK-VISPR | Li et al, 2015 | NA |
| clusterProfiler | Wu et al, 2021 | NA |
| FastQC (v0.12.1) | https://github.com/s-andrews/FastQC | NA |
| trim_galore | https://github.com/FelixKrueger/TrimGalore | NA |
| Bowtie2 | https://github.com/BenLangmead/bowtie2 | NA |
| Picard | https://broadinstitute.github.io/picard/ | NA |
| MACS2 | https://github.com/macs3-project/MACS | NA |
| deeptools | https://github.com/deeptools/deepTools | NA |
| Salmon (v1.4.0) | https://combine-lab.github.io/salmon/ | NA |
| DESeq2 | https://bioconductor.org/packages/release/bioc/html/DESeq2.html | NA |

| Reagent/Resource | Reference or Source | Identifier or Catalog Number |
|---|---|---|
| SalmonTE | https://github.com/hyunhwan-jeong/SalmonTE | NA |
| **Other** | | |
| Seamless Cloning | Beyotime | D7010M |
| CellTiter-Glo assay | Promega | G7573 |
| EdU Cell Proliferation assay | Baseclick | BCK-EdU555 |
| BeyoClick EdU Cell Proliferation Kit | Beyotime | C0081S |
| Duolink In Situ Detection Reagents Red Kit | Sigma | DUO92008 |
| KAPA Hyper Prep Kit | Roche | KK8504 |
| Opera Phoenix | PerkinElmer | NA |
| LSM980 | Zeiss | NA |

## Plasmids

sgRNA sequences targeting individual genes were cloned into lentiCRISPRv2 (Addgene, #52961) or LentiGuide (Addgene, #52963) plasmids carrying different selection markers using BsmBI (NEB). The sgRNA sequences are listed in Dataset EV3. Coding sequences for human *CTDNEP1*, *DNMT1*, *KDM1A*, and RCOR1 were cloned from cDNA of HAP1 cells. 3×FLAG-tagged DNMT1 and KDM1A, and EGFP-tagged KDM1A and RCOR1 were cloned into pLenti-EF1A-PGK-puro vector by Seamless Cloning (Beyotime) according to the manufacturer's instructions. Point mutations or deletions were introduced into DNMT1, KDM1A, and RCOR1 to generate catalytic-dead DNMT1-C1226A and KDM1A-AE/KA and 3DA, and indicated truncations of KDM1A and RCOR1. Cas9, mClover3-tagged CTDNEP1, EJ7-GFP reporter, and DR-GFP reporter were clone into AAVS1-Neo-CAG-Flpe-ERT2 (Addgene, #68460) to replace the Flpe-ERT2 sequence for *AAVS1* locus-targeted knock-in. To tag endogenous KDM1A with degron FKBP at the N terminal, corresponding homologous arms were cloned from genomic DNA isolated from HAP1 cells and an in-frame PuroR-P2A-FKBP encoding sequence was inserted immediately after the start codon ATG by Seamless Cloning.

## Cell lines

### Cell culture

The human haploid chronic myeloid leukemia cell line HAP1 (Horizon) was maintained in Iscove's modified Dulbecco's medium (IMDM) supplemented with 10% fetal bovine serum (FBS) and 1% penicillin–streptomycin (P/S). All HAP1 cell lines used in this study were cultured as diploids. HEK293FT were maintained in Dulbecco's Modified Eagle's Medium (DMEM) supplemented with 10% FBS and 1% P/S. JeKo-1 (Procell) were maintained in RPMI-1640 medium supplemented with 20% FBS and 1% P/S. Jurkat E6 (Procell) were maintained in RPMI-1640 medium supplemented with 10% FBS and 1% P/S. U2OS (the Cell Resource Center, Peking Union Medical College) were maintained in McCoy's 5A medium

supplemented with 10% FBS and 1% P/S. Breast cancer cell lines and corresponding mediums were purchased from Procell. As the manufacturer's instructions, T47D, BT549, and HCC1937 cells were maintained in RPMI-1640 medium supplemented with 10% FBS, 10 μg/mL insulin, and 1% P/S. SKBR3 cells were cultured in McCoy's 5A medium supplemented with 10% FBS and 1% P/S. ZR75-1 cells were maintained in RPMI-1640 medium supplemented with 10% FBS and 1% P/S. BT474 cells were maintained in RPMI-1640 medium supplemented with 20% FBS, 2 mM L-glutamine, 10 μg/mL insulin, and 1% P/S. MDA-MB-436 cells were maintained in DMEM supplemented with 15% FBS, 10 μg/mL insulin, 16 μg/mL glutathione, and 1% P/S. All cell lines were grown at 37 °C and 5% $CO_2$.

### Cell lines generation

For generation of pooled knockout cell lines, cells were infected with lentiCRISPRv2 lentivirus targeting genes of interest and selected by corresponding antibiotics for subsequent experiments.

Isogenic HAP1 cell lines with gene depletion were generated using CRISPR/Cas9 system. Briefly, HAP1 cells were transfected with a pair of lentiCRISPRv2 plasmids carrying sgRNA targeting the same genes, including *DCTD* (sgRNAs #3 and #4), *KDM1A* (sgRNAs #1 and #2), *ZMYM2* (sgRNAs #1 and #3), *ZMYM3* (sgRNAs #1 and #3), *RCOR1* (sgRNAs #1 and #2), and *RCOR2* (sgRNAs #1 and #2), by Lipofectamine 3000 under instructions of the manufacturer. Twenty-four hours post transfection, cells were selected by 1 μg/ml puromycin for 1 day, and then seeded as single cells into 96-well plates for clonal selection. Clones were expanded and verified by genotyping and immunoblot. To generate *AAVS1* locus integrated with Cas9, mClover3-CTDNEP1, EJ7-GFP reporter, or DR-GFP reporter, HAP1 cells were transfected with a lentiCRISPRv2 plasmid containing sgRNA targeting *AAVS1* and a repair template plasmid AAVS1-NeoR inserted with indicated sequences. Cells were selected by 1 mg/ml G418. FKBP-KDM1A clones expressing the degron FKBP tagged at the N-terminal of endogenous KDM1A were generated by a similar strategy. Knock-in clones were verified as described above.

To generate KDM1A complemented cells lines, KDM1A KO cells were infected with lentivirus expressing empty or 3×FLAG-tagged KDM1A derivatives for stable expression. Transduced cells were selected by 1 μg/ml puromycin until stable. Similarly, HeLa cells stably expressing 3×FLAG-tagged DNMT1 proteins and U2OS cells stably expressing EGFP-tagged KDM1A or RCOR1 were constructed.

### Virus production and infection

HEK293FT cells were plated into dishes coated with poly-D-lysine the day before transfection. pMDLg/pRRE (Addgene, #12251), pCMV-V-SVG (Addgene, # 8454), pRSV-Rev (Addgene, #12253), and lentiviral plasmids were transfected into 293FT cells using Vigofect (Vigorous) following the manufacturer's instructions. Virus-containing medium were collected twice at 48 h and 72 h post-transfection, filtered through a 0.45 μm filter, and used for transduction. For screen libraries, lentivirus was concentrated by ultracentrifugation at 25,000 rpm for 2 h, resuspended, aliquoted, and stored at −80 °C before use. Forty-eight hours after infection, cells were selected by corresponding antibiotics.

### Genome-wide CRISPR KO screen

Thirty million HAP1-Cas9-F5 cells were infected with the human CRISPR Brunello lentiviral pooled library at a low multiplicity of infection (MOI = 0.3) as previously described (Joung et al, 2017). After 7 days of puromycin selection, the surviving cells were expanded to achieve a maximal knockout efficiency. Ten million infected cells, for a 100-fold representation of the sgRNA library, were treated with DMSO or decitabine (DAC) at 500 nM for 10 days. The screen was performed in two replicates. Cells were collected at Day 0, 2, 4, 6, 8, and 10 post-DAC treatment for genomic DNA purification by phenol-chloroform extraction and ethanol precipitation. Appropriate genomic DNA were used for a two-step PCR to construct sequencing libraries. Briefly, the first step amplified the gRNA sequences and the second step added barcodes for deep sequencing.

### CRISPR-Cas9 mediated KDM1A domain scanning

Similar to above genome-wide knockout screen, 1 million HAP1-Cas9-F5 cells were infected with the custom lentiviral pooled library for KDM1A domain scanning, including 58 sgRNA covering *KDM1A* coding region, 2 sgRNAs targeting *DCK*, and 10 non-targeting sgRNAs. One million infected cells were treated with DMSO or DAC at 500 nM for 6 days. The screen was performed in two replicates. Cells were collected at Day 0, 2, 4, and 6 post-DAC treatment. Genomic DNA were extracted for library construction and deep sequencing.

### Cell viability assay

Diluted cells were seeded into 96-well plates at the densities of 5000 cells per well for HAP1 cell, 2000 cells per well for breast cancer cell lines, and 20,000 cells per well for Jurkat and JeKo-1 cells. Twelve hours after seeding, cells were treated with DMSO or serially diluted concentrations of chemicals as indicated for 3–8 days. The medium was replaced with fresh medium with or without chemicals every day for HAP1 cell and every two days for breast cancer cell lines. Cell viability was measured by CellTiter-Glo assay (Promega) as the manufacturer's instructions.

### Immunofluorescence staining

Cells were plated at 50,000 cells per well into 24 well glass-bottom plates (Cellvis) and treated with DMSO or DAC at indicated concentrations. After 24 h, cells were washed by PBS for 3 times, fixed in 4% paraformaldehyde for 15 min at room temperature, and washed by PBS for another 3 times. Cells were permeabilized by 0.5% Triton X-100 in PBS for 15 min and incubated with IF Block buffer (10% FBS in PBS with 0.1% Triton X-100 (PBST)) for 1 h at room temperature. Samples were then incubated with primary antibodies targeting γH2A.X in IF Block buffer for 1.5 h at room temperature. After washed with PBST for 3 times, primary antibodies were detected with Alexa Fluor-conjugated secondary antibodies for 1.5 h at room temperature. Hoechst 33342 staining was used to detect nuclei. To track the kinetics of γH2A.X, DNMT1, PARylation, and XRCC1 upon DAC treatment, U2OS or HAP1 cells were synchronized by a single round of thymidine block (2 mM) for 18–20 h, and treated with or without a 30-min DAC

pulse (10 μM) upon releasing into S phase, and collected at indicated time points. To detect chromatin-bound proteins, cells were pre-extracted with CSK buffer (10 mM PIPES, pH 6.8, 100 mM NaCl, 300 mM sucrose, 3 mM MgCl$_2$, and 1 mM EGTA) supplemented with 0.5% Triton X-100 for 5 min on ice, and incubated with Stripping buffer (10 mM Tris-HCl, pH 7.4, 10 mM NaCl, 3 mM MgCl$_2$, 1% Tween 20, and 0.5% sodium deoxycholate) for 5 min on ice followed by PBS wash before fixation. EdU incorporation was detected using the EdU Cell Proliferation assay Kit (Baseclick) according to the manufacturer's protocol. Imaging was detected by high content screening system (Opera Phoenix, PerkinElmer), using a 40× lens. Relative fluorescence intensity analysis was performed using the Harmony 4.9 software.

## Laser micro-irradiation

Cells were seeded on a 35 mm μ-Dish with a glass bottom (IBIDI). Prior to micro-irradiation, cells were sensitized with 10 μg/mL Hoechst 33342 for 20 min. Laser-induced micro-irradiation was carried out using a Zeiss LSM980 confocal microscope. Cells were irradiated with a fixed wavelength 405-nm laser at 60% power. Live cell images were captured in certain intervals after irradiation. The fluorescence intensity of EGFP-tagged protein in the irradiated region was measured and normalized with the surrounding area in the same cell. Quantification analyses were performed with Fiji software (v.2.15.0). For immunofluorescence staining, cells were washed by PBS for 3 times after irradiation, extracted with CSK buffer supplemented with 0.5% Triton X-100 for 5 min on ice, and incubated with Stripping buffer for 5 min on ice. Then cells were washed again with PBS and fixed in 4% paraformaldehyde for 15 min at room temperature followed with conventional immunofluorescence staining steps to detect indicated targets.

## Proximity ligation assay (PLA)

U2OS cells expressing EGFP-tagged KDM1A or ZMYM3 were released from thymidine block and treated with a DAC pulse. Twenty-four hours later, cells were pre-extracted and fixed. PLA were then performed with γH2A.X and GFP antibodies. The Duolink In Situ PLA Anti-Mouse Minus and Anti-Rabbit Plus probes (Sigma) and Duolink In Situ Detection Reagents Red Kit (Sigma) were then used according to the manufacturer's instructions. Hoechst 33342 staining was used to detect nuclei. Imaging was detected by high content screening system (Opera Phoenix, PerkinElmer), using a 60× lens. Quantification of PLA foci number was performed using the Harmony 4.9 software.

## Immunoblot

Cells were collected by trypsinization, washed by PBS once, and resuspended in PBS. Cell suspension was lysed with equal volume of 2× Lysis buffer (40 mM Tris-HCl, pH 8.0, 2% SDS, and 10 mM EDTA) and boiled at 100 °C for 30 min. Protein concentration was quantified by Bradford Protein Assay Kit (Beyotime). Equal amounts of lysates were run on SDS-PAGE and then transferred to PVDF membrane. Membranes were blocked in 5% skimmed milk in TBST (TBS buffer (Tris-HCl, pH 7.4, 137 mM NaCl, and 27 μM KCl) supplemented with 0.15% Tween-20) for 1 h at room temperature and incubated with primary antibodies for 1 h at room temperature. After washed by TBST for three times, membranes were incubated with secondary antibodies conjugated with HRP (horse radish peroxidase). Membranes were washed by TBST three times before developing.

## Immunoprecipitation

KDM1A KO cells complemented with 3×FLAG-labeled KDM1A WT or ΔSWIRM were collected by trypsinization and washed by PBS once. The nuclei were isolated through incubating the cells in Buffer A (20 mM Tris-HCl, pH 8.0, 1.5 mM MgCl$_2$, 10 mM KCl, 0.5 mM DTT, and 10% glycerol) on ice for 10 min. After centrifugation, the nuclei were resuspended in Buffer C (20 mM Tris-HCl, pH 8.0, 1.5 mM MgCl$_2$, 420 mM NaCl, 0.5 mM DTT, 0.2 mM EDTA, and 10% glycerol) and incubated on ice for 30 min. After centrifugation, the supernatant containing the soluble nuclear extract (NE) was quantified and diluted to obtain a final concentration of 150 mM for NaCl by Dilution buffer (20 mM Tris-HCl, pH 8.0, 1.5 mM MgCl$_2$, 0.5 mM DTT, 0.2 mM EDTA, 10% glycerol, and 0.1% NP-40). The samples were incubated with M2 agarose beads (Sigma) overnight on a rotating wheel at 4 °C. The following day, beads were washed once with BC150 buffer (20 mM Tris-HCl, pH 8.0, 150 mM NaCl, 1 mM DTT, 1 mM EDTA, 10% glycerol, and 0.1% NP-40) and twice with BC250 buffer (20 mM Tris-HCl, pH 8.0, 250 mM NaCl, 1 mM DTT, 1 mM EDTA, 10% glycerol, and 0.1% NP-40). The bound material was eluted by competition using 0.5 mg/mL 3×FLAG peptide in BC100 (20 mM Tris-HCl, pH 8.0, 100 mM NaCl, 1 mM DTT, 1 mM EDTA, and 10% glycerol) or boiled with Buffer C and an equal volume of 2× SDS Loading buffer (100 mM Tris-HCl, pH 6.8, 4% SDS, 0.2% BPB, 20% glycerol, and 2% β-mercaptoethanol) for 10 min. Eluted samples were loaded on SDS-PAGE for subsequent analysis.

## Label-free quantitative mass spectrometry

The gel bands were manually excised and digested individually as below. The protein bands were cut into small plugs, washed twice with distilled water. After dehydration, dry, reduction, and alkylation, the gel plugs were washed with 50% acetonitrile in 25 mM ammonium bicarbonate twice. The gel plugs were then dried and digested with sequence-grade modified trypsin in 25 mM NH$_4$HCO$_3$ overnight at 37 °C. The enzymatic reaction was stopped by adding formic acid to a 1% final concentration. The solution was then transferred to a sample vial for LC-MS/MS analysis. NanoLC-MS/MS experiments were performed on a Orbitrap Exploris 480 (Thermo Scientific) equipped with an Easy n-LC 1200 HPLC system (Thermo Scientific) by faculties in the Laboratory of Proteomics Technology, IBP.

The raw data from Orbitrap Exploris 480 were analyzed with Proteome Discovery (v.2.4.1.15) using Sequest HT search engine against the Uniprot human protein database for protein identification. FDR analysis was performed with Percolator and FDR < 1% was set for protein identification. Proteins label-free quantification was also performed on Proteome Discovery using the areas of identified peptides. The outputs were manually filtered for the KDM1A interactome analysis. Interacted proteins should be identified with more than 1 unique peptide, found in WT triplicates. Contaminants were removed by an arbitrary abundance filter of >1,000,000 in WT samples. Statistical analysis was done in Perseus (v.2.0.11) as previously described (Tyanova et al, 2016).

Missing values were imputed and differential interactors were determined using multiple *t*-test using a permutation-based FDR approach with the significance threshold set to FDR = 0.01, S0 = 2 and visualized by a volcano plot.

## LC-MS/MS analysis of dC and 5mC

Genomic DNA was purified using Wizard Genomic DNA Purification Kit (Promega) as manufacturer's instructions. For the UHPLC-MS/MS analysis of dC and 5mC, 1 μg of genomic DNA was digested into nucleosides as previously described (Song et al, 2022). Briefly, the genomic DNA was mixed with Digestion buffer (10 mM Tris-HCl, pH 8.0, 1 mM $MgCl_2$, 1.0 U Super Nuclease (Sino Biological), 0.02 U snake venom phosphodiesterase I (Sangon), and 1.0 U calf intestine alkaline phosphatase (NEB)) for digestion at 37 °C for 4 h. Enzymes were removed by ultrafiltration tubes (Pall) following digestion. UHPLC-MS/MS analysis was performed on an Agilent 1290 Infinity LC system coupled with an Agilent 6410 triple quadrupole mass spectrometer (Santa Clara, CA, USA). A reversed-phase Agilent Zorbax Eclipse Plus C18 column was used for the UHPLC separation. Solvent A (an aqueous solution of 0.1% formic acid) and solvent B (methanol) were used as the mobile phase. The UHPLC separation was used with 5% B and 95% A as the isocratic separation. The stable isotopes were used as internal standards for calibrating the UHPLC-MS/MS quantitation of dC and 5mC, respectively.

## EJ7-GFP and DR-GFP report assay and FACS analysis

Cells integrated with either DR-GFP or EJ7-GFP reporter cassettes were first infected with lentiviruses expressing Cas9-sgRNA from the lentiCRISPRv2 vector to delete genes-of-interest. After blasticidin selection for 7 days, DR-GFP reporter cells were transduced with lentiviruses expressing I-Sce-I-mCherry and EJ7-GFP reporter cells were transduced with viruses expressing sgRNA-7a-BFP and sgRNA-7b-mCherry. For chemicals treatment, cells were incubated with indicated inhibitors while DSB induction. Five days later, cells were collected by trypsinization, resuspended, and analyzed for the GFP-positive rate through FACS.

## Cell cycle analysis

To label cells in S phase, cells were incubated with 20 μM EdU for 30 min before collection. Cells were then trypsinized, washed once with PBS, and fixed in ice-cold 75% ethanol overnight at 4 °C. EdU incorporation was detected using the BeyoClick EdU Cell Proliferation Kit with Alexa Fluor 647 (Beyotime C0081S) according to the manufacturer's protocol. After that, genomic DNA was stained with 0.1 mg/ml propidium iodide (Beyotime ST1569) and 0.1 mg/ml RNase A in PBS for 30 min at 37 °C before analysis by FACS on BD FACSCalibur instrument.

## ChIP-seq

ChIP experiments were performed as previous described as previous (Zhang et al, 2022). Briefly, HAP1 cells cultured in 10-cm dishes were collected by trypsinization, washed with PBS once, and resuspended in 1 mL of PBS. Then cells were cross-linked with 2 mM DSG for 30 min, followed by treatment with 1% formaldehyde for 10 min at room temperature. Crosslinking was terminated by adding 2 M glycine to a final concentration of 125 mM. Then the cells were collected by centrifugation and resuspended in lysis buffer 1 (50 mM HEPES, pH 7.9, 140 mM NaCl, 1 mM EDTA, 10% glycerol, 0.5% IGEPAL CA-630, and 0.25% Triton X-100) and incubated for 10 min on ice. After centrifugation, the cells were resuspended in lysis buffer 2 (10 mM Tris, pH 8.0, 200 mM NaCl, 1 mM EDTA, and 0.5 mM EGTA) and incubated for 10 min at room temperature. After centrifugation, the nuclei were resuspended in lysis buffer 3 (10 mM Tris, pH 8.0, 100 mM NaCl, 1 mM EDTA, 0.5 mM EGTA, 0.1% SDS, 0.1% sodium deoxycholate, 0.5% sodium N lauroylsarcosine, and 1 × protease inhibitor cocktail) and sonicated to 100–500 bp fragments by Covaris M220. Sonicated chromatin was incubated with antibody against KDM1A at 4 °C overnight and recovered with pre-washed Dynabeads Protein A for 2 h. Then the beads were sequentially washed with the following buffers: twice with low salt buffer (20 mM Tris, pH 8.0, 150 mM NaCl, 2 mM EDTA, and 1% Triton X-100), twice with high salt buffer (20 mM Tris, pH 8.0, 500 mM NaCl, 2 mM EDTA, and 1% Triton X-100), once with LiCl buffer (10 mM Tris, pH 8.0, 250 mM LiCl, 1 mM EDTA, 0.5% IGEPAL CA-630, and 0.5% sodium deoxycholate) and twice with TE buffer (10 mM Tris, pH 8.0 and 1 mM EDTA). Bound chromatin was eluted twice with 250 μL of freshly prepared elution buffer (1% SDS, 100 mM $NaHCO_3$, and 200 mM NaCl) for 15 min at room temperature with gentle rotation. Incubation at 65 °C overnight was performed to reverse crosslinking. After treatment with RNase A and proteinase K, DNA was purified by phenol-chloroform extraction and isopropanol precipitation. The ChIP-seq libraries were constructed with KAPA Hyper Prep Kit according to the manufacturer's instructions.

## RNA-seq

Total RNA was isolated by TRIzol (Thermo Scientific) following the manufacturer's instructions. Total RNA samples were sent to Annoroad for library construction and sequencing.

## Data analysis

### Genome-wide CRISPR KO screen
CRISPR-Cas9 KO screen data were analyzed using the MAGeCK (Li et al, 2014b) and MAGeCK-VISPR (Li et al, 2015) algorithms. MAGeCK calculated the read counts for each sgRNA and normalized read counts using predesigned non-targeting sgRNAs as control. The MAGeCK-VISPR (mageck mle) computes beta scores and the associated statistics for all genes in all conditions (DMSO and DAC treatments with various days). The beta scores were then normalized by cell cycle normalization method to make the scores comparable in different conditions. The differential beta score for each gene was calculated by subtracting the beta score of DMSO treatment from the corresponding beta score of DAC treatment. For each time point of screen, genes were ranked by differential beta score and the standard deviation (σ) of the differential beta scores for each sample were estimated. Genes with lower beta scores than mean-2σ and higher beta scores than mean+2σ were regarded as negatively and positively selected genes, respectively. Genes that were consistently positively or negatively identified in any two of day 6, day 8, or day 10 post-DAC treatments were defined as high-fidelity hits. GO enrichment analysis was performed using R package 'clusterProfiler' (Wu et al, 2021).

### KDM1A domain scanning

KDM1A domain scanning data were analyzed in a way like the Genome-wide CRISPR KO screen data, in which each domain of KDM1A was regarded as a "gene" targeted by several predesigned sgRNAs. The beta value of each domain was calculated in the same way as above. As to the analysis of the individual sgRNA, raw read counts of sgRNAs were first normalized by sequencing depth using MAGeCK and counts from replicate experiments were averaged. For data of each time point (day 2, 4, and 6), the ratios of counts from DAC-treated or DMSO-treated samples were calculated for each sgRNA. Then the ratios were normalized by the average of the 10 non-targeting control sgRNAs. The normalized ratios were used as the survival rate of cells to indicate the DAC sensitivity of each sgRNA-targeted region of KDM1A.

### ChIP-seq

Raw reads of ChIP samples and input samples were first assessed using FastQC (v0.12.1) to ensure sequencing quality. Then trim_galore (v0.6.10) was used to trim low-quality bases and sequencing adaptor contamination from raw read pairs. Trimmed read pairs were then aligned to human genome assembly (build hg38) using Bowtie2 software. Uniquely mapped reads were processed by Picard software to discard potential PCR duplicates. ChIP-seq peak calling was performed by MACS2 (v2.1.2) software based on the deduplicated reads ($q$-value < 1E−6). The deduplicated reads were used as input for deeptools (v3.5.4.post1) software to generate genome profile track files, which served as input for the subsequent meta profile and heatmap analysis.

### RNA-seq

RNA-seq reads were quantified using Salmon (v1.4.0) against transcripts annotation from human GENCODE release 36 (GRCh38). The raw counts data for all transcripts were imported into R DESeq2 package for differential expression analysis. Genes that were up- or down-regulated more than 2-fold with the s-value less than 0.05 were selected as DEGs (differentially expressed genes). Transcription levels of repetitive elements were performed using SalmonTE (v0.4). Repeat families with more than 3-fold changes were regarded as changed and used for further analysis.

Gene sets of interests were obtained from public databases: Interferon pathway from Reactome Pathway R-HSA-913531 (https://reactome.org/), DNA repair pathway from Reactome Pathway R-HSA-73894, nucleotide metabolism pathway from WikiPathways WP404 (https://www.wikipathways.org/), and pyrimidine metabolism pathway from KEGG database hsa00240 (https://www.genome.jp/kegg/pathway.html).

### Quantification and statistical analysis

Statistical tests used and sample sizes are described in figure legends.

## Data availability

The raw sequence data reported in this study have been deposited in the Genome Sequence Archive in National Genomics Data Center, China National Center for Bioinformation/Beijing Institute of Genomics, Chinese Academy of Sciences (GSA accession number: HRA006701) that are publicly accessible at https://bigd.big.ac.cn/gsa-human/browse/HRA006701.

The source data of this paper are collected in the following database record: biostudies:S-SCDT-10_1038-S44319-025-00385-w.

## Peer review information

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

## Acknowledgements

We thank Junying Jia and Shu Meng from the Core Facility of Institute of Biophysics, CAS, for flow cytometric analysis. We thank Jifeng Wang for assistance with protein mass spectrometry. We thank Ya Wang for help in using high content screening system. This work was supported by the National Natural Science Foundation of China (32288102 to BZ, 32170607 to JX, and 32000417 to WD) and the National Key Research and Development Program of China (2021YFC2300500). ZZ and JX are supported by the Youth Innovation Promotion Association (2017133 and 2020097, respectively) of the Chinese Academy of Sciences. BZ is supported by the New Cornerstone Investigator Program.

## Author contributions

**Pinqi Zhang**: Conceptualization; Data curation; Formal analysis; Investigation; Visualization; Writing—original draft; Writing—review and editing. **Zhuqiang Zhang**: Data curation; Formal analysis; Funding acquisition; Investigation; Visualization; Writing—original draft; Writing—review and editing. **Yiyi Wang**: Investigation. **Wenlong Du**: Funding acquisition; Investigation. **Xingrui Song**: Investigation. **Weiyi Lai**: Investigation. **Hailin Wang**: Resources; Project administration. **Bing Zhu**: Conceptualization; Resources; Supervision; Funding acquisition; Writing—original draft; Project administration; Writing—review and editing. **Jun Xiong**: Conceptualization; Resources; Data curation; Formal analysis; Supervision; Funding acquisition; Investigation; Visualization; Writing—original draft; Project administration; Writing—review and editing.

Source data underlying figure panels in this paper may have individual authorship assigned. Where available, figure panel/source data authorship is listed in the following database record: biostudies:S-SCDT-10_1038-S44319-025-00385-w.

## Disclosure and competing interests statement

The authors declare no competing interests.

# Expanded View Figures

**Figure EV1.   Genome-wide CRISPR-Cas9 screen results analysis to identify gene networks that regulating cellular response to DAC.**

(A) Immunoblot analysis showing that FLAG-tagged Cas9 protein was stably expressed in HAP1-Cas9-F5 cells, with β-ACTIN as a loading control. (B) Dose–response curves of HAP1-Cas9-F5 cells transduced with either control sgRNA (sgCTRL) or *DCK*-targeting sgRNA (sg*DCK*) upon DAC treatment at indicated concentrations. Cell viability was measured by CellTiter-Glo after 3 days of DAC treatment. Data are presented with ± SEM. Experiments performed in duplicates. The *p* value was determined using nonlinear regression followed by the extra sum-of-squares F test. (C) Ranked genes from CRISPR screen determined by comparing DAC to DMSO treatment. Genes are ranked by the averaged differential β scores at indicated time points. Negatively and positively selected genes are labeled with blue and red, respectively. (D) KEGG pathway enrichment of 6 clusters in Fig. 1D. The *p* values were determined using the hypergeometric distribution and adjusted by the Benjamini-Hochberg (BH) procedure. (E) STRING network of Cluster 1 hits. (F) STRING network of Cluster 6 hits.

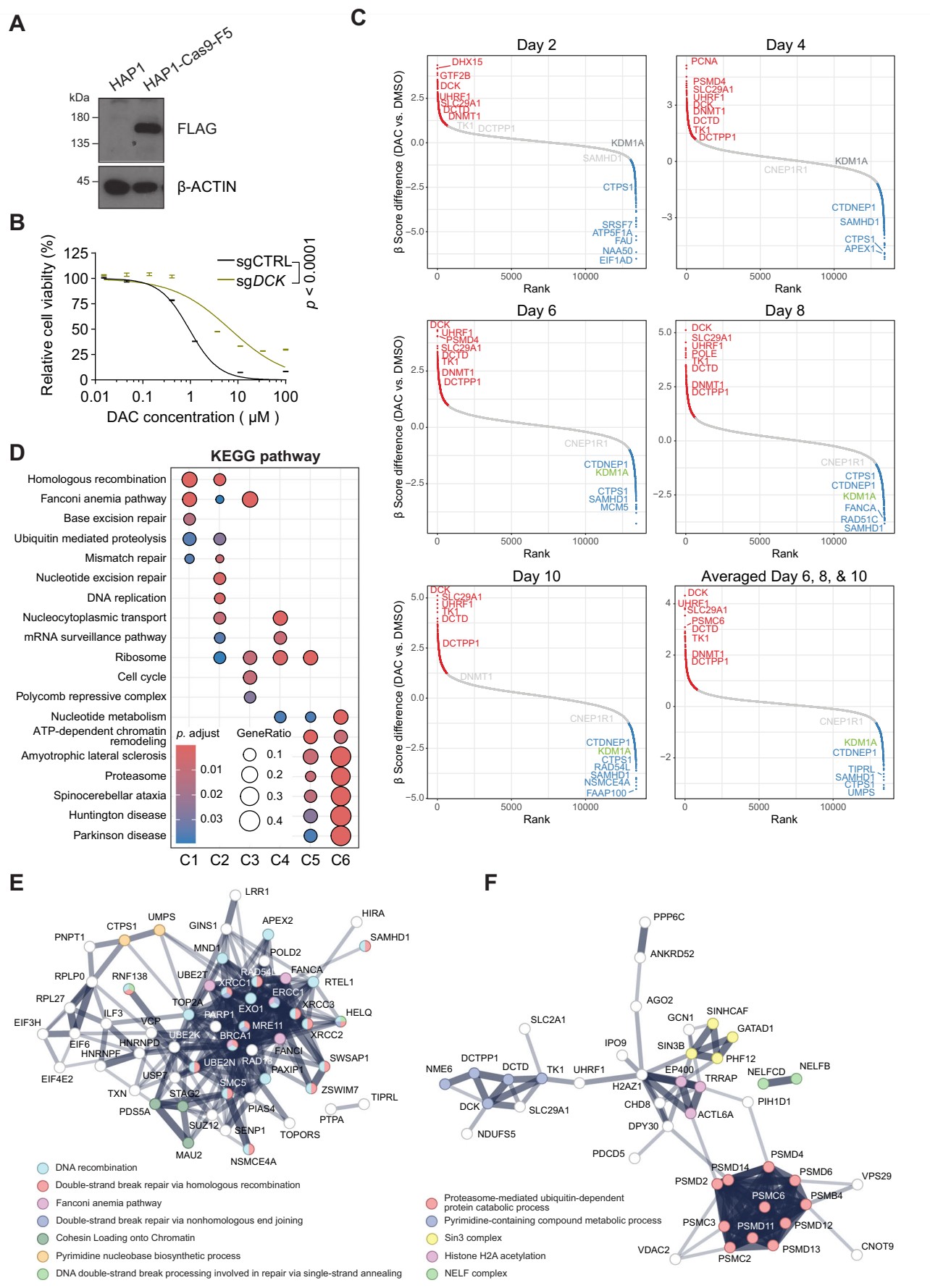

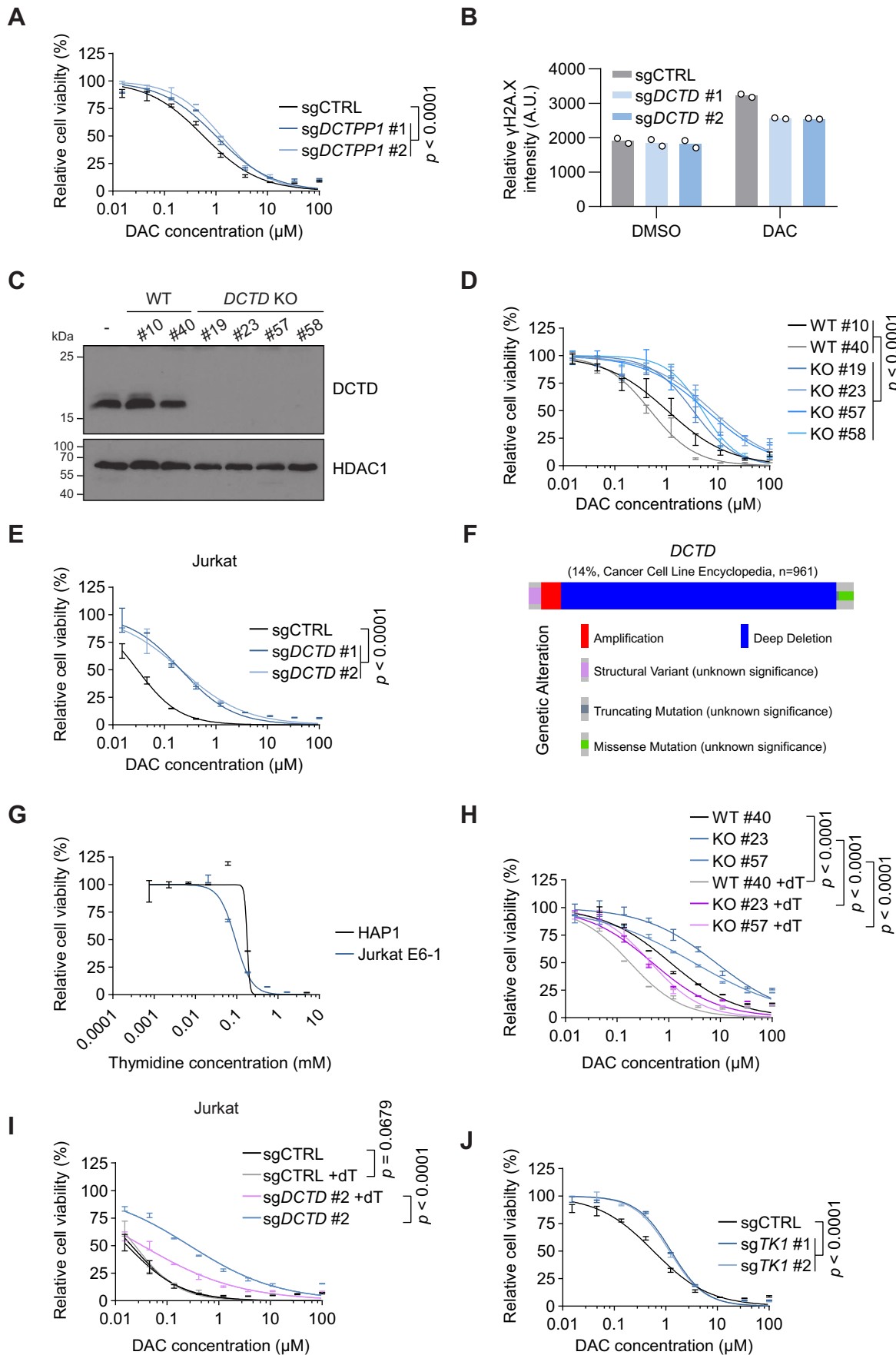

◀ **Figure EV2.  Modulating cellular nucleotide pool impacts sensitivity to DAC.**

(A) Dose–response curves of HAP1 cells transduced with sgCTRL or two different sgRNAs targeting *DCTPP1* (sg*DCTPP1* #1 and #2) upon DAC treatment at indicated concentrations. Cell viability was measured by CellTiter-Glo after 3 days of DAC treatment. Data are presented with ± SEM. Experiments performed in duplicates. The *p* values were determined using nonlinear regression followed by the extra sum-of-squares F test. (B) Quantification of γH2A.X immunostaining signal intensity in control and *DCTD*-depleted HAP1 cells after DAC treatment treated with DMSO or DAC (500 nM) for 24 h. Each point represents the mean of one experiment ($N = 2$ experiments; $n > 3700$ cells for each genotype and treatment). A.U., arbitrary unit. (C) Immunoblot analysis showing the DCTD protein level in WT and *DCTD* KO clones with HDAC1 as a loading control. (D) Dose–response curves of WT and *DCTD* KO HAP1 clones upon DAC treatment at indicated concentrations. Cell viability was measured by CellTiter-Glo after 3 days of DAC treatment. Data are presented with ± SEM ($n = 6$ replicates). The *p* values were determined using nonlinear regression followed by the extra sum-of-squares F test. (E) Dose–response curves of Jurkat cells transduced with sgCTRL or two different sgRNAs targeting *DCTD* (sg*DCTD* #1 and #2) upon DAC treatment at indicated concentrations. Cell viability was measured by CellTiter-Glo after 3 days of DAC treatment. Data are presented with ± SEM. Experiments performed in duplicates. The *p* values were determined using nonlinear regression followed by the extra sum-of-squares F test. (F) The genetic alteration profile of *DCTD* in cancer cell lines (data was from cBioPortal). (G) Dose–response curves of HAP1 and Jurkat cells treated with thymidine at indicated concentrations for 3 days. Cell viability was measured by CellTiter-Glo. Data are presented with ± SEM. Experiments performed in duplicates. (H) Dose–response curves of WT and *DCTD* KO HAP1 clones upon DAC treatment at indicated concentrations with or without (w/o) 50 µM thymidine. Cell viability was measured by CellTiter-Glo after 3 days of DAC treatment. Data are presented with ± SEM. Experiments performed in duplicates. The *p* values were determined using nonlinear regression followed by the extra sum-of-squares F test. (I) Dose–response curves of Jurkat cells transduced with control or DCTD-targeting sgRNAs upon DAC treatment at indicated concentrations w/o 10 µM thymidine for 3 days. Cell viability was measured by CellTiter-Glo. Data are presented with ± SEM. Experiments performed in duplicates. The *p* values were determined using nonlinear regression followed by the extra sum-of-squares F test. (J) Dose–response curves of HAP1 cells transduced with sgCTRL or two different sgRNAs targeting *TK1* (sg*TK1* #1 and #2) upon DAC treatment at indicated concentrations. Cell viability was measured by CellTiter-Glo after 3 days of DAC treatment. Data are presented with ± SEM. Experiments performed in duplicates. The *p* values were determined using nonlinear regression followed by the extra sum-of-squares F test.

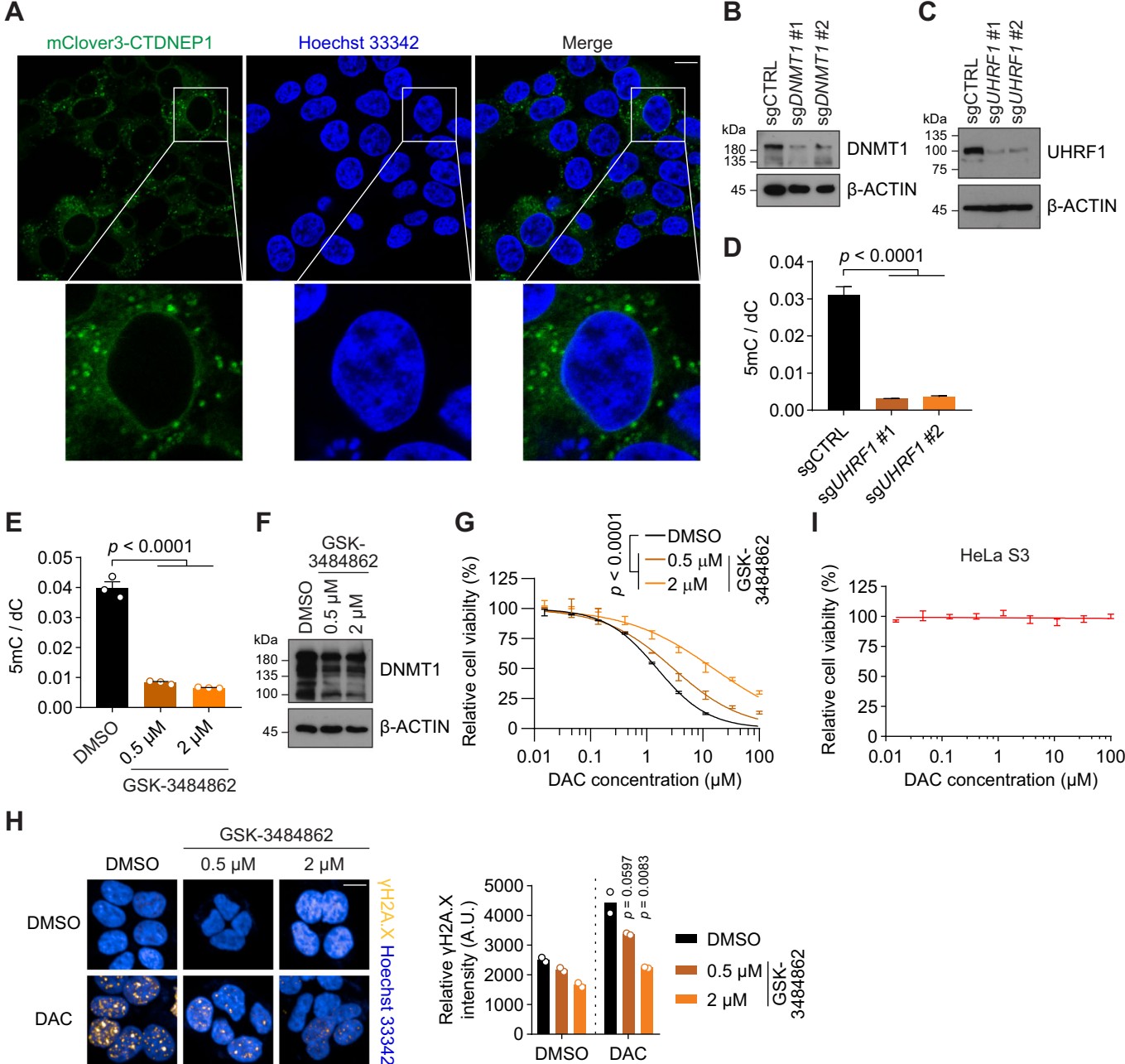

◀

**Figure EV3.  Modulating global DNA methylation level affects the extent of DAC-induced DNA damage.**

(A) Representative confocal images of mClover3-CTDNEP1 in HAP1 cells. The scale bar is 10 μm. (B) Immunoblot analysis showing the DNMT1 protein level in HAP1 cells transduced with either control sgRNA or two different sgRNAs targeting *DNMT1*, with β-ACTIN as a loading control. (C) Immunoblot analysis showing the UHRF1 protein level in HAP1 cells transduced with either control sgRNA or two different sgRNAs targeting *UHRF1*, with β-ACTIN as a loading control. (D) Percentage of 5mC/dC quantified by LC-MS/MS in genomic DNA isolated from HAP1 cells transduced with control sgRNA or two different sgRNAs targeting *UHRF1*. Data represent the mean ± SD ($n = 3$ technical replicates). The *p* values were determined using one-way ANOVA followed by Dunnett's multiple comparisons test. (E) Percentage of 5mC/dC quantified by LC-MS/MS in genomic DNA isolated from HAP1 cells treated with DMSO or DNMT1 inhibitor GSK-3484862 at indicated concentrations. Each data point represents an independent replicate. Data represent the means ± SEM ($n = 3$ replicates). The *p* values were determined using one-way ANOVA followed by Dunnett's multiple comparisons test. (F) Immunoblot analysis showing the DNMT1 protein level in HAP1 cells treated with DMSO or DNMT1 inhibitor GSK-3484862 at indicated concentrations for 3 days, with β-ACTIN as a loading control. (G) Dose–response curves of HAP1 cells pre-treated with DMSO or DNMT1 inhibitor GSK-3484862 at indicated concentrations for 3 days upon DAC treatment. Cell viability was measured by CellTiter-Glo after 3 days of DAC treatment. Data are presented with ± SEM. Experiments performed in duplicates. The *p* values were determined using nonlinear regression followed by the extra sum-of-squares F test. (H) Representative images of γH2A.X immunostaining in HAP1 cells treated with DMSO or DAC (500 nM) for 24 h (left). HAP1 cells were pre-treated with DMSO or DNMT1 inhibitor GSK-3484862 at indicated concentrations for 3 days before DAC treatment. Quantification of γH2A.X immunostaining signal intensity in these cells after DAC treatment (right). Each point represents the mean of one experiment ($N = 2$ experiments; $n > 1800$ cells for each treatment). The *p* values were determined using one-way ANOVA followed by Dunnett's multiple comparisons test. The scale bar is 10 μm. (I) Dose–response curve of HeLa S3 cells upon DAC treatment at indicated concentrations. Cell viability was measured by CellTiter-Glo after 3 days of DAC treatment. Data are presented with ± SEM. Experiments performed in duplicates.

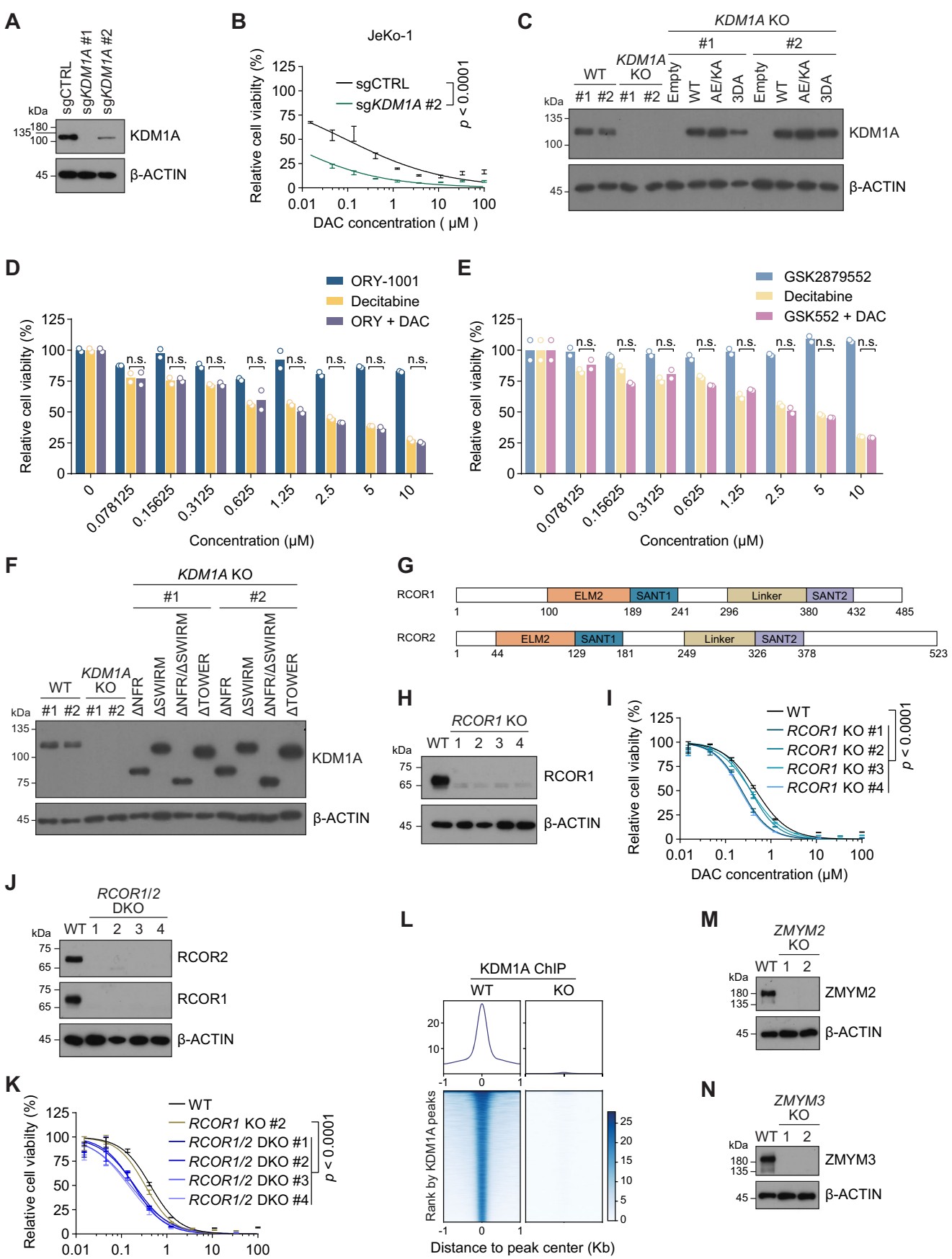

**Figure EV4. Not the demethylase activity of KDM1A, but the interactions between KDM1A with RCOR proteins and ZMYM3 contribute to the cellular response to DAC.**

(A) Immunoblot analysis showing the KDM1A protein level in HAP1 cells transduced with either sgCTRL or two different sgRNAs targeting *KDM1A*. (B) Dose–response curves of JeKo-1 cells transduced with control or *KDM1A*-targeting sgRNAs upon DAC treatment at indicated concentration. Cell viability was measured by CellTiter-Glo after 3 days of DAC treatment. Data are presented with ± SEM. Experiments performed in duplicates. The *p* value was determined using nonlinear regression followed by the extra sum-of-squares F test. (C) Immunoblot analysis showing the KDM1A protein level in WT and *KDM1A* KO HAP1 clones, and *KDM1A* KO clones complemented with either empty vector, KDM1A-WT, KDM1A-AE/KA, or KDM1A-3DA with β-ACTIN as a loading control. (D) Survival analysis of HAP1 cells after treatment with DAC, KDM1A inhibitor ORY-1001, or both at indicated concentrations for 3 days. Cell viability was measured by CellTiter-Glo. Each data point represents one independent experiment. The *p* values were determined using two-way ANOVA followed by Dunnett's multiple comparisons test (n.s., not significant). (E) Survival analysis of HAP1 cells after treatment with DAC, KDM1A inhibitor GSK2879552, or both at indicated concentrations for 3 days. Cell viability was measured by CellTiter-Glo. Each data point represents one independent experiment. The *p* values were determined using two-way ANOVA followed by Dunnett's multiple comparisons test. (F) Immunoblot analysis showing the KDM1A protein level in WT and *KDM1A* KO HAP1 clones, and *KDM1A* KO clones complemented with either KDM1A-ΔNFR, KDM1A-ΔSWIRM, KDM1A-ΔNFR/ΔSWIRM, or KDM1A-ΔTOWER with β-ACTIN as a loading control. (G) Schematic diagrams of the domain structure of full-length RCOR1 and RCOR2. (H) Immunoblot analysis showing the RCOR1 protein level in WT and *RCOR1* KO HAP1 clones with β-ACTIN as a loading control. (I) Dose–response curves of WT and *RCOR1* KO HAP1 clones upon DAC treatment at indicated concentrations. Cell viability was measured by CellTiter-Glo after 3 days of DAC treatment. Data are presented with ± SEM. Experiments performed in duplicates. The *p* values were determined using nonlinear regression followed by the extra sum-of-squares F test. (J) Immunoblot analysis showing the RCOR1 and RCOR2 protein levels in WT and *RCOR1* and *RCOR2* double knockout (DKO) HAP1 clones with β-ACTIN as a loading control. (K) Dose–response curves of WT, *RCOR1* KO #2, and *RCOR1/2* DKO HAP1 clones upon DAC treatment at indicated concentration. Cell viability was measured by CellTiter-Glo after 3 days of DAC treatment. Data are presented with ± SEM. Experiments performed in duplicates. The *p* values were determined using nonlinear regression followed by the extra sum-of-squares F test. (L) KDM1A ChIP-seq metagene profiles at KDM1A binding peaks in HAP1 WT and *KDM1A* KO clones (upper). Heatmap representation of KDM1A occupancies at KDM1A binding peaks in WT and *KDM1A* KO clones ranked by the intensity of KDM1A ChIP-seq signal (lower). (M) Immunoblot analysis showing the ZMYM2 protein level in WT and *ZMYM2* KO HAP1 clones with β-ACTIN as a loading control. (N) Immunoblot analysis showing the ZMYM3 protein level in WT and *ZMYM3* KO HAP1 clones with β-ACTIN as a loading control.

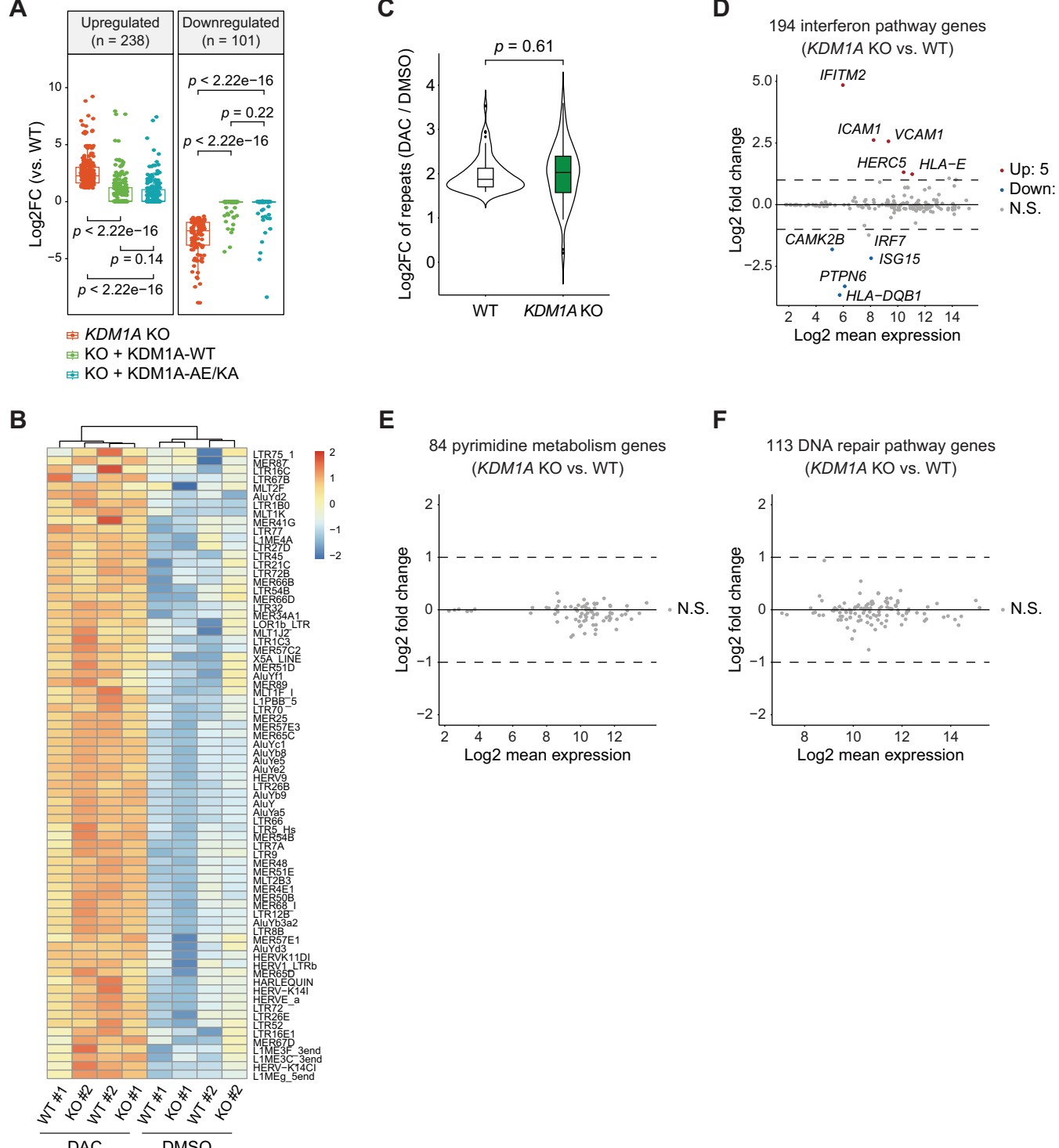

**Figure EV5. Transcriptome analysis in WT and *KDM1A* KO HAP1 clones.**

(A) Box plot depicting transcriptional changes of significantly upregulated and downregulated genes upon loss-of-KDM1A could be rescued by complementation with either KDM1A-WT or KDM1A-AE/KA. Box plots are presented with median (horizontal line) and upper and lower quartile boundaries (box range), plus 1.5 times inter-quartile range (whiskers). FC, fold change. The *p* values were determined by Wilcoxon test. (B) Heatmap depicting loss-of-KDM1A has no significant effect on the transcriptional activation of retroelements after 500 nM DAC treatment for 3 days. (C) Violin plot showing differential expression of retroelements activated by DAC treatment from (B) in HAP1 WT and *KDM1A* KO clones. Box plots are presented with median (horizontal line) and upper and lower quartile boundaries (box range), plus 1.5 times inter-quartile range (whiskers). The *p* value was determined by Wilcoxon test ($n = 74$). (D) MA-plot showing differential expression of interferon pathway genes after *KDM1A* KO in HAP1 cells. (E) MA-plot showing differential expression of pyrimidine metabolism genes after *KDM1A* KO in HAP1 cells. (F) MA-plot showing differential expression of DNA repair genes after *KDM1A* KO in HAP1 cells.

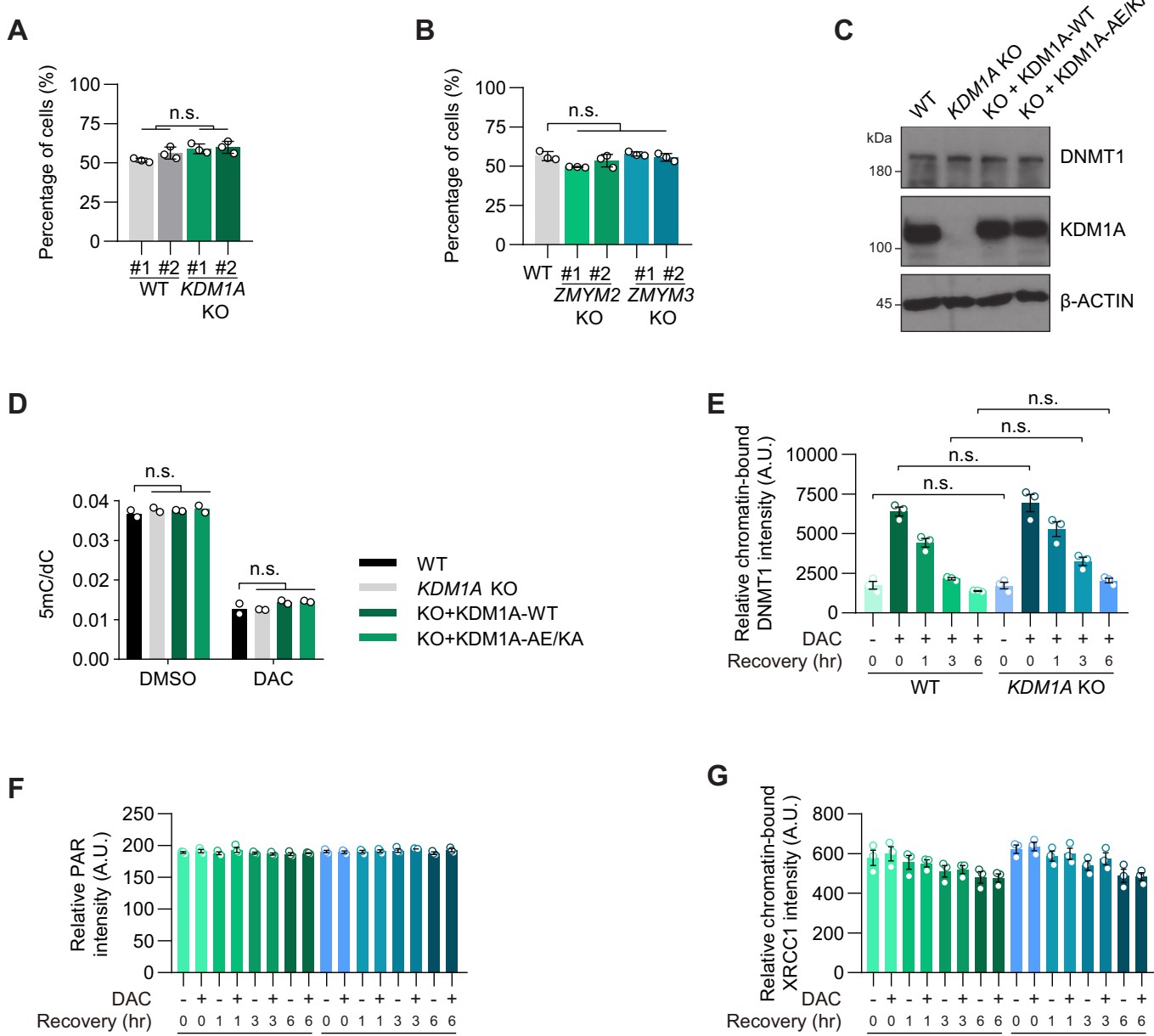

**Figure EV6. KDM1A does not affect S-phase progression, global DNA methylation, or the levels of PARylation and XRCC1 in response to DAC pulse.**

(A) Percentages of cells in S phase in WT and *KDM1A* KO clones. Data are presented as mean ± SEM ($n = 3$ replicates). The *p* values were determined using one-way ANOVA followed by Sidak's multiple comparisons test. (B) Percentages of cells in S phase in WT, *ZMYM2* KO, and *ZMYM3* KO clones. Data are presented as mean ± SEM ($n = 3$ replicates). The *p* values were determined using one-way ANOVA followed by Sidak's multiple comparisons test. (C) Immunoblot analysis showing the DNMT1 protein level in the HAP1 WT, *KDM1A* KO, and *KDM1A* KO clone complemented with either KDM1A-WT or KDM1A-AE/KA, with β-ACTIN as a loading control. (D) Percentage of 5mC/dC quantified by LC-MS/MS in genomic DNA isolated from WT and *KDM1A* KO clones, and KDM1A KO clones complemented with either KDM1A-WT or KDM1A-AE/KA treated with DMSO or 500 nM DAC for 3 days. Experiments performed in duplicates. The *p* values were determined using two-way ANOVA followed by Dunnett's multiple comparisons test. (E) DNMT1 protein levels in HAP1 WT and *KDM1A* KO cells at the indicated time points after release from single-round thymidine synchronization, with or without a 30-min DAC (10 μM) pulse. Data represent the means ± SEM ($n = 3$). The *p* values were determined using one-way ANOVA followed by Sidak's multiple comparisons test. (F) PARylation levels in HAP1 WT and *KDM1A* KO cells at the indicated time points after release from single-round thymidine synchronization, with or without a 30-min DAC (10 μM) pulse. Data represent the means ± SEM ($n = 3$). (G) Chromatin-bound XRCC1 levels in HAP1 WT and *KDM1A* KO cells at the indicated time points after release from single-round thymidine synchronization, with or without a 30-min DAC (10 μM) pulse. Data represent the means ± SEM ($n = 3$). Source data are available online for this figure.

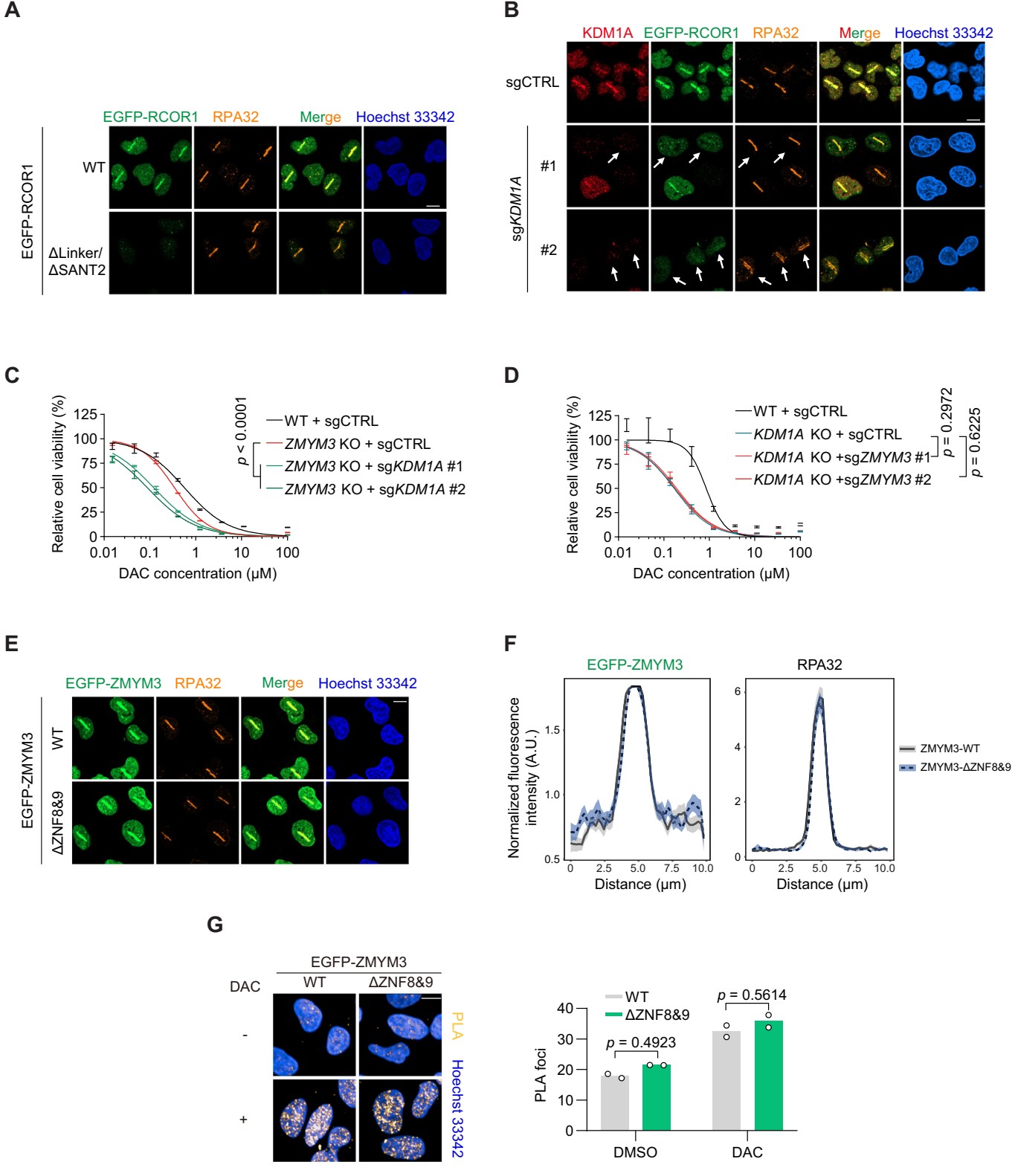

**Figure EV7. ZMYM3 is partially responsible for the recruitment of KDM1A to DNA damage sites.**

(A) Representative images showing the Linker and SANT2 domains of RCOR1 were required for its recruitment to laser-induced damage sites. EGFP-tagged WT or ΔLinker/ΔSANT2 mutant RCOR1 were expressed in U2OS cells. RPA32 was used as a DNA DSB marker. The scale bar is 10 μm. (B) Representative images showing the recruitment of EGFP-tagged RCOR1 to laser-induced damage sites in U2OS cells transduced with control sgRNA or two different sgRNAs targeting *KDM1A*. Arrows indicate *KDM1A*-depleted U2OS cells. RPA32 was used as a DNA DSB marker. The scale bar is 10 μm. (C) Dose–response curves of HAP1 WT and *ZMYM3* KO cells transduced with control sgRNA (sgCTRL) or two different sgRNAs targeting *KDM1A* (sg*KDM1A* #1 and #2) upon DAC treatment at indicated concentrations. Cell viability was measured by CellTiter-Glo after 3 days of DAC treatment. Data are presented with ± SEM. Experiments performed in duplicates. The *p* values were determined using nonlinear regression followed by the extra sum-of-squares F test. (D) Dose–response curves of HAP1 WT and *KDM1A* KO cells transduced with control sgRNA (sgCTRL) or two different sgRNAs targeting *ZMYM3* (sg *ZMYM3* #1 and #2) upon DAC treatment at indicated concentrations. Cell viability was measured by CellTiter-Glo after 3 days of DAC treatment. Data are presented with ± SEM. Experiments performed in duplicates. The *p* values were determined using nonlinear regression followed by the extra sum-of-squares F test. (E) Representative images showing the recruitment of EGFP-tagged wild-type ZMYM3 or ΔZNF8&9 mutant to laser-induced damage sites in U2OS cells. RPA32 was used as a DNA DSB marker. The scale bar is 10 μm. (F) Quantification for profiles of EGFP-tagged ZMYM3 and RPA32 immunostaining signals along the lines across damage stripes in (E). The relative signal intensities were normalized to the mean of U2OS cells expressing EGFP-tagged wild-type ZMYM3. Data represent mean ± SEM ($n = 24$ cells for ZMYM3-WT; $n = 26$ cells for ZMYM3-ΔZNF8&9). (G) Representative images (left) of PLA results between EGFP and γH2A.X in U2OS cells expressing either EGFP-tagged wild-type ZMYM3 or ΔZNF8&9 mutant 24 h after release from single-round thymidine synchronization, with or without a 30-min DAC (10 μM) pulse. Quantification of PLA foci number (right) in these cells. Each point represents the mean of one experiment ($N = 2$ experiments; $n > 5000$ cells for each condition). The *p* values were determined using one-way ANOVA followed by Sidak's multiple comparisons test. The scale bar is 10 μm.

