## [Peer Review File · EMBO Reports]

A CRISPR-Cas9 Screen Reveals Genetic Determinants of the Cellular Response to Decitabine

Pinqi Zhang, Zhuqiang Zhang, Yiyi Wang, Wenlong Du, Xingrui Song, Weiyi Lai, Hailin Wang, Bing Zhu, and Jun Xiong

Corresponding author(s): Jun Xiong (xiongjun@ibp.ac.cn) , Bing Zhu (zhubing@ibp.ac.cn)

Review Timeline:

Transfer Date:	29th Sep 24
Editorial Decision:	2nd Oct 24
Revision Received:	6th Jan 25
Editorial Decision:	23rd Jan 25
Revision Received:	27th Jan 25
Accepted:	28th Jan 25

Editor: *Esther Schnapp*

Transaction Report: This manuscript was transferred to EMBO reports following peer review at The EMBO Journal.

Referee #1:

In this manuscript, Zhang et al. explore genetic determinants of decitabine (DAC) sensitivity using CRISPR-Cas9 screens. In the first part of the manuscript, the authors validate some of the screening hits with individual guide RNAs and provide additional supporting evidence. These genes include nucleotide metabolism genes (DCTD), Fanconi anemia genes (FANCA, FANCD2, FANCL and UBE2T), CTDNEP1, and CNEP1R1, which they demonstrate influence DNA repair, as well as genes that affect DNMT1-DNA adduct levels (DNMT1 itself and UHRF1). In the latter half, they focus on one of the epigenetic gene hits, KDM1A. The authors find that KDM1A's ability to protect cells from DAC is independent of its demethylase activity but dependent on the SWIRM and TOWER domains. They show that these domains are essential for interaction with ZMYM3, a protein that localizes at DNA damage sites and facilitates homologous recombination (HR), and for KDM1A chromatin loading. However, the study shows that KDM1A KO does not impair DNA repair, yet the authors still conclude that KDM1A promotes cellular survival in response to DAC through its involvement in the DNA repair process.

As it stands, the manuscript seems more like a verification of screening results than an exploration of new mechanisms for DAC actions. The section on KDM1A begins to address this point but ultimately leaves the role of KDM1A at DNA damage sites unclear.

A similar CRISPR screening study on DAC by Carnie et al., (2024) was recently published in EMBO J. and reported novel findings of the DCTD-mediated DAC cytotoxicity and the new DNA-protein crosslink repair pathway involving TOPORS. Unfortunately, this manuscript does not have the same level of novelty or impact. Below are my comments to help the authors enhance the manuscript.

Major points:

1. Fig. 2C-E (and other similar figures): How p-values were calculated need to be explained in the Method section. The figure legends states that "the p values were determined using nonlinear regression", but nonlinear regression is about fitting curves to the data, not for hypothesis testing (whether the KO cells have different DAC sensitivity compared to the control). The authors could use t-test on IC50 values in control vs. KO #1 and control vs. KO #2 separately or choose more complex methods.
2. Fig. 4H: Whether the SWIRM and TOWER domains are necessary for chromatin loading of KDM4A should be addressed in the context of DAC treatment, as the mechanism of KDM1A loading in the absence of DAC treatments shown here could be different from DAC-induced chromatin loading. More studies are needed for DAC-induced chromatin loading of KDM4A. For example, do KDM4 and ZMYM3 colocalize with DAC-induced γ H2AX foci? Does it require the SWIRM and TOWER domains? (See also Point #4.)
3. Lines 387 and 436, Fig. 5D-F: Although the authors claim DNA repair deficiency in KDM4A KO cells, DNA repair appears intact in KDM4A KO cells in these reporter assays (HR efficiency is rather higher). If it is not required for the damage repair, what does KDM4A do at the DNA damage sites and how does it protect cells from DAC? It is very important to address these questions.
4. Fig. 6: KDM4A recruitment to DNA damage sites need to be assessed in the contest of DAC treatment, not by the laser-induced DNA damage. Does KDM4 colocalize to γ H2AX foci in DAC-

treated cells? (See also Pint #2.) If so, is it dependent on the SWIRM and TOWER domains? Is KDM4 recruitment dependent on ZMYM3, or vice versa?

5. The paragraph starting from 481: Although authors propose DAC as a low side effect drug in FA patients, the presented data might rather suggest that FA patients could experience high DAC toxicity in the bone marrow, which is highly proliferative.

Minor points:

1. The paragraph starting from Line 182 is speculative and would be more suitable for Discussion.
2. Primary research papers, not review articles, should be cited whenever possible. For example (but not limited to), lines 199, 200, 208, and 248.
3. Fig. 3F: Needs lower DAC concentrations to demonstrate dose-responsiveness.
4. Fig. 4E: Statistics is missing for clone #1
5. Fig. 4F: This domain scanning study would have been more effective with base-editing approach (for example, PMID: 35288574), as the Cas9-based approach used here truncates the protein at the guide RNA target through frameshift, essentially removing all domains C-terminal to the CRISPR targets.
6. Fig. 4K: Where are ZMYM2 and ZMYM3 ranked in the initial CRISPR screens?
7. Fig. 6C: Needs description of statistics used to determine p-values in the Methods section.
8. Fig. 6E,G: Quantitation of RPA needs to be included to show that the levels of DNA damage is comparable between samples.
9. Line 509: "(The striking) negative correlation between DNMT1 expression and DAC sensitivity ..." might be incorrect. Because high AUC means less sensitive, the negative correlation between DNMT1 expression levels and AUC values would suggest positive correlation between DNMT1 expression and DAC sensitivity, which is probably what the authors meant to say.

Referee #2:

In the manuscript entitled "A CRISPR-Cas9 screen reveals genetic determinants of cellular response to decitabine" by Zhang & Zhang et al, the authors seek to elucidate cellular mechanisms governing sensitivity and resistance to the hypomethylating agent decitabine, a clinically-relevant compound marketed as Dacogen (DAC). To this end, they perform an unbiased, genome-scale CRISPR/Cas9 screen in the CML-derived cell line HAP1. The authors perform follow-up investigation and validation of a range of hits arising from their screen that highlight both new and known genetic determinants of DAC tolerance. The authors follow up on one of these hits in particular - the lysine demethylase KDM1A - finding that KDM1A promotes cellular DAC tolerance. Interestingly, KDM1A's role in DAC tolerance is independent of its catalytic activity and rather appears to constitute a structural or scaffolding function that might mediate HDAC activity through its interaction with CoREST to alleviate the toxic effects of DAC-induced DNA damage.

This manuscript is timely and coincides with other recent publications that collectively are revealing new aspects of cellular responses to DAC that could inform further translationally-

mindful studies. The authors should be commended for trying to address multiple aspects of these responses informed by their screen. However, this approach does mean that the flow of the manuscript can feel disjointed in places, and the discussion of some of their findings is rather superficial. Despite this, the authors uncover a role for KDM1A in DAC tolerance that appears to involve its somewhat unclear role in DNA repair. While interesting, however, the function of KDM1A in this context is not explored in depth and demonstration of the relevance of these findings in more clinically-relevant contexts is lacking.

Overall, I like several aspects of this study but feel that to be suitable for publication in *The EMBO Journal*, the authors would need to strengthen their manuscript substantially. However, while I have several substantial major comments, I would not expect all of them to be addressed fully - in my opinion, deeper investigation of one or two of the three major points below would improve the manuscript more than a surface-level attempt at all three. If the authors are unable to successfully address some of the issues highlighted below, then a shorter more focused version of their work could be suitable for publication in a journal such as *EMBO Reports*.

Major comments:

1. The appearance of proteasome subunits as mild resistance hits in the CRISPR screen is very surprising, particularly given that depletion of the E3 ubiquitin ligases promoting DNMT1-DPC degradation by the proteasome causes cellular sensitivity to DAC (Liu et al *EMBO J* 2021 [this work is relevant and should be cited], Liu et al *NSMB* 2024, Carnie et al *EMBO J* 2024). The authors should attempt to validate the effects implied by their screen, although this may not be trivial in cell survival assays. As an alternative, they could assess DNA damage induction following DAC treatment in the presence/absence of proteasome inhibition and/or Ub E1 inhibition. I would recommend the latter as it is likely better tolerated by cells.

2. CTDNEP1's role is speculatively linked to DSB repair, but its connection to DAC toxicity isn't established as a direct link. I suggest looking at repair rates of DAC-induced DSBs in sgCtrl/sGCTDNEP1 cells (e.g. by gH2AX foci induction/disappearance kinetics). IP/MS or phospho-proteomic approaches to try to relevant interactors/substrates of CTDNEP1 could be valuable here.

3. For me, the nature of DNA lesion for which KDM1A is relevant is an important question to address. The authors assess DSB induction and repair kinetics using gH2AX immunofluorescence, but the origin of DSBs after DAC is unclear and could feasibly relate to proteasome activity of SUMOylated DPCs. As I see it, the likely origin of DSBs is downstream of DNMT1-DPC degradation and potentially also downstream of SSB induction upon base excision repair of unmethylated, genome-embedded 5-aza-dC/5-aza-dU. Proximal SSBs could lead to DSB formation. This begs the question: at which stage does KDM1A act? From the authors' data, it appears to be downstream of DNMT1-DPC degradation, but it's hard to say where it acts. I suggest attempting the following experiments from which relevant insights might be gained:

a. Repeating the gH2AX kinetic experiment in Fig 5C with cell cycle synchronisation and release into S-phase in the presence or absence of DAC for a short time. This will allow the authors to assess

whether DSB induction is altered in KDM1A KO cells and track the repair kinetics.

b. The use of a more sensitive readout of DNMT1-DPC induction/repair than bulk DNMT1 degradation. The recently-developed PxP assay (Weickert et Nat Comms al 2023) is one candidate, but the authors could also address this using less specific assays such as chromatin fractionation after DAC treatment/release.

c. Assessment of SSB formation/repair in KDM1A KO cells after DAC treatment by alkaline comet assay.

Minor comments:

1. The use of several timepoints in the DAC CRISPR screen is a thorough approach. However, the use of MAGeCK to analyse the screen outputs seems inferior to Chronos (Dempster et al, Genome Biology 2021), an analysis pipeline specifically designed for consolidating multiple time points. I recommend re-analysing the screens using Chronos, as this could make Figure 1 more intuitive to readers.

2. I am struggling to understand the rationale behind the KDM1A domain scanning approach. If I have interpreted it correctly, Cas9 will just induce DSBs across the gene, but given that frameshifting mutations and nonsense-mediated decay will cause many knockouts that affect more domains than just the one targeted by a particular sgRNA. If I have misinterpreted the technique, the description of it should be reworded.

3. Line 247: methylation by DNMT1 occurs during S-phase, not mitosis.

4. The innate resistance of HeLa S3 cells is quite striking - is there anything available in publica datasets that could explain this? Do HeLa cells bear a DCTD mutation, for example?

5. Are there relevant genetic interactions between KDM1A and CoREST or ZMYM3? The authors' model would predict that KDM1A depletion from CoREST or ZMYM3-depicient cells does not further sensitise cells to DAC.

6. To rule out cell cycle effects on DAC-induced DNA damage/repair, it would be helpful to see cell cycle distribution profiles of the relevant KO cell lines tested, especially those for ZMYM3 and KDM1A.

7. Line 449-452/Fig 6B. This experiment should be repeated using gH2AX immunostaining rather than RPA. This would avoid the need for pre-extraction, and retain visible GFP-KDM1A dSWIRM/dTOWER mutants within nuclei and make the results more intuitive to readers.

8. Is KDM1A increased on chromatin after DAC treatment? The authors could assess GFP-KDM1A on chromatin using pre-extraction and imaging after DAC.

9. Is KDM1A recruitment to micro-irradiation sites or chromatin after DAC SUMO-dependent? This could be through either DNMT1-DPC SUMOylation or in response to DSBs. The authors could use

inhibitors of SUMOylation machinery (e.g. TAK-981 or ML-792) to look at SUMO-dependent recruitment.

10. Line 489-494: I disagree with this line of reasoning. Given that FA patients experience severe systemic sensitivity to ICL-inducing reagents, there is no reason to expect something different for DAC. The Fanconi pathway is only active in S/G2 in proliferating cells, in the same way that DAC only induces DNMT1-DNA adducts in S-phase in proliferating cells. Such statements should be made with much more caution and should at most propose a rationale for assessment in mouse models.

11. Line 544: it would be helpful to include a visual representation of DCTD mutation rather than just a link to cbioportal's home page. I am not hugely familiar with using cbioportal but couldn't find any indication that 'DCTD homo-deletion is the most common mutation in cancer patients and cultured tumor cell lines'.

12. Fig 5A: only 2 replicates. This is an interesting phenotype so it would be nice to see another replicate.

13. It would be of broad interest to assess some of the core DAC sensitivity phenotypes with Azacytidine as well, since these compounds are considered interchangeable clinically but this might not be the case.

14. Several important studies have not been cited. These include:

- a. Orta et al Nucleic Acids Research 2013 (description of the Fanconi pathway's relevance to DAC tolerance).
- b. Borgermann et al EMBO J 2019 (SUMOylation and proteasomal degradation of DNMT1-DPCs after DAC).
- c. Liu et al EMBO J 2021 (SUMO-targeted ubiquitylation of DNMT1-DPCs by RNF4).
- d. Weickert et al Nat Comms 2023 (DNMT1-DPC degradation is proteasome-dependent but also contributed to by the DPC protease SPRTN).

Response to reviewers' comments:

Referee #1:

In this manuscript, Zhang et al. explore genetic determinants of decitabine (DAC) sensitivity using CRISPR-Cas9 screens. In the first part of the manuscript, the authors validate some of the screening hits with individual guide RNAs and provide additional supporting evidence. These genes include nucleotide metabolism genes (DCTD), Fanconi anemia genes (FANCA, FANCD2, FANCL and UBE2T), CTDNEP1, and CNEP1R1, which they demonstrate influence DNA repair, as well as genes that affect DNMT1-DNA adduct levels (DNMT1 itself and UHRF1). In the latter half, they focus on one of the epigenetic gene hits, KDM1A. The authors find that KDM1A's ability to protect cells from DAC is independent of its demethylase activity but dependent on the SWIRM and TOWER domains. They show that these domains are essential for interaction with ZMYM3, a protein that localizes at DNA damage sites and facilitates homologous recombination (HR), and for KDM1A chromatin loading. However, the study shows that KDM1A KO does not impair DNA repair, yet the authors still conclude that KDM1A promotes cellular survival in response to DAC through its involvement in the DNA repair process.

As it stands, the manuscript seems more like a verification of screening results than an exploration of new mechanisms for DAC actions. The section on KDM1A begins to address this point but ultimately leaves the role of KDM1A at DNA damage sites unclear.

A similar CRISPR screening study on DAC by Carnie et al., (2024) was recently published in EMBO J. and reported novel findings of the DCTD-mediated DAC cytotoxicity and the new DNA-protein crosslink repair pathway involving TOPORS. Unfortunately, this manuscript does not have the same level of novelty or impact. Below are my comments to help the authors enhance the manuscript.

Major points:

1. Fig. 2C-E (and other similar figures): How p-values were calculated need to be explained in the Method section. The figure legends states that "the p values were determined using nonlinear regression", but nonlinear regression is about fitting curves to the data, not for hypothesis testing (whether the KO cells have different DAC sensitivity compared to the control). The authors could use t-test on IC50 values in control vs. KO #1 and control vs. KO #2 separately or choose more complex methods.

We appreciate the reviewer's suggestion for clarification regarding the statistical methods. In our analysis, we employed nonlinear regression for fitting the dose-response curves, and the extra sum-of-squares F test was used for hypothesis testing. This test evaluates whether a single model can adequately fit the data for both control and knockout (KO) groups or if separate models with different IC50 values are necessary. Additionally, we found that similar results could be obtained using a two-way ANOVA with genotype as the main effect.

We will include a detailed description of the statistical analysis method in the Methods section, specifying the use of nonlinear regression for curve fitting and the extra sum-of-squares F test for hypothesis testing. We will also update the figure legends accordingly to accurately reflect the methods used.

2. Fig. 4H: Whether the SWIRM and TOWER domains are necessary for chromatin loading of KDM4A should be addressed in the context of DAC treatment, as the mechanism of KDM1A loading in the absence of DAC treatments shown here could be different from DAC-induced chromatin loading. More studies are needed for DAC-induced chromatin loading of KDM4A. For example, do KDM4 and ZMYM3 colocalize with DAC-induced γ H2AX foci? Does it require the SWIRM and TOWER domains? (See also Point #4.)

We appreciate the reviewer's insightful comments regarding the chromatin loading mechanism of KDM1A in the context of DAC treatment. As presented in Fig. 4H, our ChIP-seq data demonstrate that both SWIRM and TOWER domains are necessary for KDM1A's chromatin binding under unperturbed conditions, primarily at gene promoters and enhancers. Additionally, our immunofluorescence results in Fig. 6B showed that both

domains are critical for KDM1A recruitment and stable association at laser-induced DNA damage sites. Together, we conclude that SWIRM and TOWER domains are both required for chromatin association of KDM1A.

We recognize that the chromatin loading of KDM1A under DAC-induced conditions may differ from its recruitment to laser-irradiated regions. To directly address the reviewer's concern, we plan to perform additional quantitative immunofluorescence experiments and proximity ligation assay (PLA) to investigate whether KDM1A colocalizes with DAC-induced γ H2A.X foci and to verify if the recruitment is dependent on the SWIRM and TOWER domains under DAC treatment.

3. Lines 387 and 436, Fig. 5D-F: Although the authors claim DNA repair deficiency in KDM4A KO cells, DNA repair appears intact in KDM4A KO cells in these reporter assays (HR efficiency is rather higher). If it is not required for the damage repair, what does KDM4A do at the DNA damage sites and how does it protect cells from DAC? It is very important to address these questions.

We appreciate the reviewer's insightful comments. KDM1A indeed plays a crucial role in DNA repair, as evidenced by the increased sensitivity of KDM1A KO cells to both DAC and Zeocin, as well as the delayed clearance of DNA damages (Fig. 4A-B and 5A-C). However, as the reviewer pointed out, while NHEJ does not appear to be affected, HR efficiency is elevated in KDM1A KO cells, as indicated by reporter assays. We speculate that this enhanced HR rate may result from increased DNA end resection in the absence of KDM1A, which normally helps to maintain chromatin compaction at DNA damage sites. Excessive end resection could compromise genome stability, as suggested by Tomimatsu et al. (J. Biol. Chem, 2017). To further elucidate KDM1A's role in DNA repair, we plan to measure the recruitment and disengagement kinetics of key DNA repair factors (e.g. γ H2A.X, 53BP1, and RPA32) following DAC treatment in both wild-type and KDM1A KO cells, which is also suggested by Reviewer 2. These experiments will help clarify how KDM1A contributes to DNA repair and how its loss impacts genome instability.

4. Fig. 6: KDM4A recruitment to DNA damage sites need to be assessed in the context of

DAC treatment, not by the laser-induced DNA damage. Does KDM4 colocalize to γ H2AX foci in DAC-treated cells? (See also Pint #2.) If so, is it dependent on the SWIRM and TOWER domains? Is KDM4 recruitment dependent on ZMYM3, or vice versa?

We appreciate the reviewer's suggestion. As mentioned in our response to major point #2, we plan to perform PLA to detect the colocalization of KDM1A with γ H2A.X in DAC-treated cells. In addition, we will investigate whether KDM1A's recruitment is dependent on the SWIRM and TOWER domains under DAC treatment.

5. The paragraph starting from 481: Although authors propose DAC as a low side effect drug in FA patients, the presented data might rather suggest that FA patients could experience high DAC toxicity in the bone marrow, which is highly proliferative.

We thank the reviewer for highlighting this issue, which is also raised by Reviewer 2. Given the potential for high DAC toxicity in bone marrow, as suggested by the reviewers, we will remove this statement from the discussion to avoid any misinterpretation.

Minor points:

1. The paragraph starting from Line 182 is speculative and would be more suitable for Discussion.

We acknowledge the reviewer's concern regarding the speculative nature of this paragraph. However, given Reviewer 2's interest in this topic, we plan to conduct experiments to verify whether proteasome inhibition affects DNA damage in the context of DNMT1 degradation. The results of these experiments will allow us to provide more concrete data in this section. Depending on the results, we will revise the text to include this data in the Results section, and if necessary, move any remaining speculative content to the Discussion.

2. Primary research papers, not review articles, should be cited whenever possible. For example (but not limited to), lines 199, 200, 208, and 248.

We will replace review article citations with primary research papers throughout the manuscript, focusing on the specified lines and ensuring that citations are appropriate and

relevant.

3. Fig. 3F: Needs lower DAC concentrations to demonstrate dose-responsiveness.

We appreciate the reviewer's suggestion. We will conduct additional experiments with lower DAC concentrations in Fig. 3F and update the results and figure accordingly in the revised manuscript.

4. Fig. 4E: Statistics is missing for clone #1

We will include the missing statistical analysis for clone #1 in our revised manuscript.

5. Fig. 4F: This domain scanning study would have been more effective with base-editing approach (for example, PMID: 35288574), as the Cas9-based approach used here truncates the protein at the guide RNA target through frameshift, essentially removing all domains C-terminal to the CRISPR targets.

We appreciate the reviewer's suggestion regarding the use of base-editing, which is indeed a more precise approach for pinpointing functional residues in a protein. Reviewer 2 also raised a related concern about using the Cas9-based approach (Reviewer 2's minor point #2). However, for the purpose of our study, the Cas9-based domain scanning approach is more suited to our aim of identifying essential regions or domains of KDM1A under DAC treatment. Following Cas9 editing, indel mutations have a 1/3 probability of being in-frame, potentially producing a protein product with a local deletion. Cells with functional in-frame mutations are likely to survive under negative selection, while those with local deletions in critical region will be sensitized. Consequently, sgRNAs targeting essential domains will be depleted during the screening. This approach is supported by the study from Shi et al., Nat Biotechnol. (2015), which demonstrates the effectiveness of Cas9-mediated domain scanning for identifying critical protein regions. Compared to base-editing, domain scanning allows us to more quickly and efficiently determine essential domains for the purpose of this study.

6. Fig. 4K: Where are ZMYM2 and ZMYM3 ranked in the initial CRISPR screens?

We thank the reviewer for the question. Indeed, ZMYM2 and ZMYM3 were not significant hits in our CRISPR screen. However, as demonstrated by our data, loss of ZMYM3, but not ZMYM2, sensitizes cells to DAC (Fig. 4K). Additionally, only depletion of ZMYM3 reduced KDM1A recruitment to DNA damage sites (Fig. 6D-G). This reduction is partial, which likely explains why ZMYM3 was not a significant hit in the screen. We speculate that other redundant proteins may help stabilize KDM1A chromatin binding, thus reducing the impact of ZMYM3 loss in the screen.

7. Fig. 6C: Needs description of statistics used to determine p-values in the Methods section.

We appreciate the reviewer's attention to this detail. We have already annotated the statistical method used to determine p-values in the figure legends. However, to ensure clarity and consistency, we will also add this information to the Methods section in our revised manuscript.

8. Fig.6E,G: Quantitation of RPA needs to be included to show that the levels of DNA damage is comparable between samples.

We appreciate the reviewer's suggestion and will perform the RPA quantification analysis during the revision.

9. Line 509: "(The striking) negative correlation between DNMT1 expression and DAC sensitivity ..." might be incorrect. Because high AUC means less sensitive, the negative correlation between DNMT1 expression levels and AUC values would suggest positive correlation between DNMT1 expression and DAC sensitivity, which is probably what the authors meant to say.

We thank this reviewer for pointing out this mistake, we will rephrase our description to accurately reflect the correlation between DNMT1 expression levels and DAC sensitivity in our revised manuscript.

Referee #2:

In the manuscript entitled "A CRISPR-Cas9 screen reveals genetic determinants of cellular response to decitabine" by Zhang & Zhang et al, the authors seek to elucidate cellular mechanisms governing sensitivity and resistance to the hypomethylating agent decitabine, a clinically-relevant compound marketed as Dacogen (DAC). To this end, they perform an unbiased, genome-scale CRISPR/Cas9 screen in the CML-derived cell line HAP1. The authors perform follow-up investigation and validation of a range of hits arising from their screen that highlight both new and known genetic determinants of DAC tolerance. The authors follow up on one of these hits in particular - the lysine demethylase KDM1A - finding that KDM1A promotes cellular DAC tolerance. Interestingly, KDM1A's role in DAC tolerance is independent of its catalytic activity and rather appears to constitute a structural or scaffolding function that might mediate HDAC activity through its interaction with CoREST to alleviate the toxic effects of DAC-induced DNA damage.

This manuscript is timely and coincides with other recent publications that collectively are revealing new aspects of cellular responses to DAC that could inform further translationally-minded studies. The authors should be commended for trying to address multiple aspects of these responses informed by their screen. However, this approach does mean that the flow of the manuscript can feel disjointed in places, and the discussion of some of their findings is rather superficial. Despite this, the authors uncover a role for KDM1A in DAC tolerance that appears to involve its somewhat unclear role in DNA repair. While interesting, however, the function of KDM1A in this context is not explored in depth remains unclear and demonstration of the relevance of these findings in more clinically-relevant contexts is lacking.

Overall, I like several aspects of this study but feel that to be suitable for publication in The EMBO Journal, the authors would need to strengthen their manuscript substantially. However, while I have several substantial major comments, I would not expect all of them to be addressed fully - in my opinion, deeper investigation of one or two of the three major

points below would improve the manuscript more than a surface-level attempt at all three. If the authors are unable to successfully address some of the issues highlighted below, then a shorter more focused version of their work could be suitable for publication in a journal such as EMBO Reports.

Major comments:

1. The appearance of proteasome subunits as mild resistance hits in the CRISPR screen is very surprising, particularly given that depletion of the E3 ubiquitin ligases promoting DNMT1-DPC degradation by the proteasome causes cellular sensitivity to DAC (Liu et al EMBO J 2021 [this work is relevant and should be cited], Liu et al NSMB 2024, Carnie et al EMBO J 2024). The authors should attempt to validate the effects implied by their screen, although this may not be trivial in cell survival assays. As an alternative, they could assess DNA damage induction following DAC treatment in the presence/absence of proteasome inhibition and/or Ub E1 inhibition. I would recommend the latter as it is likely better tolerated by cells.

We thank the reviewer for pointing out this interesting observation and for suggesting relevant studies. The involvement of proteasome subunits in DAC resistance is intriguing. Following the reviewer's suggestion, we will conduct experiments to investigate whether proteasome inhibition affects DNA damage in the context of DNMT1 degradation.

2. CTDNEP1's role is speculatively linked to DSB repair, but its connection to DAC toxicity isn't established as a direct link. I suggest looking at repair rates of DAC-induced DSBs in sgCtrl/sgCTDNEP1 cells (e.g. by gH2AX foci induction/disappearance kinetics). IP/MS or phospho-proteomic approaches to try to relevant interactors/substrates of CTDNEP1 could be valuable here.

We thank the reviewer for raising a valuable point regarding the role of CTDNEP1 in DAC toxicity and DNA repair, which aligns with our own research interests. We are currently pursuing a project on this topic and anticipate gaining further insights in the near future. We recognize the importance of establishing a direct link between CTDNEP1 and these

processes and will aim to explore this further as our research progresses.

3. For me, the nature of DNA lesion for which KDM1A is relevant is an important question to address. The authors assess DSB induction and repair kinetics using gH2AX immunofluorescence, but the origin of DSBs after DAC is unclear and could feasibly relate to proteasome activity of SUMOylated DPCs. As I see it, the likely origin of DSBs is downstream of DNMT1-DPC degradation and potentially also downstream of SSB induction upon base excision repair of unmethylated, genome-embedded 5-aza-dC/5-aza-dU. Proximal SSBs could lead to DSB formation. This begs the question: at which stage does KDM1A act? From the authors' data, it appears to be downstream of DNMT1-DPC degradation, but it's hard to say where it acts. I suggest attempting the following experiments from which relevant insights might be gained:

The origin of DSB induced by DAC is a highly intriguing question, and we thank the reviewer for raising this point. As supported by our data and the recent study from Carnie et al. (EMBO J, 2024), SSBs are primarily induced by 5-aza-dC/5-aza-dU under DAC treatment, and DCTD-deficient cells are resistant to DAC. If KDM1A promotes cellular resistance to DAC through SSB induction or repair, the loss of KDM1A should not sensitize DCTD-deficient cells to DAC anymore. However, indeed, KDM1A is one of the top hits, whose loss sensitizes cells to DAC in the absence of DCTD, as shown in Figure 3G of Carnie et al. (we have adopted and pasted this figure below, highlighting KDM1A with a red rectangle). Notably, KDM1A ranks even higher in the screen for DAC sensitivity with DCTD-deficient cells compared to DCTD-intact cells (Carnie et al., EMBO J, 2024). Additionally, KDM1A does not show enrichment at the DNMT1-DPC sites, as indicated by iPOND experiments, where TOPORS and other related proteins are recruited (Carnie et al., EMBO J, 2024), suggesting that KDM1A is unlikely to be involved in DNMT1-DPC formation or degradation. Instead, these observations support the idea that KDM1A functions downstream of DNMT1-DPC degradation, potentially playing a role in resolving SSB or other DNA repair pathways. We would like to elaborate on this point further in our revise manuscript.

a. Repeating the gH2AX kinetic experiment in Fig 5C with cell cycle synchronisation and release into S-phase in the presence or absence of DAC for a short time. This will allow the authors to assess whether DSB induction is altered in KDM1A KO cells and track the repair kinetics.

We appreciate the reviewer's suggestion, and we will perform this experiment to dissect where KDM1A functions upon DAC treatment.

b. The use of a more sensitive readout of DNMT1-DPC induction/repair than bulk DNMT1 degradation. The recently-developed PxP assay (Weickert et Nat Comms al 2023) is one candidate, but the authors could also address this using less specific assays such as chromatin fractionation after DAC treatment/release.

We appreciate the reviewer's suggestion, and we plan to re-evaluate the induction and degradation of DNMT1-DNA adducts under DAC treatment by quantitative immunofluorescence experiments with pre-extraction.

c. Assessment of SSB formation/repair in KDM1A KO cells after DAC treatment by alkaline comet assay.

We appreciate the reviewer's insightful suggestion regarding the nature of DNA lesions following DAC treatment. However, single- and double-strand breaks are both prevalent in DAC-treated cells, and while alkaline comet assay is highly sensitive, it detects both types

of lesions without clear differentiation. Given this limitation and focus of our current study, we believe that a detailed analysis of specific DNA lesions induced by DAC-treatment is beyond the current scope. Nevertheless, we plan to conduct immunostaining of single-strand break repair factors, such as XRCC1 or poly ADP-ribose (PAR), to monitor the levels and dynamics of single-strand breaks induced by DAC-treatment. This approach will provide an indication of single-strand break repair following DAC exposure in KDM1A KO cells.

Minor comments:

1. The use of several timepoints in the DAC CRISPR screen is a thorough approach. However, the use of MAGeCK to analyse the screen outputs seems inferior to Chronos (Dempster et al, Genome Biology 2021), an analysis pipeline specifically designed for consolidating multiple time points. I recommend re-analysing the screens using Chronos, as this could make Figure 1 more intuitive to readers.

We appreciate the reviewer's suggestion and we will perform this analysis.

2. I am struggling to understand the rationale behind the KDM1A domain scanning approach. If I have interpreted it correctly, Cas9 will just induce DSBs across the gene, but given that frameshifting mutations and nonsense-mediated decay will cause many knockouts that affect more domains than just the one targeted by a particular sgRNA. If I have misinterpreted the technique, the description of it should be reworded.

We appreciate the reviewer for raising this concern. Cas9-based domain scanning is well-suited to our aim of identifying essential regions or domains of KDM1A under DAC treatment. Following Cas9 editing, indel mutations have a 1/3 probability of being in-frame, potentially producing a protein product with a local deletion. Cells with functional in-frame mutations are likely to survive under negative selection, while those with critical region deletions will be sensitized. Consequently, sgRNAs targeting essential domains will be depleted during the screening. On the other hand, frameshift mutations that can be caused by all sgRNAs will lead to knockout clones, and those clones will behave similarly,

perishing equally under negative selection. This approach is supported by the study from Shi et al., Nat Biotechnol. (2015), which demonstrates the effectiveness of domain scanning for identifying critical protein regions.

3. Line 247: methylation by DNMT1 occurs during S-phase, not mitosis.

This will be corrected in our revised manuscript.

4. The innate resistance of HeLa S3 cells is quite striking - is there anything available in public datasets that could explain this? Do HeLa cells bear a DCTD mutation, for example?

We appreciate the reviewer's insightful comment regarding the innate resistance of HeLa cells to DAC. This resistance has also been observed by Palić et al. in their 2008 publication in Molecular and Cellular Biology. Following the reviewer's suggestion, we will survey gene mutations in HeLa cell as annotated in available public datasets. We plan to conduct a cross-analysis of this mutation profile with our screening hits to provide potential explanations for the innate resistance of HeLa cells to DAC.

5. Are there relevant genetic interactions between KDM1A and CoREST or ZMYM3? The authors' model would predict that KDM1A depletion from CoREST or ZMYM3-deficient cells does not further sensitise cells to DAC.

We appreciate the reviewer's insightful question regarding the genetic interaction between KDM1A and CoREST or ZMYM3. While KDM1A forms stable protein complex with CoREST, knocking out ZMYM3 led to a partial reduction in KDM1A recruitment to DNA damage sites (Fig. 6D-G). Thus, we speculate that KDM1A depletion would still further sensitise ZMYM3-deficient cells to DAC, as other redundant proteins may exist to help stabilize KDM1A's chromatin binding. We will conduct the suggested experiments and include the results in our revised manuscript.

6. To rule out cell cycle effects on DAC-induced DNA damage/repair, it would be helpful to see cell cycle distribution profiles of the relevant KO cell lines tested, especially those for

ZMYM3 and KDM1A.

We appreciate the reviewer's suggestion, and we will perform these experiments.

7. Line 449-452/Fig 6B. This experiment should be repeated using γ H2AX immunostaining rather than RPA. This would avoid the need for pre-extraction, and retain visible GFP-KDM1A dSWIRM/dTOWER mutants within nuclei and make the results more intuitive to readers.

We appreciate the reviewer's suggestion. Due to the loss of signals of KDM1A without SWIRM or TOWER domain after pre-extraction, we did the live cell imaging after laser micro-irradiation. This result (Figure 6C) confirmed that KDM1A without SWIRM or TOWER domain shows reduced enrichment at DNA damage sites, compared to WT KDM1A. However, to avoid potential misunderstanding, we will include immunofluorescence results without pre-extraction and taking γ H2A.X immunostaining in our revised manuscript.

8. Is KDM1A increased on chromatin after DAC treatment? The authors could assess GFP-KDM1A on chromatin using pre-extraction and imaging after DAC.

We appreciate the reviewer's suggestion. Reviewer 1 raised a related concern about DAC-induced KDM1A chromatin binding. We will conduct quantitative immunofluorescence experiments to investigate DAC-induced KDM1A chromatin binding and to verify if the recruitment is dependent on the SWIRM and TOWER domains under DAC treatment.

9. Is KDM1A recruitment to micro-irradiation sites or chromatin after DAC SUMO-dependent? This could be through either DNMT1-DPC SUMOylation or in response to DSBs. The authors could use inhibitors of SUMOylation machinery (e.g. TAK-981 or ML-792) to look at SUMO-dependent recruitment.

We appreciate the reviewer's suggestion, and we will perform these experiments.

10. Line 489-494: I disagree with this line of reasoning. Given that FA patients experience

severe systemic sensitivity to ICL-inducing reagents, there is no reason to expect something different for DAC. The Fanconi pathway is only active in S/G2 in proliferating cells, in the same way that DAC only induces DNMT1-DNA adducts in S-phase in proliferating cells. Such statements should be made with much more caution and should at most propose a rationale for assessment in mouse models.

We thank the reviewer for raising this issue, which is also highlighted by Reviewer 1. Given the potential for high DAC toxicity in the bone marrow due to active cell proliferation, as suggested by the reviewers, we will remove this statement from the discussion to avoid any misinterpretation.

12. Fig 5A: only 2 replicates. This is an interesting phenotype so it would be nice to see another replicate.

We appreciate the reviewer's suggestion, and we will add replicates.

13. It would be of broad interest to assess the some of the core DAC sensitivity phenotypes with Azacytidine as well, since these compounds are considered interchangeable clinically but this might not be the case.

We appreciate the reviewer's suggestion regarding the assessment of core DAC sensitivity phenotypes with Azacytidine (AZA). Previous studies, such as Hollenbach et al. (PLoS One, 2010), have highlighted differences between DAC and AZA. While its incorporation into RNA induces another layer of activity, AZA shares a similar mechanism with DAC on DNA. Consistent with this, recent two studies focusing on AZA screening have found that cells deficient in TOPORS, also identified in two other DAC screenings, is more sensitive to AZA (Truong et al., Nat Commun. 2024; Kaito et al., Nat Commun. 2024; Carnie et al., EMBO J. 2024; Liu et al., Nat Struct Mol Biol. 2024).

14. Several important studies have not been cited. These include:

- a. Orta et al Nucleic Acids Research 2013 (description of the Fanconi pathway's relevance to DAC tolerance).
- b. Borgermann et al EMBO J 2019 (SUMOylation and proteasomal degradation of

DNMT1-DPCs after DAC).

c. Liu et al EMBO J 2021 (SUMO-targeted ubiquitylation of DNMT1-DPCs by RNF4).

d. Weickert et al Nat Comms 2023 (DNMT1-DPC degradation is proteasome-dependent but also contributed to by the DPC protease SPRTN).

We appreciate the reviewer's suggestion, and we will cite these important studies during revision.

Dear Dr. Xiong,

Thank you for the transfer of your peer-reviewed manuscript to EMBO reports, and for your proposed revision plan. I think your plan for how to revise your ms is good, and I would like to invite you to revise your ms along the lines you suggest.

It will be important that stronger data on (1) the recruitment of KDM1A to γ H2A.X-marked DNA damage sites under DAC treatment and (2) the impact of KDM1A depletion on DNA repair will be provided.

I would thus like to invite you to revise your manuscript with the understanding that the referee concerns must be fully addressed and their suggestions taken on board. Please address all referee concerns in a complete point-by-point response. Acceptance of the manuscript will depend on a positive outcome of a second round of review. It is EMBO reports policy to allow a single round of major revision only and acceptance or rejection of the manuscript will therefore depend on the completeness of your responses included in the next, final version of the manuscript.

We realize that it is difficult to revise to a specific deadline. In the interest of protecting the conceptual advance provided by the work, we recommend a revision within 3 months (2nd Jan 2025). Please discuss the revision progress ahead of this time with the editor if you require more time to complete the revisions.

- 1) A data availability section providing access to data deposited in public databases is missing. If you have not deposited any data, please add a sentence to the data availability section that explains that.
- 2) Your manuscript contains statistics and error bars based on $n=2$. Please use scatter blots in these cases. No statistics should be calculated if $n=2$.

5) a complete author checklist, which you can download from our author guidelines <https://www.embopress.org/page/journal/14693178/authorguide>. Please insert information in the checklist that is also reflected in the manuscript. The completed author checklist will also be part of the RPF.

6) Please note that all corresponding authors are required to supply an ORCID ID for their name upon submission of a revised manuscript (<https://orcid.org/>). Please find instructions on how to link your ORCID ID to your account in our manuscript tracking system in our Author guidelines <https://www.embopress.org/page/journal/14693178/authorguide#authorshipguidelines>

7) Before submitting your revision, primary datasets produced in this study need to be deposited in an appropriate public database (see <https://www.embopress.org/page/journal/14693178/authorguide#datadeposition>). Please remember to provide a

reviewer password if the datasets are not yet public. The accession numbers and database should be listed in a formal "Data Availability" section placed after Materials & Method (see also <https://www.embopress.org/page/journal/14693178/authorguide#datadeposition>). Please note that the Data Availability Section is restricted to new primary data that are part of this study. * Note - All links should resolve to a page where the data can be accessed. *

10) Regarding data quantification (see Figure Legends:

<https://www.embopress.org/page/journal/14693178/authorguide#figureformat>)

- the name of the statistical test used to generate error bars and P values,
- the number (n) of independent experiments (please specify technical or biological replicates) underlying each data point,
- the nature of the bars and error bars (s.d., s.e.m.),
- If the data are obtained from n Program fragment delivered error `Can't locate object method "less" via package "than" (perhaps you forgot to load "than"?) at //ejpvfs23/sites23b/embor_www/letters/embor_decision_revise_and_review.txt line 56.' 2, use scatter blots showing the individual data points.

12) All Materials and Methods need to be described in the main text using our 'Structured Methods' format, which is required for all research articles. According to this format, the Methods section includes a Reagents and Tools Table (listing key reagents, experimental models, software and relevant equipment and including their sources and relevant identifiers) and a Methods and Protocols section describing the methods using a step-by-step protocol format. The aim is to facilitate adoption of the methodologies across labs. More information on how to adhere to this format as well as a downloadable template (.docx) for the Reagents and Tools Table can be found in our author guidelines:

An example of a Method paper with Structured Methods can be found here: <https://www.embopress.org/doi/full/10.1038/s44320-024-00037-6#sec-4>

You are able to opt out of this by letting the editorial office know (emboreports@embo.org). If you do opt out, the Review Process File link will point to the following statement: "No Review Process File is available with this article, as the authors have

chosen not to make the review process public in this case."

I look forward to seeing a revised form of your manuscript when it is ready.

Yours sincerely,

Response to reviewers' comments:

Referee #1:

In this manuscript, Zhang et al. explore genetic determinants of decitabine (DAC) sensitivity using CRISPR-Cas9 screens. In the first part of the manuscript, the authors validate some of the screening hits with individual guide RNAs and provide additional supporting evidence. These genes include nucleotide metabolism genes (DCTD), Fanconi anemia genes (FANCA, FANCD2, FANCL and UBE2T), CTDNEP1, and CNEP1R1, which they demonstrate influence DNA repair, as well as genes that affect DNMT1-DNA adduct levels (DNMT1 itself and UHRF1). In the latter half, they focus on one of the epigenetic gene hits, KDM1A. The authors find that KDM1A's ability to protect cells from DAC is independent of its demethylase activity but dependent on the SWIRM and TOWER domains. They show that these domains are essential for interaction with ZMYM3, a protein that localizes at DNA damage sites and facilitates homologous recombination (HR), and for KDM1A chromatin loading. However, the study shows that KDM1A KO does not impair DNA repair, yet the authors still conclude that KDM1A promotes cellular survival in response to DAC through its involvement in the DNA repair process.

As it stands, the manuscript seems more like a verification of screening results than an exploration of new mechanisms for DAC actions. The section on KDM1A begins to address this point but ultimately leaves the role of KDM1A at DNA damage sites unclear.

A similar CRISPR screening study on DAC by Carnie et al., (2024) was recently published in EMBO J. and reported novel findings of the DCTD-mediated DAC cytotoxicity and the new DNA-protein crosslink repair pathway involving TOPORS. Unfortunately, this manuscript does not have the same level of novelty or impact. Below are my comments to help the authors enhance the manuscript.

We thank the reviewer for the comments and feedback, which have helped us to improve this work.

Major points:

1. Fig. 2C-E (and other similar figures): How p-values were calculated need to be explained in the Method section. The figure legends states that "the p values were determined using nonlinear regression", but nonlinear regression is about fitting curves to the data, not for hypothesis testing (whether the KO cells have different DAC sensitivity compared to the control). The authors could use t-test on IC50 values in control vs. KO #1 and control vs. KO #2 separately or choose more complex methods.

We appreciate the reviewer's suggestion for clarification regarding the statistical methods. In our analysis, we employed nonlinear regression for fitting the dose-response curves, and the extra sum-of-squares F test was used for hypothesis testing. This test evaluates whether a single model can adequately fit the data for both control and knockout (KO) groups or if separate models with different IC50 values are necessary. Additionally, we found that similar results could be obtained using a two-way ANOVA with genotype as the main effect. In our revised manuscript, we have included a detailed description of the statistical analysis methods in the figure legends accordingly to accurately reflect the methods used.

2. Fig. 4H: Whether the SWIRM and TOWER domains are necessary for chromatin loading of KDM4A should be addressed in the context of DAC treatment, as the mechanism of KDM1A loading in the absence of DAC treatments shown here could be different from DAC-induced chromatin loading. More studies are needed for DAC-induced chromatin loading of KDM4A. For example, do KDM4 and ZMYM3 colocalize with DAC-induced γ H2AX foci? Does it require the SWIRM and TOWER domains? (See also Point #4.)

We appreciate the reviewer's insightful comments regarding the chromatin loading mechanism of KDM1A in the context of DAC treatment. As demonstrated by Liu et al. (EMBO J, 2021), DAC does not elicit a robust DNA damage response immediately following incorporation or DNMT-DNA adduct degradation. To investigate the kinetics of DAC-induced γ H2A.X signals, as also suggested by Reviewer 2, we monitored the levels of DNMT1-DNA adducts and γ H2A.X signals in synchronized U2OS cells subjected to a

30-minute DAC pulse during early S phase. Surprisingly, despite the rapid trapping and subsequent degradation of DNMT1, as previously reported (Liu et al., EMBO J, 2021; new Fig. 4A), we did not observe a concomitant increase in γ H2A.X. Instead, a pronounced increase in γ H2A.X signal was detected 24 hours post-DAC pulse (new Fig. 4B), when most cells had entered the second S phase. These findings suggest a temporal separation between DNMT1-DPC (DNA-protein crosslink) degradation and the induction of extensive DNA damage by DAC, revealing a previously unrecognized trans-cell cycle effect. Further analyses confirmed that DNA replication in the subsequent S phase is essential for the full induction of this robust DNA damage response (new Fig. 4C-F).

Therefore, to examine the recruitment of KDM1A to DAC-induced DNA damage sites, we employed proximity ligation assay (PLA) to visualize the colocalization of KDM1A and γ H2A.X 24 hours after DAC pulse, corresponding to the second S phase. We detected KDM1A/ γ H2A.X colocalization in S-phase cells from the DMSO-treated group (new Fig. 7D). Consistent with our previous findings, ablation of either the SWIRM or TOWER domain abolished this colocalization in the DMSO group (new Fig. 7D). The number of KDM1A/ γ H2A.X colocalization foci was significantly increased in cells pulsed with DAC (DAC group; new Fig. 7D), indicating that DAC-induced DNA damage promotes KDM1A recruitment to these sites. Importantly, ablation of either the SWIRM or TOWER domain also abolished the observed colocalization in the DAC-treated group (new Fig. 7D). These results demonstrate that both the SWIRM and TOWER domains are essential for KDM1A recruitment and binding to DAC-induced DNA damage sites.

In response to the suggestion in Major Point #4, we investigated the dependence of KDM1A recruitment on ZMYM3 in the context of DAC treatment using PLA as well. Our results demonstrated that loss of *ZMYM3* reduced, but did not completely abolish, the colocalization of KDM1A with γ H2A.X in DAC-pulsed cells (new Fig. 7I), mirroring its impact on KDM1A recruitment to micro-irradiation-induced DNA damage sites (new Fig. 7H). These findings suggest that ZMYM3 plays a partial role in KDM1A localization at DNA damage sites and further indicate the existence of redundant mechanisms for KDM1A recruitment.

Using PLA, we also detected colocalization of ZMYM3 with γ H2A.X in both DAC-pulsed

and control cells in S phase (new Fig. EV5G). To determine whether KDM1A facilitates the recruitment of ZMYM3 to DNA damage sites, we expressed a ZMYM3 mutant lacking zinc fingers 8 and 9 (ZMYM3 Δ ZNF8&9), which has been shown to mediate its interaction with KDM1A (Shapson-Coe et al, PLoS One, 2019). We observed efficient recruitment of both WT and Δ ZNF8&9 mutants to DNA damage sites induced by micro-irradiation (new Fig. EV5E,F) and DAC (new Fig. EV5G). These findings suggest that ZMYM3 operates upstream of KDM1A at DNA damage sites to promote KDM1A recruitment.

New Fig. 7D

New Fig. 7I

New Fig. EV5G

New Fig. EV5E

New Fig. EV5F

New Fig. 7. (D) Representative images (upper) of PLA results between EGFP and γ H2A.X in U2OS cells expressing either EGFP-tagged KDM1A-WT, KDM1A- Δ SWIRM, or KDM1A- Δ TOWER 24 hours after released from single-round thymidine synchronization with or without a 30-minute DAC pulse. Quantification of PLA foci number (lower) in these cells. Each point represents the mean of one experiment (N = 2 experiments, n > 6000 cells for each condition). The p values were determined using one-way ANOVA followed by Sidak's multiple comparisons test. The scale bar is 10 μ m. (I) Representative images (upper) of PLA results between EGFP and γ H2A.X in EGFP-KDM1A-expressing U2OS cells transduced with either sgCTRL or two different sgRNAs targeting ZMYM3 (sgZMYM3 #1 and #2) 24 hours after released from single-round thymidine synchronization with or without a 30-minute DAC pulse. Quantification of PLA foci number (lower) in these cells. Each point represents the mean of one experiment (N = 2 experiments, n > 5000 cells for each condition). The p values were determined using one-way ANOVA followed by Sidak's multiple comparisons test.

New Fig. EV5. (E) Representative images of immunostaining for EGFP-ZMYM3 and the DSB marker

RPA32 in U2OS cells expressing either EGFP-tagged ZMYM3-WT, ZMYM3- Δ ZNF8&9 after laser-induced DNA damage. (F) Quantification for profiles of EGFP-tagged ZMYM3 and RPA32 immunostaining signals along the lines across damage stripes in (E). The relative signal intensities were normalized to the mean of the U2OS cells expressing EGFP-ZMYM3 WT. Data represent mean \pm SEM (n = 24 cells for ZMYM3-WT group; n = 26 cells for ZMYM3- Δ ZNF8&9 group). (G) Representative images (upper) of PLA results between EGFP and γ H2A.X in U2OS cells expressing either EGFP-tagged ZMYM3-WT and ZMYM3- Δ ZNF8&9 24 hours after released from single-round thymidine synchronization with or without a 30-minute DAC pulse. Quantification of PLA foci number (lower) in these cells. Each point represents the mean of one experiment (N = 2 experiments, n > 5000 cells for each condition). The p values were determined using one-way ANOVA followed by Sidak's multiple comparisons test.

3. Lines 387 and 436, Fig. 5D-F: Although the authors claim DNA repair deficiency in KDM4A KO cells, DNA repair appears intact in KDM4A KO cells in these reporter assays (HR efficiency is rather higher). If it is not required for the damage repair, what does KDM4A do at the DNA damage sites and how does it protect cells from DAC? It is very important to address these questions.

We appreciate the reviewer's insightful comments, which have prompted us to clarify our description of the DNA repair deficiency observed in *KDM1A* KO cells. Indeed, *KDM1A* plays a crucial role in DNA repair, as demonstrated by the increased sensitivity of *KDM1A* KO cells to both DAC (new Fig. 5A-E) and Zeocin (new Fig. 6A), as well as the delayed clearance of DNA damage, as indicated by γ H2A.X levels, following a Zeocin pulse (new Fig. 6B, C). Using DNA repair reporter assays, we have shown that while NHEJ was unaffected, HR efficiency was elevated in *KDM1A* KO cells (new Fig. 6D,E). It is important to note that reporter assay, by introducing a single DSB and measuring successfully repaired products, primarily reflect the choice between different repair pathways rather than the overall DNA repair capacity of the cell. In contrast, cell survival assays and the kinetics of γ H2A.X clearance following treatment with DNA damaging agents such as DAC and Zeocin provide a more comprehensive assessment of overall DNA repair capacity. Consistent with our findings, Mosammaparast et al. (J Cell Biol, 2013) have also reported increased sensitivity to irradiation and elevated HR rates in *KDM1A*-deficient cells.

Given that NHEJ constitutes the majority of DNA repair events in cells and HR accounts for less than 10%, a significant but modest upregulation in HR is unlikely to substantially alter NHEJ rates as measured in the NHEJ reporter system. However, in the context of

massive DNA damage induced by DAC or other DNA damaging agents, the imbalanced pathway choice observed in *KDM1A* KO cells can lead to persistent DNA damage, increased mutation rates, and ultimately, cell death. A similar effect has been documented for POLQ, whose loss is associated with increased HR rates but normal NHEJ efficiency (Yousefzadeh et al., PLOS Genetics, 2014; Ceccaldi et al., Nature, 2015). We speculate that the enhanced HR rate observed in *KDM1A* KO cells may be a consequence of increased DNA end resection in the absence of KDM1A, which normally functions to maintain a closed chromatin state at DNA damage sites. Given that DAC induces massive DSB specifically in S phase, excessive end resection in *KDM1A* KO cells may contribute to genomic instability, as suggested by Tomimatsu et al. (J Biol Chem, 2017). While elucidating the precise molecular mechanism by which KDM1A promotes DNA repair is beyond the scope of the present study, our further investigation (in response to Reviewer 2, Major Point #3) have excluded a direct impact of KDM1A on the formation and degradation of DNMT-DNA adducts and the induction of DNA damage following DAC treatment, further confirming that KDM1A modulates the cellular sensitivity to DAC primarily through its role in DNA repair. The following sentence has been incorporated into the Discussion section of the revised manuscript: The exact pathway by which the altered choice of DNA repair pathways in *KDM1A*-deficient cells contributes to cellular vulnerability following extensive DNA damage requires further investigation.

4. Fig. 6: KDM4A recruitment to DNA damage sites need to be assessed in the context of DAC treatment, not by the laser-induced DNA damage. Does KDM4 colocalize to γ H2AX foci in DAC-treated cells? (See also Pint #2.) If so, is it dependent on the SWIRM and TOWER domains? Is KDM4 recruitment dependent on ZMYM3, or vice versa?

These concerns raised here are related to Major Point #2. Please refer to our responses to all of these questions in the section addressing Major Point #2.

5. The paragraph starting from 481: Although authors propose DAC as a low side effect drug in FA patients, the presented data might rather suggest that FA patients could experience high DAC toxicity in the bone marrow, which is highly proliferative.

We thank the reviewer for highlighting this issue, which is also raised by Reviewer 2. Given the potential for high DAC toxicity in bone marrow, as noted by the reviewers, we have removed this statement from the discussion to avoid any misinterpretation.

Minor points:

1. The paragraph starting from Line 182 is speculative and would be more suitable for Discussion.

We appreciate the reviewer's concern regarding the speculative nature of this paragraph. In response to Reviewer 2's interest in this topic, we attempted to conduct experiments to verify whether proteasome inhibition affects DNA damage by interfering with the degradation of DNMT1-DNA adducts. However, as detailed in our response to Major Point #1 from Review 2, due to the nature of DAC-induced DNA damage, through a previously unrecognized trans-cell cycle effect, and the high toxicity of proteasome inhibitor during prolonged treatment, we were unable to validate the impact of proteasome inhibition on cellular sensitivity to DAC. As a result, we have moved this part to the Discussion part in our revised manuscript.

2. Primary research papers, not review articles, should be cited whenever possible. For example (but not limited to), lines 199, 200, 208, and 248.

We thank the reviewer for pointing this out. Following the reviewer's suggestion, we have replaced review article citations with primary research papers throughout the manuscript, with particular attention to the specified lines. We have ensured that citations are appropriate and relevant.

3. Fig. 3F: Needs lower DAC concentrations to demonstrate dose-responsiveness.

We appreciate the reviewer's suggestion. As requested, we conducted the experiment using lower DAC concentrations in HeLa cell, and have replaced the original figure with the updated results (new Fig. 3F).

New Fig. 3F

New Fig. 3. (F) Immunoblot analysis (left) showing the DNMT1 protein level in HeLa S3 cells transduced with empty vector (Empty) and lentiviral vectors expressing wild-type DNMT1 (WT) or catalytically inactive DNMT1-C1226A mutant with β -ACTIN as a loading control. Dose-response curves of these cells upon DAC treatment at indicated concentrations (right). Cell viability was measured by CellTiter-Glo after 3 days of DAC treatment. Experiments performed in duplicates. The p value was determined using nonlinear regression followed by the extra sum-of-squares F test.

4. Fig. 4E: Statistics is missing for clone #1

We have included the missing statistical analysis for clone #1 in our revised manuscript.

Please refer to the new Fig. 5E.

5. Fig. 4F: This domain scanning study would have been more effective with base-editing approach (for example, PMID: 35288574), as the Cas9-based approach used here truncates the protein at the guide RNA target through frameshift, essentially removing all domains C-terminal to the CRISPR targets.

We appreciate the reviewer's suggestion regarding the use of base-editing, which is indeed a more precise approach for pinpointing functional residues in a protein. Reviewer 2 also raised a related concern about the use of Cas9-based approach (Reviewer 2's Minor Point #2). However, for the specific aim of our study, the Cas9-based domain scanning approach is more suited for identifying essential regions or domains of KDM1A under DAC treatment. Following Cas9 editing, indel mutations have a 1/3 probability of being in-frame, potentially producing a protein product with a local deletion. Cells with functional in-frame mutations are likely to survive under negative selection, while those with deletions in critical regions will be sensitized. Consequently, sgRNAs targeting essential domains will be depleted during the screening. This approach is supported by the study from Shi et al. (Nat Biotechnol, 2015), which demonstrated the effectiveness of

Cas9-mediated domain scanning for identifying critical protein regions. Compared to base-editing, domain scanning allows us to more quickly and efficiently determine essential domains for the purpose of this study. We have added a description in our revised manuscript to clarify this method, as followed: Cas9-induced indels have a 1/3 probability of being in-frame, potentially producing proteins with local deletions. Cells with functional in-frame mutations are likely to survive under negative selection, while those with deletions in critical regions will be sensitized. Consequently, sgRNAs targeting essential domains will be depleted during the screening.

6. Fig. 4K: Where are ZMYM2 and ZMYM3 ranked in the initial CRISPR screens?

We thank the reviewer for the question. Indeed, ZMYM2 and ZMYM3 were not significant hits in our CRISPR screen. However, as demonstrated by our data, loss of ZMYM3, but not ZMYM2, sensitizes cells to DAC (new Fig. 5K). Furthermore, depletion of ZMYM3 reduced KDM1A recruitment to DNA damage sites induced by micro-irradiation or DAC (new Fig. 7G-I, please refer to our response to Major Point #2 for a detailed discussion). This reduction in KDM1A recruitment is partial, which likely explains why ZMYM3 did not emerge as a significant hit in the screen. We speculate that other redundant proteins may help stabilize KDM1A binding to chromatin, thereby mitigating the impact of ZMYM3 loss in the screen.

7. Fig. 6C: Needs description of statistics used to determine p-values in the Methods section.

We appreciate the reviewer's attention to this detail. Following the reviewer's suggestion, we have included the statistical methods used to determine the p-values in the corresponding figure legends. Additionally, in the Quantification and statistical analysis part of the Methods section, we have clarified that the details of the analysis are described in the figure legends of our revised manuscript.

8. Fig.6E,G: Quantitation of RPA needs to be included to show that the levels of DNA damage is comparable between samples.

We appreciate the reviewer's suggestion and have conducted the RPA quantification analysis. The result showed that the levels of DNA damage are comparable between samples (new Fig. 7F,H).

New Fig. 7

New Fig. 7. (F) Quantification for profiles of ZMYM2, EGFP-tagged KDM1A and RPA32 immunostaining signals along the lines across damage stripes in (E). The relative signal intensities were normalized to the mean of the control U2OS cells. Data represent mean \pm SEM ($n = 20$ cells for sgCTRL; $n = 30$ cells for sgZMYM2, two sgRNAs combined). (H) Quantification for profiles of ZMYM3, EGFP-tagged KDM1A and RPA32 immunostaining signals along the lines across damage stripes in (G). The relative signal intensities were normalized to the mean of the control U2OS cells. Data represent mean \pm SEM ($n = 20$ cells for sgCTRL; $n = 30$ cells for sgZMYM3, two sgRNAs combined).

9. Line 509: "(The striking) negative correlation between DNMT1 expression and DAC sensitivity ..." might be incorrect. Because high AUC means less sensitive, the negative correlation between DNMT1 expression levels and AUC values would suggest positive correlation between DNMT1 expression and DAC sensitivity, which is probably what the authors meant to say.

We thank the reviewer for pointing out this mistake. The description has been revised in the manuscript to accurately reflect the correlation between DNMT1 expression and DAC sensitivity.

Referee #2:

In the manuscript entitled "A CRISPR-Cas9 screen reveals genetic determinants of cellular response to decitabine" by Zhang & Zhang et al, the authors seek to elucidate cellular mechanisms governing sensitivity and resistance to the hypomethylating agent

decitabine, a clinically-relevant compound marketed as Dacogen (DAC). To this end, they perform an unbiased, genome-scale CRISPR/Cas9 screen in the CML-derived cell line HAP1. The authors perform follow-up investigation and validation of a range of hits arising from their screen that highlight both new and known genetic determinants of DAC tolerance. The authors follow up on one of these hits in particular - the lysine demethylase KDM1A - finding that KDM1A promotes cellular DAC tolerance. Interestingly, KDM1A's role in DAC tolerance is independent of its catalytic activity and rather appears to constitute a structural or scaffolding function that might mediate HDAC activity through its interaction with CoREST to alleviate the toxic effects of DAC-induced DNA damage.

This manuscript is timely and coincides with other recent publications that collectively are revealing new aspects of cellular responses to DAC that could inform further translationally-minded studies. The authors should be commended for trying to address multiple aspects of these responses informed by their screen. However, this approach does mean that the flow of the manuscript can feel disjointed in places, and the discussion of some of their findings is rather superficial. Despite this, the authors uncover a role for KDM1A in DAC tolerance that appears to involve its somewhat unclear role in DNA repair. While interesting, however, the function of KDM1A in this context is not explored in depth remains unclear and demonstration of the relevance of these findings in more clinically-relevant contexts is lacking.

Overall, I like several aspects of this study but feel that to be suitable for publication in The EMBO Journal, the authors would need to strengthen their manuscript substantially. However, while I have several substantial major comments, I would not expect all of them to be addressed fully - in my opinion, deeper investigation of one or two of the three major points below would improve the manuscript more than a surface-level attempt at all three. If the authors are unable to successfully address some of the issues highlighted below, then a shorter more focused version of their work could be suitable for publication in a journal such as EMBO Reports.

We thank the reviewer for the comments and feedback, which have helped us to improve

this work.

Major comments:

1. The appearance of proteasome subunits as mild resistance hits in the CRISPR screen is very surprising, particularly given that depletion of the E3 ubiquitin ligases promoting DNMT1-DPC degradation by the proteasome causes cellular sensitivity to DAC (Liu et al EMBO J 2021 [this work is relevant and should be cited], Liu et al NSMB 2024, Carnie et al EMBO J 2024). The authors should attempt to validate the effects implied by their screen, although this may not be trivial in cell survival assays. As an alternative, they could assess DNA damage induction following DAC treatment in the presence/absence of proteasome inhibition and/or Ub E1 inhibition. I would recommend the latter as it is likely better tolerated by cells.

We appreciate the reviewer's insightful observation and the relevant literature they have suggested. The involvement of proteasome subunits in DAC resistance is intriguing. However, a comprehensive understanding of this phenomenon necessitates first elucidating the mechanisms by which DAC induces DNA damage, a topic addressed in our response to Major Point #3. In our first attempt to address this, we monitored the levels of DNMT1-DNA adducts and γ H2A.X signals in synchronized U2OS cells pulsed with DAC for 30 minutes in early S phase. Surprisingly, despite rapid DNMT1 trapping and degradation in a short period (Liu et al., EMBO J, 2021; new Fig. 4A), we did not observe a corresponding increase in γ H2A.X. Instead, a pronounced increase in γ H2A.X signal was detected 24 hours post-DAC pulse (new Fig. 4B), when most cells had entered the second S phase. These findings suggest a temporal separation between DNMT1-DPC (DNA-protein crosslink) degradation and DAC-induced extensive DNA damage, revealing a previously unrecognized trans-cell cycle effect of DAC.

New Fig. 4

New Fig. 4. (A) The levels of DNMT1 proteins in U2OS cells at indicated times after released from single-round thymidine synchronization with or without a 30-minute DAC (10 μ M) pulse. Data represent the means \pm SEM ($n > 2,000$ cells for each condition). (B) The levels of γ H2A.X in U2OS cells at indicated times after released from single-round thymidine synchronization with or without a 30-minute DAC pulse. Data represent the means \pm SEM ($n > 2,000$ cells for each condition). The p values were determined using one-way ANOVA followed by Sidak's multiple comparisons test.

Based on this knowledge, we designed experiments to assess the effect of proteasome inhibition on DAC-induced DNA damage. However, due to the toxicity of prolonged MG132 treatment, which resulted in substantial cell death, we were unable to assess γ H2A.X levels over extended periods (e.g., 24 hours). Therefore, we focused on determining whether the accumulation of DNMT1-DPCs in the presence of MG132 triggers a DNA damage response or interferes with the processing of other DAC-induced DNA lesions during the early stages of repair in the first S phase. Synchronized cells were pulsed with DAC in the presence of MG132 and allowed to recover for 6 hours under continued MG132 treatment. Consistent with previous findings (Liu et al., EMBO J, 2021), we observed no significant DNA damage response in the first S phase under these conditions (as shown below, Panel A). The increase in γ H2A.X intensity observed 6 hours post-release in both DAC-treated and control cells (Panel A) likely reflects the known G2/M arrest induced by MG132. Furthermore, MG132 treatment resulted in elevated levels of chromatin-bound XRCC1 in both DAC-treated and control cells at all time points examined, indicating that the effect of MG132 on XRCC1 is independent of DAC (as shown below, Panel B). These results indicate that proteasome inhibition does not have

an immediate impact on the cellular response to DAC. However, we acknowledge that a trans-cell cycle effect of proteasome inhibition on DAC-induced DNA damage cannot be ruled out at this time. As suggested by Reviewer 1, we have relocated the proteasome part to the Discussion section of the revised manuscript.

(A) The levels of γ H2A.X in U2OS cells at indicated times after released from single-round thymidine synchronization in the presence or absence of MG132 with or without a 30-minute DAC pulse. (B) The levels of chromatin-bound XRCC1 in U2OS cells at indicated times after released from single-round thymidine synchronization in the presence or absence of MG132 with or without a 30-minute DAC pulse.

2. CTDNEP1's role is speculatively linked to DSB repair, but its connection to DAC toxicity isn't established as a direct link. I suggest looking at repair rates of DAC-induced DSBs in sgCtrl/sgCTDNEP1 cells (e.g. by γ H2AX foci induction/disappearance kinetics). IP/MS or phospho-proteomic approaches to try to relevant interactors/substrates of CTDNEP1 could be valuable here.

We thank the reviewer for raising a valuable point regarding the role of CTDNEP1 in DAC toxicity and DNA repair, which aligns with our own research interests. We are currently pursuing a project on this topic and anticipate gaining further insights in the near future. We recognize the importance of establishing a direct link between CTDNEP1 and these processes and will aim to explore this further as our research progresses.

3. For me, the nature of DNA lesion for which KDM1A is relevant is an important question to address. The authors assess DSB induction and repair kinetics using γ H2AX immunofluorescence, but the origin of DSBs after DAC is unclear and could feasibly relate to proteasome activity of SUMOylated DPCs. As I see it, the likely origin of DSBs is

downstream of DNMT1-DPC degradation and potentially also downstream of SSB induction upon base excision repair of unmethylated, genome-embedded 5-aza-dC/5-aza-dU. Proximal SSBs could lead to DSB formation. This begs the question: at which stage does KDM1A act? From the authors' data, it appears to be downstream of DNMT1-DPC degradation, but it's hard to say where it acts. I suggest attempting the following experiments from which relevant insights might be gained:

We appreciate the reviewer's insightful question regarding the origin of DAC-induced DSBs, which has substantially enhanced our understanding of DAC's mechanism of action and improved the manuscript. By monitoring γ H2A.X in synchronized cells following DAC pulse, we uncovered a previously unrecognized trans-cell cycle effect (please refer to our response to Major Point #1; new Fig. 4). To determine if subsequent DNA replication is required for this effect, we labeled S-phase cells with EdU prior to sample collection. We observed elevated γ H2A.X only in cells that had progressed into the second S phase and were actively synthesizing DNA (EdU-positive; new Fig. 4C,D). Furthermore, pharmacological inhibition of CDK4/6 with palbociclib, which prevents entry into the subsequent S phase, abolished the DAC-induced upregulation of γ H2A.X (new Fig. 4E). Consistent results were obtained using a second thymidine block to arrest cells at the G1/S boundary; only DAC-pulsed cells released into the second S phase exhibited significantly elevated γ H2A.X levels compared to untreated controls, whereas cells maintained at the G1/S boundary showed comparable γ H2A.X signals (new Fig. 4F). Collectively, these findings demonstrate that DAC does not directly induce extensive DNA damage but requires subsequent rounds of DNA replication to exert its cytotoxic effects. Both residual DNMT-DPCs and persistent SSBs can induce replication stress. Consistent with the findings of Carnie et al. (EMBO J, 2024), we showed that the incorporation of 5-aza-dU via the activity of DCTD may contribute to DAC-induced SSBs, and that DCTD deficiency confers resistance to DAC. To investigate whether the SSB repair pathway is engaged in the early processing of DAC-induced DNA damage, we monitored the levels of PARylation and chromatin-bound XRCC1 in synchronized cells following DAC pulse. Treatment with H₂O₂, used as a positive control for SSB induction, resulted in a robust increase in both PARylation and chromatin-bound XRCC1 levels. In contrast, the DAC

pulse did not elicit a significant change in either marker (new Fig. 4G,H). These results indicate that DAC incorporation does not immediately elicit a robust SSB response. However, we acknowledge that a more subtle contribution of SSB repair to the mitigation of DAC-induced DNA damage remains a possibility.

New Fig. 4

New Fig. 4. (C) EdU levels in U2OS cells at the indicated time points after release from single-round thymidine synchronization, with or without a 30-minute DAC (10 μ M) pulse. Prior to sample collection, cells were pulsed with 10 μ M EdU for 1 hour to label cells undergoing DNA replication. Data represent

the means \pm SEM ($n > 2,500$ cells for each condition). The p values were determined using one-way ANOVA followed by Sidak's multiple comparisons test. (D) γ H2A.X in U2OS cells at the indicated time points after release from single-round thymidine synchronization, with or without a 30-minute DAC (10 μ M) pulse. Prior to sample collection, cells were pulsed with 10 μ M EdU for 1 hour to label cells undergoing DNA replication. Data represent the means \pm SEM ($n > 500$ cells for each condition). The p values were determined using one-way ANOVA followed by Sidak's multiple comparisons test. (E) γ H2A.X in U2OS cells 24 hours after release from single-round thymidine synchronization, with or without a 30-minute DAC (10 μ M) pulse. Palbociclib (1 μ M) was added at 9 hours after thymidine release to prevent progression into the next S phase. Data represent the means \pm SEM ($n > 4,000$ cells for each condition). The p values were determined using one-way ANOVA followed by Sidak's multiple comparisons test. (F) γ H2A.X in U2OS cells 25 hours after release from the first round of thymidine synchronization, with or without a 30-minute DAC (10 μ M) pulse. A second round of thymidine block was applied 9 hours after the first release to arrest cells at the G1/S boundary. Prior to sample collection, cells were released into the subsequent S phase for 1 hour (25 hours after the initial thymidine release). Data represent the means \pm SEM ($n > 1,500$ cells for each condition). The p values were determined using one-way ANOVA followed by Sidak's multiple comparisons test. (G) PARylation levels in U2OS cells at the indicated time points after release from single-round thymidine synchronization, with or without a 30-minute DAC (10 μ M) pulse. A 30-minute H₂O₂ (100 μ M) treatment was used as a positive control. Data represent the means \pm SEM ($n > 1,900$ cells for each condition). (H) Chromatin-bound XRCC1 levels in U2OS cells at the indicated time points after release from single-round thymidine synchronization, with or without a 30-minute DAC (10 μ M) pulse. A 30-minute H₂O₂ (100 μ M) treatment was used as a positive control. Data represent the means \pm SEM ($n > 1,900$ cells for each condition).

These findings prompted us to hypothesize that incomplete resolution of residual DNMT-DNA adducts prior to the subsequent S phase could impede replisome progression during DNA replication. This hypothesis is substantiated by the observed reduction in EdU incorporation in DAC-pulsed cells during the second round of DNA replication (new Fig. 4C). Moreover, 5-aza-dU, generated from 5-aza-dC by DCTD, constitutes an additional source of replication stress. In support of this, we observed diminished γ H2A.X levels in *DCTD*-depleted cells (new Fig. EV1B). These new findings and a discussion of the trans-cell cycle effect of DAC on DNA damage induction have been incorporated into the revised manuscript.

New Fig. EV1 B

New Fig. EV1. (B) Quantification of γ H2A.X immunostaining signal intensity in control and DCTD-depleted HAP1 cells after DAC treatment treated with DMSO or DAC (500 nM) for 24 hours. Each point represents the mean of one experiment (N = 2 experiments; n > 3,700 cells for each genotype and treatment). A.U., arbitrary unit.

a. Repeating the γ H2AX kinetic experiment in Fig 5C with cell cycle synchronisation and release into S-phase in the presence or absence of DAC for a short time. This will allow the authors to assess whether DSB induction is altered in *KDM1A* KO cells and track the repair kinetics.

We appreciate the reviewer's suggestion. To investigate the dynamics of γ H2A.X signals following DAC pulse in synchronized cells, we initially employed a thymidine block (2 mM, 18-20 hours) to arrest HAP1 WT and *KDM1A* KO clones at the G1/S boundary, followed by a 30-minute DAC pulse and subsequent analysis of γ H2A.X levels. However, we observed that thymidine treatment itself induced significant replication stress, which was further exacerbated in *KDM1A* KO cells, likely due to their compromised DNA repair capacity (as shown below, Panel A and B). As a consequence, even in the absence of DAC, γ H2A.X levels were markedly elevated in *KDM1A* KO cells compared to WT controls 24 hours after release from the thymidine block, suggesting a delayed recovery from thymidine-induced DNA damage. This observation was corroborated by increased cell death in *KDM1A* KO cells following 3 days of thymidine treatment (as shown below, Panel C). While the trans-cell cycle effect of DAC on γ H2A.X induction was still apparent (with significantly elevated γ H2A.X levels observed in both WT and *KDM1A* KO cells 24 hours post-DAC pulse, and a further increase observed in *KDM1A* KO cells at 30 hours), the elevated basal γ H2A.X levels resulting from the thymidine block precluded a definitive conclusion regarding the impact of *KDM1A* deletion on DAC-induced DNA damage.

Therefore, we explored alternative cell cycle synchronization strategies. Although mimosine has been reported to induce G1 arrest without significant DNA damage, we found that, under our experimental conditions, mimosine treatment at its established working concentration induced substantial DNA damage and cell death in both HAP1 WT and *KDM1A* KO cells (as shown below, Panel D).

(A) The levels of γ H2A.X in HAP1 WT and *KDM1A* KO cells at indicated times after released from single-round thymidine synchronization with or without a 30-minute DAC pulse. Data represent the means \pm SEM ($n > 2,300$ cells for each condition). (B) The levels of γ H2A.X in HAP1 WT and *KDM1A* KO cells at indicated timepoints after released from single-round thymidine synchronization with or without a 30-minute DAC (10 μ M) pulse. Data represent the means \pm SEM ($n = 2$). The p values were determined using two-way ANOVA followed by Sidak's multiple comparisons test. (C) Survival analysis of WT and *KDM1A* KO cells after treatment with thymidine at indicated concentrations for 3 days. Cell viability was measured by CellTiter-Glo. Each point represents one experiment ($n = 2$ replicates). (D) The levels of γ H2A.X in HAP1 cells treated with 400 μ M mimosine for 12 hours ($n > 7,000$ cells for each condition).

Our further analysis revealed that *KDM1A* deletion does not influence the cell cycle progression (please refer to our response to Minor Point #6; new Fig. EV4A,B), DNMT1-DNA adducts formation or degradation (new Fig. EV4E), or other early DNA repair processing intermediates (please refer to our responses within this major point; new Fig. EV4F,G). Our previous investigation using Zeocin as an alternative DNA damaging agent demonstrated a delayed clearance of γ H2A.X signals in *KDM1A* KO cells compared to WT controls (new Fig. 6B). Considering these cumulative findings, we propose that *KDM1A* primarily participates in the DNA repair process during subsequent S phases to mitigate DAC-induced DNA damage. This discussion has been incorporated into the Discussion section in the revised manuscript.

b. The use of a more sensitive readout of DNMT1-DPC induction/repair than bulk DNMT1 degradation. The recently-developed PxP assay (Weickert et Nat Comms al 2023) is one candidate, but the authors could also address this using less specific assays such as chromatin fractionation after DAC treatment/release.

As suggested, we tracked DNMT1-DNA adduct levels after DAC pulse at the indicated time points using quantitative immunofluorescence with pre-extraction. The results showed that DNMT1 was rapidly crosslinked to DNA upon DAC incorporation and degraded within 6 hours through DPC repair (new Figs. 4A and EV4E). There is no obvious difference between WT and *KDM1A* KO clones (new Fig. EV4E), indicating that *KDM1A* does not affect DNMT1-DPC induction and degradation.

New Fig. EV4. (E) The levels of DNMT1 proteins in HAP1 WT and *KDM1A* KO cells at indicated times after released from single-round thymidine synchronization with or without a 30-minute DAC (10 μ M) pulse. Data represent the means \pm SEM ($n = 3$). The p values were determined using one-way ANOVA followed by Sidak's multiple comparisons test.

c. Assessment of SSB formation/repair in *KDM1A* KO cells after DAC treatment by alkaline comet assay.

We appreciate the reviewer's insightful suggestion regarding the nature of DNA lesions following DAC treatment. However, single- and double-strand breaks are both prevalent in DAC treated cells, and while alkaline comet assay is highly sensitive, it detects both types of lesions without clear differentiation. Given this limitation and the focus of our current study, we conducted quantitative immunofluorescence to monitor PARylation and chromatin-bound XRCC1 as markers of SSB in synchronized WT and *KDM1A* KO cells

post-DAC pulse. The results showed comparable levels of PARylation and XRCC1 between WT and *KDM1A* KO cells at early repair stages (new Fig. EV4F,G), suggesting that *KDM1A* loss does not affect potential early repair intermediates.

New Fig. EV4

New Fig. EV4. (F) The levels of PARylation in HAP1 WT and *KDM1A* KO cells at indicated times after released from single-round thymidine synchronization with or without a 30-minute DAC (10 μ M) pulse. Data represent the means \pm SEM ($n = 3$). (G) The levels of chromatin-bound XRCC1 in HAP1 WT and *KDM1A* KO cells at indicated times after released from single-round thymidine synchronization with or without a 30-minute DAC (10 μ M) pulse. Data represent the means \pm SEM ($n = 3$).

In summary, our analysis demonstrates that *KDM1A* deletion does not affect cell cycle progression (please refer to our response to Minor Point #6 below), DNMT1-DNA adducts formation or degradation, or the levels of potential intermediates during early response. Notably, *KDM1A* is recruited to DAC-induced DNA damage sites marked by γ H2A.X (new Fig. 7D, please refer to our response to Reviewer 1's Major Point #2). Furthermore, *KDM1A* depletion impairs DAC-induced DNA repair, indicating *KDM1A* confers cellular resistance to DAC by promoting DNA repair processes in subsequent S phases.

Minor comments:

1. The use of several timepoints in the DAC CRISPR screen is a thorough approach. However, the use of MAGeCK to analyse the screen outputs seems inferior to Chronos (Dempster et al, Genome Biology 2021), an analysis pipeline specifically designed for consolidating multiple time points. I recommend re-analysing the screens using Chronos, as this could make Figure 1 more intuitive to readers.

We appreciate the reviewer's suggestion and we have performed this analysis using

Chronos. Genes with effect scores greater than three standard deviations above or below the mean were considered as positive and negative hits respectively (as shown below, Panel A). A substantial overlap was observed between Chronos and MAGeCK hits (as shown below, Panels B and C), providing further support for our screen. As MAGeCK analysis incorporate kinetic information, we prefer to present those results.

(A) Ranked genes from the CRISPR screens with multiple timepoints analyzed with Chronos. Genes are ranked by the effect scores. Negatively and positively selected genes are labeled with blue and red, respectively. (B) Venn diagram showing the overlap between positive hits from MAGeCK's and Chrono's results. (C) Venn diagram showing the overlap between negative hits from MAGeCK's and Chrono's results.

2. I am struggling to understand the rationale behind the KDM1A domain scanning approach. If I have interpreted it correctly, Cas9 will just induce DSBs across the gene, but given that frameshifting mutations and nonsense-mediated decay will cause many knockouts that affect more domains than just the one targeted by a particular sgRNA. If I have misinterpreted the technique, the description of it should be reworded.

We appreciate the reviewer for raising this concern. Reviewer 1 also raised a related concern about this method (Minor Point #5 from Reviewer 1). Cas9-based domain scanning is well-suited to our aim of identifying essential regions or domains of KDM1A

under DAC treatment. This approach leverages the fact that Cas9-induced indels have a 1/3 probability of being in-frame, potentially producing proteins with local deletions. While cells with functional in-frame mutations are likely to survive under negative selection, those with deletions in critical regions will be sensitized, leading to depletion of targeting sgRNAs. Conversely, frameshift mutations, induced by all sgRNAs, result in knockout clones that are similarly sensitive to negative selection. This approach is supported by the study from Shi et al. (Nat Biotechnol, 2015), which demonstrated the effectiveness of domain scanning for identifying critical protein regions. We have added a description in our revised manuscript to clarify this method, as followed: Cas9-induced indels have a 1/3 probability of being in-frame, potentially producing proteins with local deletions. Cells with functional in-frame mutations are likely to survive under negative selection, while those with deletions in critical regions will be sensitized. Consequently, sgRNAs targeting essential domains will be depleted during the screening.

3. Line 247: methylation by DNMT1 occurs during S-phase, not mitosis.

We thank the reviewer for identifying this error. This has been corrected in our revised manuscript.

4. The innate resistance of HeLa S3 cells is quite striking - is there anything available in publica datasets that could explain this? Do HeLa cells bear a DCTD mutation, for example?

We appreciate the reviewer's insightful comment regarding HeLa cell's innate resistance to DAC, which has been previously reported by Palii et al. in their 2008 publication in Molecular and Cellular Biology. As suggested, we performed a cross-analysis of the mutation profile in HeLa cells (focusing on homozygous deletions) with our positive selection hits. However, there was no overlap between the two sets. Given that overexpression of wild-type, but not catalytic inactive, DNMT1 sensitizes HeLa cells to DAC (new Fig. 3F), we compared DNMT1 protein levels between HeLa and HAP1 cells. We observed substantially higher DNMT1 levels in HAP1 cells (as shown below), potentially explaining the differential DAC response within our experimental timeframe.

Immunoblot analysis showing DNMT1 protein levels in HeLa S3 and HAP1 cells with β -ACTIN as a loading control.

5. Are there relevant genetic interactions between KDM1A and CoREST or ZMYM3? The authors' model would predict that KDM1A depletion from CoREST or ZMYM3-deficient cells does not further sensitise cells to DAC.

We appreciate the reviewer's insightful question regarding the genetic interaction between KDM1A and CoREST or ZMYM3. While KDM1A forms stable protein complex with CoREST, ZMYM3 depletion only partially reduced KDM1A recruitment to DNA damage sites (New Fig. 7G-I). Because the interaction between KDM1A and CoREST proteins stabilizes these proteins, KDM1A protein levels are extremely low in *RCOR1/2* DKO cells (Zeng et al., Nat Commun, 2023), effectively mimicking *KDM1A* KO. Therefore, genetic depletion of KDM1A in *RCOR1/2* DKO cells is unnecessary.

To address the interaction between KDM1A and ZMYM3, following the suggestion by the reviewer, we assessed the impact of reciprocal depletion on cellular sensitivity to DAC. As the results shown, depleting KDM1A in *ZMYM3* KO cells further sensitized those cells to DAC (new Fig. EV5C), whereas depleting ZMYM3 in *KDM1A* KO cells had no effect (new Fig. EV5D). These results are consistent with our previous findings that *ZMYM3* KO leads to a milder phenotype than *KDM1A* KO (new Fig. 5B,K) and only partially impairs KDM1A recruitment to DNA damage sites induced by laser micro-irradiation (new Fig. 7G,H) or DAC (new Fig. 7I, please refer to our response to Review 1's Major Point #2), suggesting the existence of redundant KDM1A recruitment mechanisms.

To test whether ZMYM3 acts upstream of KDM1A, we expressed EGFP-tagged ZMYM3 WT and a Δ ZNF8&9 mutant (lacking zinc fingers 8 and 9, which mediate KDM1A interaction; Shapson-Coe et al., PLoS One, 2019) in U2OS cells and induced DNA damage via micro-irradiation or DAC. Both ZMYM3 WT and Δ ZNF8&9 were recruited to

DNA damage sites comparably in both contexts (new Fig. EV5E-G, please refer to our response to Review 1's Major Point #2), indicating that ZMYM3 localization to DNA damage sites does not require interaction with KDM1A.

New Fig. EV5

New Fig. EV5 (C) Dose-response curves of HAP1 WT and ZMYM3 KO cells transduced with control sgRNA (sgCTRL) or two different sgRNAs targeting KDM1A (sgKDM1A #1 and #2) upon DAC treatment at indicated concentrations. Cell viability was measured by CellTiter-Glo after 3 days of DAC treatment. Experiments performed in duplicates. The p values were determined using nonlinear regression followed by the extra sum-of-squares F test. (D) Dose-response curves of HAP1 WT and KDM1A KO cells transduced with control sgRNA (sgCTRL) or two different sgRNAs targeting ZMYM3 (sg ZMYM3 #1 and #2) upon DAC treatment at indicated concentrations. Cell viability was measured by CellTiter-Glo after 3 days of DAC treatment. Experiments performed in duplicates. The p values were determined using nonlinear regression followed by the extra sum-of-squares F test. (E) Representative images of immunostaining for EGFP-ZMYM3 and the DSB marker RPA32 in U2OS cells expressing either EGFP-tagged ZMYM3-WT, ZMYM3- $\Delta\text{ZNF8\&9}$ after laser-induced DNA damage. (F) Quantification for profiles of EGFP-tagged ZMYM3 and RPA32 immunostaining signals along the lines across damage stripes in (E). The relative signal intensities were normalized to the mean of the U2OS cells expressing EGFP-ZMYM3 WT. Data represent mean \pm SEM ($n = 24$ cells for ZMYM3-WT group; $n = 26$ cells for ZMYM3- $\Delta\text{ZNF8\&9}$ group). (G) Representative images (upper) of PLA results between EGFP and $\gamma\text{H2A.X}$ in U2OS cells expressing either EGFP-tagged ZMYM3-WT and ZMYM3- $\Delta\text{ZNF8\&9}$ 24 hours after released from single-round thymidine synchronization with or without a 30-minute DAC pulse. Quantification of PLA foci number (lower) in these cells. Each point represents the mean of one

experiment (N = 2 experiments, n > 5000 cells for each condition). The p values were determined using one-way ANOVA followed by Sidak's multiple comparisons test.

6. To rule out cell cycle effects on DAC-induced DNA damage/repair, it would be helpful to see cell cycle distribution profiles of the relevant KO cell lines tested, especially those for ZMYM3 and KDM1A.

Following the reviewer's suggestion, we performed cell cycle analysis in HAP1 WT, *KDM1A* KO, *ZMYM2* KO, and *ZMYM3* KO clones, by PI staining and 30-minute EdU labeling. The percentages of cells in S phase were not substantially differed between the KO clones and WT cells.

New Fig. EV4

New Fig. EV4. (A) Percentages of cells in S phase in WT and *KDM1A* KO clones. Data represent the means \pm SEM (n = 3). The p values were determined using one-way ANOVA followed by Sidak's multiple comparisons test. (B) Percentages of cells in S phase in WT, *ZMYM2* KO, and *ZMYM3* KO clones. Data represent the means \pm SEM (n = 3). The p values were determined using one-way ANOVA followed by Sidak's multiple comparisons test.

7. Line 449-452/Fig 6B. This experiment should be repeated using gH2AX immunostaining rather than RPA. This would avoid the need for pre-extraction, and retain visible GFP-KDM1A dSWIRM/dTOWER mutants within nuclei and make the results more intuitive to readers.

We appreciate the reviewer's suggestion. We recognize that *KDM1A*'s abundance as a nuclear protein can lead to high background signal in immunostaining, potentially masking its localization at DNA damage sites without pre-extraction. Therefore, we incorporated a pre-extraction step into our protocol. However, this pre-extraction resulted in signal loss for *KDM1A* lacking the SWIRM/TOWER domain. To address this, we presented *KDM1A* protein levels in these cells (new Fig.7A) and performed live-cell imaging after laser micro-irradiation (new Fig. 7C). This live-cell imaging confirmed reduced recruitment of

KDM1A lacking the SWIRM or TOWER domain at DNA damage sites compared to WT KDM1A (new Fig. 7C). However, we also noticed the relatively low enrichment of KDM1A at DNA damage sites in the live-cell imaging, indicating high background signals of KDM1A in nucleus.

As requested, we performed immunofluorescence without pre-extraction, using γ H2A.X as a DNA damage marker. While these experiments also showed reduced KDM1A recruitment upon ablation of the SWIRM or TOWER domain (as shown below, Panels A and B), high background made it difficult to discern clear differences between WT and mutant KDM1A. Therefore, we prefer to present the results obtained with the pre-extraction protocol, as these more accurately reflect the differences in KDM1A recruitment to DNA damage sites.

(A) Representative images of immunostaining without pre-extraction for EGFP-KDM1A and the DSB marker γ H2A.X in U2OS cells expressing either EGFP-tagged KDM1A-WT, KDM1A- Δ NFR, KDM1A- Δ SWIRM, or KDM1A- Δ TOWER after laser-induced DNA damage. The scale bar is 10 μ m. (B) Quantification of enrichment at laser-induced damage sites for EGFP-tagged KDM1A-WT, KDM1A- Δ SWIRM, and KDM1A- Δ TOWER in U2OS cells. The relative enrichments were normalized to unirradiated regions in the same cells. Data represent the means \pm SEM ($n = 20$ for KDM1A-WT, KDM1A- Δ NFR and KDM1A- Δ TOWER cells; $n = 19$ for KDM1A- Δ SWIRM cells).

8. Is KDM1A increased on chromatin after DAC treatment? The authors could assess GFP-KDM1A on chromatin using pre-extraction and imaging after DAC.

We appreciate the reviewer's suggestion, which is related to Reviewer 1's concern regarding DAC-induced KDM1A chromatin binding. As suggested, we performed

immunostaining with pre-extraction of γ H2A.X and chromatin-bound EGFP-KDM1A in cells treated with a 30-minute DAC pulse or mock treatment after release from a single-round thymidine block. While γ H2A.X signals increased 24 hours after DAC pulse compared to controls (new Fig. 4B), no corresponding increase in chromatin-bound KDM1A was observed (as shown below). In addition, the chromatin-bound KDM1A level remained stable at the early stages following DAC incorporation. These results suggest that DAC-induced DNA damage does not globally increase chromatin associated KDM1A. However, using the proximity ligation assay, we detected increased colocalization between γ H2A.X and KDM1A 24 hours post-DAC pulse (new Fig. 7D, please refer to our response to Reviewer 1's Major Point #2), indicating KDM1A recruitment to DAC-induced DNA damage sites. There may be two possible explanations to these results: (1) KDM1A may not be recruited to all DAC-induced DNA damage sites, suggesting diversified forms of DNA lesions raised in the subsequent S phase in DAC-pulsed cells; or (2) abundant chromatin-associated KDM1A may mask recruitment to specific damage sites as discussed previously.

The levels of chromatin-bound EGFP-KDM1A in U2OS cells expressing EGFP-tagged KDM1A at indicated times after released from single-round thymidine synchronization with or without a 30-minute DAC pulse. Data represent the means \pm SEM ($n > 1,400$ cells for each condition).

9. Is KDM1A recruitment to micro-irradiation sites or chromatin after DAC SUMO-dependent? This could be through either DNMT1-DPC SUMOylation or in response to DSBs. The authors could use inhibitors of SUMOylation machinery (e.g. TAK-981 or ML-792) to look at SUMO-dependent recruitment.

We appreciate the reviewer's suggestion. The result of the iPOND experiments by Carnie et al. (EMBO J, 2024) showed that KDM1A is not enriched at DNMT1-DPC sites (highlighting KDM1A with a red rectangle in above figure adapted from their work), indicating that KDM1A does not directly participate in DNMT1-DPC formation or resolution. Therefore, the DAC-induced colocalization of KDM1A and γ H2A.X in the next S phase is likely the direct result of KDM1A recruitment to DSBs. Given ZMYM3's reported SUMO-binding activity (Guzzo et al., PLoS One, 2014) and its contribution to KDM1A recruitment as shown by our work, thus, as suggested, we investigated whether SUMOylation affects KDM1A recruitment to DSBs. We measured KDM1A enrichment at micro-irradiation-induced DNA damage sites in U2OS cells expressing EGFP-KDM1A treated with 10 μ M ML-792 or Olaparib for 1 hour before and during irradiation. The results showed that loss of SUMOylation did not influence the recruitment of KDM1A to DNA damage sites (as shown below, Panels A and B). However, Olaparib reduced KDM1A enrichment (as shown below, Panels A and B), consistent with PARP1's vital role in recruiting downstream repair factors.

(A) Representative images of immunostaining for EGFP-KDM1A and the DSB marker RPA32 in U2OS cells expressing EGFP-tagged KDM1A after laser-induced DNA damage in the presence of DMSO, ML-792 or Olaparib. The scale bar is 10 μ m. (B) Quantification for profiles of EGFP-tagged KDM1A and RPA32 immunostaining signals along the lines across damage stripes in (A). The relative signal intensities were normalized to the mean of the control U2OS cells. Data represent mean \pm SEM (n = 20 cells for each group).

10. Line 489-494: I disagree with this line of reasoning. Given that FA patients experience severe systemic sensitivity to ICL-inducing reagents, there is no reason to expect something different for DAC. The Fanconi pathway is only active in S/G2 in proliferating cells, in the same way that DAC only induces DNMT1-DNA adducts in S-phase in proliferating cells. Such statements should be made with much more caution and should at most propose a rationale for assessment in mouse models.

We thank the reviewer for raising this issue, which is also highlighted by Reviewer 1. Given the potential for high DAC toxicity in the bone marrow due to active cell proliferation, as suggested by the reviewers, we have removed this statement from the discussion to avoid any misinterpretation.

11. Line 544: it would be helpful to include a visual representation of *DCTD* mutation rather than just a link to cbiportal's home page. I am not hugely familiar with using cbiportal but couldn't find any indication that '*DCTD* homo-deletion is the most common mutation in cancer patients and cultured tumor cell lines'.

We thank the reviewer's suggestion and have added a new panel to visualize the *DCTD* mutation profile in cancer cell lines from the Cancer Cell Line Encyclopedia (CCLE) project. This analysis revealed that 14% of cancer cell lines harbor *DCTD* mutations, predominantly homozygous deletions (colored in blue; new Fig. EV1F), making our point more intuitive to readers.

New Fig. EV1F

New Fig. EV1. (F) The genetic alteration profile of *DCTD* in cancer cell lines collected by Cancer Cell Line Encyclopedia project (data was from cBioPortal). Fourteen percent of cell lines harbor mutations in *DCTD*.

12. Fig 5A: only 2 replicates. This is an interesting phenotype so it would be nice to see another replicate.

We appreciate the reviewer's suggestion. As suggested, we have added replicates to the result (new Fig. 6A), which solidifies our conclusion.

New Fig. 6A

New Fig. 6. (A) Survival analysis of WT, KDM1A KO clones, and KDM1A KO clone complemented with either empty vector, KDM1A-WT, or KDM1A-AE/KA after treatment with Zeocin at indicated concentrations for 3 days. Cell viability was measured by CellTiter-Glo. Each point represents one experiment (n = 6 replicates). The p values were determined using two-way ANOVA followed by Dunnett's multiple comparisons test.

13. It would be of broad interest to assess the some of the core DAC sensitivity phenotypes with Azacytidine as well, since these compounds are considered interchangeable clinically but this might not be the case.

We appreciate the reviewer's suggestion regarding the assessment of core DAC sensitivity phenotypes with Azacytidine (AZA). Previous studies, such as Hollenbach et al. (PLoS One, 2010), have highlighted differences between DAC and AZA. While its

incorporation into RNA induces another layer of activity, AZA shares a similar mechanism with DAC on DNA. Consistent with this, recent two studies focusing on AZA screening have found that cells deficient in TOPORS, also identified in two other DAC screenings, is more sensitive to AZA (Truong et al., Nat Commun, 2024; Kaito et al., Nat Commun, 2024; Carnie et al., EMBO J, 2024; Liu et al., Nat Struct Mol Biol, 2024).

14. Several important studies have not been cited. These include:

- a. Orta et al Nucleic Acids Research 2013 (description of the Fanconi pathway's relevance to DAC tolerance).
- b. Borgermann et al EMBO J 2019 (SUMOylation and proteasomal degradation of DNMT1-DPCs after DAC).
- c. Liu et al EMBO J 2021 (SUMO-targeted ubiquitylation of DNMT1-DPCs by RNF4).
- d. Weickert et al Nat Comms 2023 (DNMT1-DPC degradation is proteasome-dependent but also contributed to by the DPC protease SPRTN).

We appreciate the reviewer's suggestion, and we have cited these important studies in our revised manuscript.

Dear Dr. Xiong,

Thank you for the submission of your revised manuscript. We have now received the enclosed reports from the referees and I am happy to say that both support its publication now. Only a few editorial requests will need to be addressed before we can proceed with the official acceptance of your manuscript:

- In the author checklist please answer all questions on the statistics and send us a new, completed checklist.
- Please reduce the number of keywords to 5.
- Please remove the author credits from the manuscript file. All credits need to be entered during online ms submission.
- Please rename the conflict of interest subheading to "Disclosure and Competing Interest Statement"
- Please rename the EV tables to "Dataset EV1-EV3 " both in the files, the file names and in the ms callouts.
- The 2 Appendix figures can be changed into EV figures, as we allow now more than 5 EV figures. Please upload them as individual figure files and add the legends to the main ms file.
- Please remove the Reagents and Tools Table from the ms file and upload as a separate file.
- Our systematic image analysis of ms to be accepted detected a possible splice site in Figure EV4C - DNMT1. Please send us the Source Data for this figure.
- "Materials and Methods" should be renamed "Methods".
- The EV (to be Dataset) table legends need to be removed from the ms file and need to be included in the Dataset table files.

Figure legends:

1. Please note that the exact p values are not provided in the legends of figures 1H, I; 2C, D, E; 3A, B, D, F; 4B, C, D, E, F; 5A, B, C, E, G, K; 6A, C, D, F; 7C, D; EV1 A, D, E, H, I, J; EV2 D, E, G; EV3 B, I, K; EV5 C; supplementary figure(s) 1B, 2A. Please add exact p-values as reasonable.
2. Please indicate the statistical test used for data analysis in the legends of figures 1E, 3E; supplementary figure(s) 1B.
3. Please note that the box plots need to be defined in terms of minima, maxima, centre, bounds of box and whiskers, and percentile in the legends of supplementary figures 2A, C.
4. Please note that information related to n is missing in the legends of figures 2B, 3E, F; EV1 G, EV2 I, supplementary figure 2C.
5. Please note that the error bars are not defined in the legends of figures 2D, E; 3A, B, D, F, G; 5A, B, E, K; 6D-F; EV1 A, E, G, H, I, J; EV2 E, G, H, I; EV3 B, D, E, I, K; EV4 D, EV5 C, D; supplementary figure 1B.
6. Please note that scale bar and its definition are missing for figure EV2 A.

I would like to suggest some changes to the title and abstract. Please let me know whether you agree with the following:

A CRISPR-Cas9 Screen Reveals Genetic Determinants of the Cellular Response to Decitabine

Decitabine (DAC), a well-recognized DNA hypomethylating agent, has been applied to treat acute myeloid leukemia. However, clinical investigations revealed that DNA methylation reduction does not correlate with a clinical response, and relapse is prevalent. To gain a better understanding of its anti-tumor mechanism, we perform a temporally resolved CRISPR-Cas9 screen to identify factors governing the DAC response. We show that DNA damage generated by DNMT-DNA adducts and 5-aza-dUTP misincorporation through DCTD [Please explain what DCTD is] acts as driver of DAC-induced acute cytotoxicity. The DNA damage that arises during the next S phase is dependent on DNA replication, unveiling a trans-cell cycle effect of DAC on genome stability. By exploring candidates for synthetic lethality, we unexpectedly uncover that KDM1A promotes survival after DAC treatment through interactions with ZMYM3 and CoREST, independent of its demethylase activity or regulation of viral mimicry. These findings emphasize the importance of DNA repair pathways in the DAC response and provide potential biomarkers.

I also made a few minor changes to the short summary and bullet points. Do you agree with:

A temporally resolved CRISPR-Cas9 screen reveals DAC-induced DNA damage drives trans-cell cycle cytotoxicity that depends on DNA replication. KDM1A promotes survival in DAC-treated cells via DNA repair and independent of its demethylase activity.

- A temporally resolved CRISPR-Cas9 screen provides a comprehensive analysis of factors influencing the DAC response.
- DNMT-DNA adducts and 5-aza-dU, produced by DCTD, are sources of DAC-induced DNA damage.
- DAC-induced DNA damage results in trans-cell cycle cytotoxicity that is dependent on DNA replication.
- KDM1A promotes survival after DAC treatment by facilitating demethylase-independent DNA repair through interactions with ZMYM3 and CoREST.

The items and text on the synopsis image at its final size of 550x470 pixels are a little too small. Can you please send us an improved synopsis image at its final size?

Referee #1:

In the revised manuscript, Zhang et al. have significantly improved their study.

The authors included new data demonstrating that DNA lesions caused by DAC require a subsequent round of DNA replication in the next S phase to induce DNA breaks. While this observation is logical, as DAC-induced lesions are formed behind replication forks, it is surprising and intriguing that these lesions persist in some forms until the next S phase and create issues during the following round of DNA replication.

Furthermore, the authors have strengthened their model that KDM1A promotes cellular survival in response to DAC through its role in DNA repair by providing the following additional evidence:

- They clarified the importance of KDM1A in DNA repair by pointing out the delayed γ H2AX resolution in KDM1A knockout cells.
- They demonstrated that KDM1A localizes to DNA damage sites in DAC-treated cells during the subsequent S phase.
- They elucidated the mechanism of KDM1A recruitment to DNA damage sites after DAC treatment, showing that it depends on ZMYM3 as well as the SWIRM and TOWER domains of KDM1A.

The revised manuscript is a substantial improvement, and I believe it is now suitable for publication in EMBO Reports in its present form.

Referee #2:

This is a well conducted study that addresses important cellular mechanisms of relevance to cancer therapy. The authors have successfully addressed most of the issues I raised relating to the original submission. Consequently, the revised manuscript is significantly enhanced from the original submission.

I feel like overall the authors have been able to rule more things out rather than nailing down mechanistic details, but there is value in this.

I therefore think that this work is of sufficient quality and general interest to warrant publication in EMBO Reports.

All editorial and formatting issues were resolved by the authors.

Jun Xiong
Institute of Biophysics, Chinese Academy of Sciences
Key Laboratory of Epigenetic Regulation and Intervention
15 Datun Road
Chao Yang District
Beijing, Beijing 100101
China

Dear Dr. Xiong,

I am very pleased to accept your manuscript for publication in the next available issue of EMBO reports. Thank you for your contribution to our journal.

Yours sincerely,
